

# BPS chaos

Yiming Chen, Henry W. Lin and Stephen H. Shenker

Stanford Institute for Theoretical Physics, Stanford University, Stanford, CA 94305, USA

## Abstract

Black holes are chaotic quantum systems that are expected to exhibit random matrix statistics in their finite energy spectrum. Lin, Maldacena, Rozenberg and Shan (LMRS) have proposed a related characterization of chaos for the ground states of BPS black holes with finite area horizons. On a separate front, the "fuzzball program" has uncovered large families of horizon-free geometries that account for the entropy of holographic BPS systems, but only in situations with sufficient supersymmetry to exclude finite area horizons. The highly structured, non-random nature of these solutions seems in tension with strong chaos. We verify this intuition by performing analytic and numerical calculations of the LMRS diagnostic in the corresponding boundary quantum system. In particular we examine the 1/2 and 1/4-BPS sectors of $\mathcal{N} = 4$ SYM, and the two charge sector of the D1-D5 CFT. We find evidence that these systems are only *weakly* chaotic, with a Thouless time determining the onset of chaos that grows as a power of $N$. In contrast, finite horizon area BPS black holes should be *strongly* chaotic, with a Thouless time of order one. In this case, finite energy chaotic states become BPS as $N$ is decreased through the recently discovered "fortuity" mechanism. Hence they can plausibly retain their strongly chaotic character.

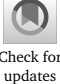

# 1 Introduction

## 1.1 Review and context

Black holes are strongly chaotic quantum systems. One indication of chaos is the random matrix behavior of the spectrum of black hole microstates [1–3]. As in any chaotic many-body quantum system, one expects the spectrum to display random matrix statistics, a universal pattern of repulsion between eigenvalues. One diagnostic is the level spacing histogram. Given the precise energy levels $E_1 < E_2 < E_3 < \dots$ in some microcanonical window, level repulsion implies that the distribution $P(s)$ of the differences between adjacent levels $s_i = E_{i+1} - E_i$, after suitable unfolding that removes the overall density of states, should be well approximated by

a universal distribution corresponding to the associated symmetry class,[1] which goes to zero at $s = 0$ and has a small tail at large $s$. On the other hand, for an integrable systems that lacks level repulsion, the distribution starts out large at $s = 0$ and has a heavier tail, which for uncorrelated eigenvalues is given by the Poisson distribution [4]:

$$\text{density of nearest-neighbor eigenvalue spacings} = \qquad\qquad\qquad\qquad (1)$$

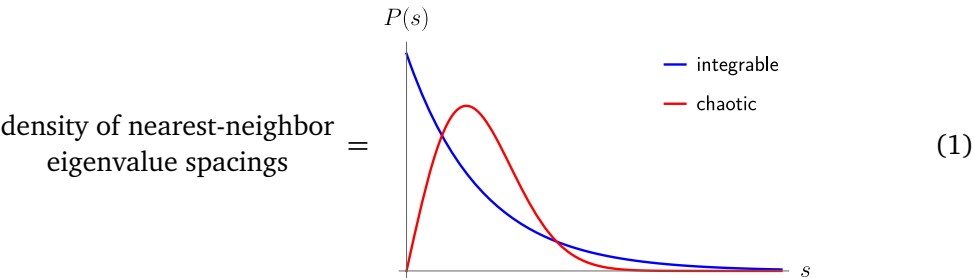

The distribution of level spacings only probes the existence of level repulsion at the finest possible scale, namely $\delta E \sim e^{-S}$, where $S$ is the entropy. One can further quantify the quality of random matrix universality by asking how far apart in the spectrum level repulsion extends, which defines an energy scale $E_{\text{Thouless}}$ [5,6]. One can extract this quantity by studying the spectral form factor [3,7,8], in which the level repulsion gets reflected in a linear ramp at late time, and the inverse of the Thouless energy, i.e. the Thouless time $t_{\text{Thouless}}$ corresponds to the time where the linear ramp starts [9–12].

The notion of Thouless time provides a way to distinguish weakly chaotic systems, such as a gas of weakly interacting gravitons, from strongly chaotic quantum systems. In the graviton gas, for example, we would expect the Thouless time to be of order the collision time between gravitons, of order $1/G_N \sim N^2$, the number of degrees of freedom in the system. On the other hand, strongly chaotic systems should have a much smaller Thouless time. In particular, one would expect generic black holes to have a Thouless time that is of order one, which can be seen by studying the universal double-cone wormhole configuration in gravity and fluctuations around it [13–15].[2]

Rather than studying generic black holes, the focus of this paper will be an extreme limit where the temperature of the black hole goes to zero. In particular, we will consider theories with supersymmetry, where at zero temperature one can have a large degenerate subspace of BPS states.[3] This is an interesting limit to consider since there exist completely different physical descriptions for the BPS subspaces. In some cases, certain BPS states are described by smooth horizonless geometries, sometimes called "microstate geometries". These geometries are best understood when they preserve many supersymmetries. There are a plethora of examples of this kind. Some of the examples relevant to our later discussion include the 1/4-BPS Lunin-Mathur (LM) geometries in $\text{AdS}_3 \times S^3 \times T^4$ [17], the 1/2-BPS Lin-Lunin-Maldacena (LLM) geometries in $\text{AdS}_5 \times S^5$ [18] and their 1/4-BPS generalizations [19–23]. It has been an active research endeavor in trying to find more such solutions, particularly in situations with less supersymmetries, see [24] for a review. A general feature of such geometries is that unlike black holes, there is no horizon and there is a large phase space explicit already at the supergravity level, which to a large extent reflects the Hilbert space of the microstates (sometimes also called fuzzball states[4]).

---

[1]This is often approximated by the "Wigner Surmise." Concretely, $P_{\text{GUE}}(s) \approx \frac{32}{\pi^2} s^2 e^{-\frac{4}{\pi} s^2}$, $P_{\text{GOE}}(s) \approx \frac{\pi}{2} s e^{-\frac{\pi}{4} s^2}$, where GUE/GOE stand for Gaussian unitary/orthogonal ensemble, respectively.

[2]Note that the Thouless time in the SYK model is of order $\log N$ [13]. A bulk explanation for this is the existence of $N$ bulk matter fields, as opposed to the order one number in conventional holographic systems.

[3]In theories without supersymmetry, quantum effects become important near extremality and one does not get a large number of states, see [16] for a review of recent developments.

[4]For a review of the fuzzball program, see [24–26] and [27] for a critical take.

On the other hand, in some other cases, usually with less supersymmetries and more entropy,[5] the black hole picture persists in the BPS limit. The supersymmetric black holes have macroscopic horizons, similar to the Reissner-Nordström black holes.[6] Examples of this type include the three-charge black holes whose entropy was first counted by Strominger and Vafa [29], 1/16-BPS black holes in $AdS_5 \times S^5$ [30–33], and so on. Such black holes generically come with an $AdS_2$ throat and are expected to be governed by an effective $\mathcal{N} \geq 2$ super-Schwarzian theory [34, 35].

Our goal is to understand whether the horizonless geometries and macroscopic black holes are genuinely distinct descriptions of various BPS subspaces, or they are in fact similar and the difference only arises due to our inability to resolve the microstructure of horizons. We will attempt to argue that they are genuinely different, as one can separate the two cases cleanly using the notion of chaos. However, before we can make our case, we need to first review the notion of chaos in the BPS subspace.

## 1.2 BPS chaos

The usual ways to diagnose chaos do not apply once we restrict our attention to the BPS subspace. Once we fix the charges, all the BPS states are constrained to have the same energy and therefore do not exhibit level repulsion. At first glance one might think that these ground states cannot be chaotic. However, this is not necessarily the case as one can still probe the chaos in other ways. A possible definition of chaos for BPS states was discussed by Lin, Maldacena, Rozenberg and Shan (LMRS) [36, 37]. Pick a "simple" operator $O$ in the UV theory. (Here "simple" could mean a primary operator that corresponds to a particular supergravity mode; we will further discuss the meaning of "simple operator" shortly.) Then project $O$ into the BPS subspace to obtain a new operator $\hat{O}$:

$$\hat{O} = P_{\text{BPS}} O P_{\text{BPS}}. \tag{2}$$

We will call the projected operator the LMRS operator. It was proposed by LMRS that whether the BPS subspace is chaotic or not can be seen by studying the properties of $\hat{O}$. We formulate their proposal as

**LMRS criterion:**   chaotic BPS subspace $\longleftrightarrow$ $\hat{O}$ shows random matrix behavior.

We note that the random matrix properties of $\hat{O}$ can now be tested with standard methods, such as level spacings, spectral form factors, etc, and similar notions such as Thouless time generalize straightforwardly. In particular, a BPS subspace with *strong* chaos requires that $\hat{O}$ has a Thouless time of order one. It is expected that whether the BPS subspace shows strong chaos does not rely on which simple operator one picks, even though the properties of $\hat{O}$ might differ in details given different choices of $O$.

One way to understand this proposal intuitively is to notice that it is a generalization of the Eigenstate Thermalization Hypothesis (ETH) [38–40] to the case of a degenerate subspace. In the usual ETH discussion, the matrix element of a simple operator $O$ between two eigenstates with $E_i \approx E_j \approx E$ takes the form

$$\langle E_i | O | E_j \rangle = f(E)\delta_{ij} + e^{-S(E)/2} R_{ij}, \tag{3}$$

where $f(E)$ is a smooth function of energy and the noise term contains a matrix $R_{ij}$ that has random matrix statistics.[7] Applying (3) to (2) where the microcanonical window is now taken

---

[5]Note that this rough characterization is not always true. For example, the three-charge microstate geometries superstrata [28] preserve the same amount of supersymmetries as the three-charge black holes.

[6]By macroscopic, we mean a horizon whose size is parametrically larger than the string scale or Planck scale.

[7]We do not mean that $R_{ij}$ is precisely Gaussian. Small non-Gaussianities can significantly affect the moments of $O$, which can lead to a non-semicircle distribution, e.g., (8). See [41] for related discussion.

to be a degenerate subspace, we see that (up to a constant term) $\hat{O}$ should display random matrix behavior.

To further see why the BPS subspace can have ETH behavior, we note that in the situation that we have a random matrix ensemble with supersymmetries, the BPS subspace will be a random hyperplane with respect to a given simple basis. Although it is not directly related to our main discussion, we can consider the simplest case with $\mathcal{N} = 1$ supersymmetry [42] and ask whether one can see that the supersymmetric ground states are "random". Here $H = Q^2$ and in a basis where $(-1)^F$ is block diagonal, the supercharge takes the form

$$Q = \begin{pmatrix} 0 & \mathcal{Q} \\ \mathcal{Q}^\dagger & 0 \end{pmatrix}, \tag{4}$$

where $\mathcal{Q}$ is an $L_b \times L_f$ rectangular matrix, $L_b, L_f$ are the total number of bosonic and fermionic states and we assume $L_f - L_b = \nu > 0$ for convenience. In a pure random matrix theory with $\mathcal{N} = 1$ supersymmetry [42], $\mathcal{Q}$ is a rectangular random matrix and the measure is invariant under $\mathcal{Q} = U_{L_b}^{-1} \mathcal{Q} U_{L_f}$ where $U_{L_b}, U_{L_f}$ are elements of $U(L_b), U(L_f)$. In each member of the ensemble, we can use the unitaries to put $\mathcal{Q}$ into the following form

$$U_{L_b}^{-1} \mathcal{Q} U_{L_f} = \begin{pmatrix} \lambda_1 & & & 0 & \\ & \dots & & & \dots \\ & & \lambda_{L_b} & & 0 \end{pmatrix}, \tag{5}$$

and the wavefunction of the supersymmetric ground states are given by the last $\nu$ columns of $U_{L_f}$. Since $U_{L_f}$ are random unitaries in the ensemble, in the regime where $\nu \ll L_f, L_b$, namely when the number of supersymmetric ground states is much smaller than the total dimension of the Hilbert space, the last $\nu$ columns of $U_{L_f}$ are essentially random vectors.[8] This implies that they should behave as if they are generic high energy states, and particularly the matrix element of $O$ between two BPS states will have the same behavior as in (3). In [44] by Turiaci and Witten, random matrix ensembles with $\mathcal{N} = 2$ supersymmetries were constructed and we expect a similar argument to go through there, namely the BPS states form an ensemble of random vectors and the projection $P_{\text{BPS}}$ becomes the projection into a random hyperplane. The case of $\mathcal{N} = 2$ random matrix theory is of direct interest to us since it was proposed in [35] that its gravity dual $\mathcal{N} = 2$ super-JT governs the low temperature limit of 1/16-BPS black holes in $\text{AdS}_5 \times S^5$.

The above discussion has been fairly general; let us now specialize to the case where the UV theory is a CFT in $d$-dimensions. In the situation where the BPS states are given by superconformal primaries, we can write the matrix element (3) as an OPE coefficient:[9]

$$\langle O_i | O | O_j \rangle = C_{iOj}. \tag{6}$$

For a CFT with a bulk dual, one can imagine computing this OPE coefficient using the microstates for $i$ and $j$, if they are known to sufficient precision. Alternatively, one can obtain statistical information about these OPE coefficients if the BPS subspace is described by an extremal black hole.

Let's now review that for black holes with an $\text{AdS}_2 \times X$ throat, one can use the super-Schwarzian theory to extract moments of these OPE coefficients which display strong chaos [36, 37]. One can implement the projection onto the BPS sector by evolving with the ADM

---

[8]If $\nu$ is comparable to $L_f$, then correlations among the vectors become important. We will not be interested in such situations, but see [43] for relevant discussion.

[9]This assumes that $O$ is a primary. Note that even if $O$ is not a primary, we can expand it in terms of primaries and descendants $O = \sum_\Delta O_\Delta$. Then the 3-pt function with large Euclidean separations will pick out the operator $O_\Delta$ with the smallest conformal dimension.

Hamiltonian by an infinite amount of Euclidean time, e.g., $P_{\text{BPS}} = e^{-\infty H}$ where $H = \{Q, Q^\dagger\}$. When we go to these long Euclidean times, the super-Schwarzian mode dominates over all other fluctuations in the metric and thus the moments of this operator can be computed by gravitationally dressing the appropriate Witten diagram and then integrating over the soft modes:[10]

$$\sum_{\substack{i_1, i_2, \cdots, i_6 \\ \text{BPS primaries}}} C_{i_1 O i_2} \cdots C_{i_6 O i_1} \sim \text{Tr}_{\text{BPS}}\left(\widehat{O}_\Delta^6\right) \quad = \quad$$ 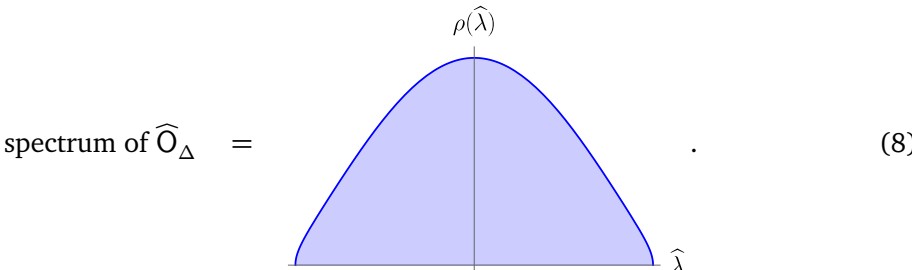 $$. \tag{7}$$

Here the red blob represents any gravitationally dressed Witten diagram that contributes to the $n$-pt function (for simplicity, we drew $n = 6$). Furthermore, $O_\Delta$ denotes an AdS$_2$ primary of AdS$_2$ dimension $\Delta$, which is not quite the same as the AdS$_{d+1}$ primary; in general, an AdS$_{d+1}$ primary will mix with many AdS$_2$ primaries $O = \sum_\Delta c_\Delta O_\Delta$; the dominant contribution will be from the lightest $O_\Delta$. Ultimately, this is the criteria for an operator $O$ to be "simple"; in the gravity context it should flow to an AdS$_2$ primary $O_\Delta$ with some $\Delta \sim O(1)$.

From the moments of the operator we can infer the spectrum of its eigenvalues. Qualitatively,[11]

$$\text{spectrum of } \widehat{O}_\Delta \quad = \quad$$

$$\rho(\widehat{\lambda})$$

$$\widehat{\lambda} \tag{8}$$

The width of this spectrum is $\sim 1/S_0^\Delta$, where $S_0$ is the extremal entropy. This width can be understood as being a consequence of the typical length of the 2-sided wormhole, which is $\ell \sim 2 \log S_0$, since $\langle \hat{O}_\Delta \hat{O}_\Delta \rangle \sim \langle e^{-\Delta \ell} \rangle \sim 1/S_0^{2\Delta}$. Notice that the typical length of the wormhole is only finite due to quantum Schwarzian corrections [36, 37]. Classically, an extremal black hole has an infinite length throat, but this is no longer true once quantum corrections are taken into account.

As we have already commented on, this spectrum is expected to exhibit eigenvalue repulsion of the type expected from random matrix theory. From the AdS$_2$ gravity point of view, one can imagine computing the spectral form factor of the projected operator $\hat{O}_\Delta$. In other words, one imagines "evolving" the system by $e^{iu\hat{O}_\Delta}$. One can compute $Z(u) = \text{Tr}_{\text{BPS}} e^{iu\hat{O}_\Delta}$ by expanding down the exponential. When one computes $\langle Z(u)Z^*(u) \rangle$ there is a spacetime wormhole contribution where the matter propagates from side of the wormhole to the other. In random matrix theory, it is natural to normalize the operator $\widehat{O}_\Delta^{\text{RMT}} = \frac{1}{Z}\widehat{O}_\Delta$ by requiring that $\text{Tr}_{\text{BPS}}(\widehat{O}_\Delta^{\text{RMT}})^2 \propto \dim \mathcal{H}_{\text{BPS}}$. With this normalization, the above gravity computation leads to a linear ramp with an order one Thouless time.

---

[10]The long Euclidean time is schematically represented by the very wiggly boundary.

[11]When $\Delta \gg 1$, it was argued that the distribution becomes a semi-circle. For general $\Delta$, it is expected that the projected operator behaves like a random matrix with a more general matrix potential.

## 1.3  Conjecture and plan for the paper

We've reviewed that for BPS sectors that are described by macroscopic supersymmetric black holes, since the BPS limit is governed by the Schwarzian theory, the BPS subspace should be strongly chaotic under the LMRS criterion. On the other hand, one might expect a very different behavior for the horizonless geometries. A classical analogue of the statement of whether the projection $P_{\text{BPS}}$ being random or not is how the classical phase space of the BPS configurations are embedded in the full supergravity phase space. In many known horizonless geometries, such an embedding is "simple", in the sense that the BPS configurations are only characterized by one or a few smooth profiles of supergravity fields. For this reason, it seems highly unlikely that upon quantization, the BPS subspace can exhibit strong chaos. We will support this intuition through many explicit computations in latter parts of this paper.

Therefore, we propose that within the BPS subspaces, one can use quantum chaos as a tool to cleanly separate horizonless geometries from macroscopic black holes. We formulate this idea as the following conjecture

> **Conjecture:** BPS subspaces that are described by supersymmetric black holes with macroscopic horizons exhibit strong LMRS chaos; those that are described by horizonless geometries do not exhibit strong LMRS chaos.

Let's highlight a few details in this conjecture. First, it is important to note the word "strong", which means a Thouless time of the LMRS operator that is of order one. As we will see in explicit examples in Section 3 and 6, horizonless geometries can indeed carry weak chaos. We emphasize that the difference between weak and strong chaos is sharp in large $N$ systems. Second, here we are only making a conjecture about BPS configurations. In non-BPS situation, the situation appears to be more subtle. One possible counter-example to a naive generalization of the conjecture to non-BPS situations is the Horowitz-Polchinski solution [45], which is a horizonless geometry describing the self-gravitating highly excited fundamental strings. In [46] it was suggested that there could be a smooth crossover between this solution and the black hole solution in the Heterotic string theory, and therefore one would expect it to also exhibit strong chaos (for example, an order one Lyapunov exponent). It would be interesting to understand whether this is true. The solution breaks down near the BPS bound so it is not a counter-example to the conjecture above.

If our conjecture holds, it means that there is a genuine difference between horizonless geometries and macroscopic black holes at the quantum level, and seeing this difference does not require us to go out of the BPS sector. In other words, the set of fuzzball states that one can get by quantizing the phase space of horizonless geometries have very different properties compared to the BPS black hole microstates. We should note that, however, sometimes the notion of fuzzball states is used in a broader sense. They are not necessarily those states that come from quantizing horizonless geometries, but could refer to highly quantum object that replace the horizon at the horizon scale [24]. Our conjecture does not directly rule out this possibility, but puts further constraint that the fuzzball configurations will have to mimic the strong chaos that BPS black holes have. It remains a challenge for the fuzzball idea to demonstrate this in a controllable set-up.

Recently, there has been an important progress [47] in the understanding of BPS states, particularly in the case of $\mathcal{N} = 4$ SYM. It was shown that the BPS states can be classified into two types, fortuitous and monotone, based on their behavior as one varies $N$. In particular, it was conjectured in [47] that monotone BPS states are dual to smooth horizonless geometries, and fortuitous ones are responsible for typical black hole microstates. In Section 5, we review this idea and related progress in more detail, and point out a hint for how fortuity and chaos are related. It would be interesting to connect two conjectures at a more quantitative level.

The rest of the paper will be devoted to verifying our conjecture in various cases.

In Section 2 through Section 5, we will be studying the case of 4d $\mathcal{N} = 4$ SU($N$) SYM, which is dual to Type IIB string theory in AdS$_5 \times S^5$. We will study the properties of several BPS sectors at weak 't Hooft coupling, in light of the LMRS criterion. In Section 2, we will provide some review of the various BPS sectors in the theory. In Section 3, we will study the 1/2-BPS sector. In Section 4, we will study the weak coupling perturbation theory of the 1/4-BPS sector. Both of these sectors are known to be described by horizonless geometries. As we will show, neither of them display strong chaos under the LMRS criterion. In Section 5, we make some comments about the 1/16-BPS sector which does contain black holes. We use the fortuity idea to give a plausible argument for why it can exhibit strong chaos.

In Section 6, we move our attention to the D1-D5 CFT and study the 1/2-BPS states that are described by the two charge microstate geometries. The discussion is similar to the discussion of the 1/2-BPS sector in $\mathcal{N} = 4$ SYM in some aspects, so the readers can also choose to read Section 6 together with Section 3 and then proceed to the other sections.

In Section 7 we conclude and discuss some generalizations.

## 2 BPS states in the weak coupling limit of $\mathcal{N} = 4$ SYM

In the next few sections, we will study the properties of BPS operators 4d $\mathcal{N} = 4$ SYM with gauge group SU($N$), in the weak coupling limit where the 't Hooft coupling $\lambda = g_{\text{YM}}^2 N \ll 1$. Throughout this paper, we will work in the one-loop approximation where we study the perturbation theory at leading order in $\lambda$. We will explain the precise meaning of this approximation momentarily. Even though the weak coupling limit cannot be directly compared to the bulk supergravity picture except few quantities that are protected by supersymmetries, it still provides a useful arena to probe various questions about chaos. One expects that generic high energy states in $\mathcal{N} = 4$ SYM to be strongly chaotic as long as $\lambda$ is nonzero. Therefore, if indeed there are differences in chaos between various BPS sectors at strong coupling, we expect to be able to see the differences already in the weak coupling expansion.

The $\mathcal{N} = 4$ SYM theory has conformal symmetry SO(2, 4) and $R$-symmetry SU(4)$_R$. In addition, it has Poincare supercharges $Q^{\alpha,i}, \bar{Q}_i^{\dot{\alpha}}$ together with superconformal generators $S_{\alpha j}, \bar{S}_{\dot{\alpha}}^j$, where $\alpha = \{+, -\}, \dot{\alpha} = \{\dot{+}, \dot{-}\}$ and $i, j = 1, ..., 4$ are indices under the Lorentz group and R-symmetry group, respectively. These together form the superconformal group PSU(2, 2|4). We are interested in studying the theory on $S^3 \times R_t$, where we fix the radius of the 3-sphere to be one. The conformal dimension/energy $E$ of a state is given by the eigenvalue of the dilatation operator $\mathcal{D}$. We will denote the two Cartan generators for the SO(4) rotations of $S^3$ as $J_1, J_2$, and the three Cartan generators for SO(6) $\cong$ SU(4)$_R$ as $R_1, R_2, R_3$.

In the free theory, we can construct operators in the theory by forming gauge invariant "words" out of "letters"

$$\text{Tr}[a_1...a_{L_1}]\text{Tr}[b_1...b_{L_2}]...\text{Tr}[c_1...c_{L_k}], \tag{9}$$

where the letters $\{a_i, b_i, c_i, ...\}$ are chosen from fields transforming in the adjoint representation and their derivatives. We will call such operators "multi-traces". A general operator can be written as a linear superposition of them. The field content of $\mathcal{N} = 4$ SYM contains a vector field $A_\mu$, six scalars $\Phi_{ij} = -\Phi_{ji}$ ($\bar{\Phi}^{ij} = \frac{1}{2}\epsilon^{ijkl}\Phi_{kl}$) and chiral fermions $\Psi_{i\alpha}, \bar{\Psi}_{\dot{\alpha}}^i$. Therefore, a basis of operators and equivalently a basis of states under state-operator correspondence are given by product of traces of these fields and derivatives $\partial_{\alpha\dot{\beta}}$ acting on them.

In the free theory, the energy $E$ of an operator of the form (9) is simply the sum over the dimensions of the letters. At low energy $1 \ll E \ll N^2$, the number of the operators grows exponentially with $E$. However, after a Hagedorn transition at energy $E \sim N^2$, the trace basis becomes highly redundant due to trace relations and the growth of number of states becomes black-hole like, as $\exp(N^2 s(E/N^2))$ where $s(x)$ is an order one function that is slower than linear growth [48, 49]. We will be interested in the latter regime, namely $E \sim N^2$.

The spectrum of the free theory has exponentially large degeneracies at high energies. However, once we turn on a non-zero 't Hooft coupling $\lambda$, these degeneracies are generically lifted as the operators develop anomalous dimensions. At weak coupling, the dilatation operator $\mathcal{D}$ has expansion[12]

$$\mathcal{D} = \mathcal{D}_0 + g^2 \mathcal{D}_2 + \mathcal{O}(g^4). \tag{10}$$

Therefore, to find the new energy levels and their corresponding wavefunctions at order $g^2 \sim \lambda$, one has to study the degenerate perturbation theory in which the first step is to diagonalize the one-loop dilatation operator $\mathcal{D}_2$ in the subspace of states with the same classical dimension $E_0$, namely the conformal dimension in the free theory. In fact, this is the only necessary step, since the one-loop dilatation operator commutes with the classical piece [50]

$$[\mathcal{D}_0, \mathcal{D}_2] = 0. \tag{11}$$

In other words, operators with different classical dimensions do not mix under the perturbation theory, therefore it suffices to restrict to a subspace with a particular dimension. However, it is important to note the range of validity of this perturbation expansion. One expects it to break down when the splitting of the degenerate eigenvalues becomes comparable to the gap in the free theory, since by that point, mixing between states with different classical dimensions become "inevitable" in order to avoid level crossings.[13] Therefore, one needs to consider $\lambda$ small enough such that the splitting is small compared to the gap in the free theory in order for this approximation to be valid. We will comment more on this in Section 4.

In general, this degenerate perturbation theory can be difficult to study due to the large number of letters involved. However, the situation is improved if we focus on states that preserve certain amount of supersymmetries. As implied by the discussion above, in the one-loop approximation, in order to find the BPS operators, we only need to study the degenerate perturbation theory in the subspace of words that saturate the BPS bounds in the free theory. These words are built out of only letters which saturate the BPS bounds. This cuts down the number of letters involved in the problem.

Depending on the amount of preserved supersymmetries, we can separate the discussion into various different sectors. Even though our discussion will not cover all the possible sectors, we will study three different representative cases that each has distinct features. In the order of increasing complexity, the cases we consider are

- $\frac{1}{2}$-**BPS sector**. In the free theory, the half-BPS subspace is spanned by all the multi-traces that are built out of letter $Z \equiv \bar{\Phi}^{43}$, such as $\mathrm{Tr}[Z^2], \mathrm{Tr}[Z^3]\mathrm{Tr}[Z^5], ...,$ etc. These operators preserve half of the supersymmetries, as indicated by the name. The letter $Z$ has $E = R_3 = 1$, $R_1 = R_2 = J_1 = J_2 = 0$.[14] In the interacting theory, these operators remain BPS, namely $\mathcal{D}_2$ as well as higher order terms in the dilatation operator vanish. Therefore, the one-loop problem is trivial in this sector. Due to this simplification, we are able to study the LMRS problem analytically in this sector, which we discuss in Section 3.

- $\frac{1}{4}$-**BPS sector**. In the free theory, we consider the linear space spanned by all the multi-traces that are built out of letters $Z$ and $X \equiv \bar{\Phi}^{41}$, such as $\mathrm{Tr}[ZXZX], \mathrm{Tr}[ZXZ]\mathrm{Tr}[ZZXX]$, ..., etc. The letter $X$ has $E = R_1 = 1$, $R_2 = R_3 = J_1 = J_2 = 0$. Due to the different ways of ordering $Z$ and $X$ inside a trace, the number of states has a Hagedorn growth at low energy.

---

[12]We adopt the notation in [50], where $g^2 \equiv g_{\mathrm{YM}}^2 N/(8\pi) = \lambda/(8\pi)$.

[13]The resolution of the level crossing goes beyond $\lambda$ perturbation theory, but can be studied through a numerical bootstrap approach, see [51] for recent discussion.

[14]Here and in the following, we always use the letters to denote the spatial zero mode of the field on $S^3$. The higher Kaluza-Klein modes are non-BPS.

A generic operator built out of $Z$ and $X$ is not protected and develops anomalous dimension in the interacting theory. The one-loop dilatation operator is given by [52]

$$\mathcal{D}_2 = -\frac{1}{N} :\text{Tr}\big[[Z,X][\check{Z},\check{X}]\big]: ,\tag{12}$$

where the symbols $\check{Z},\check{X}$ are derivatives carrying matrix indices,

$$\check{Z}_{ij}Z_{kl} = \check{X}_{ij}X_{kl} = \delta_{il}\delta_{jk} - \frac{1}{N}\delta_{ij}\delta_{kl}, \quad \check{Z}X = \check{X}Z = 0,\tag{13}$$

and $::$ indicates normal ordering, i.e. the derivatives do not act on the $[Z,X]$ inside the dilatation operator. The null space of $\mathcal{D}_2$ contains the one-loop quarter-BPS states, which preserve a quarter of the supersymmetries.

In the planar limit where $N \to \infty$, when acting the dilatation operator (12) on a single trace operator, the output remains a single trace. Therefore, one can map a general single trace operator to a spin chain configuration, where $Z$ and $X$ are $|\uparrow\rangle$ and $|\downarrow\rangle$, respectively. The corresponding spin chain Hamiltonian is integrable and solvable using the Bethe Ansatz [52–54] (see review [55] for more references). The integrability results imply that in the $N \to \infty$ limit, the system is not chaotic even at finite $\lambda$, and we expect this non-chaotic behavior applies to the BPS subspace as well.

On the other hand, we are interested in the opposite limit, that we take $N$ to be finite and consider a subspace with $E_0 \sim N^2$. In this limit, the splitting and joining of traces are no longer suppressed. In fact, the number of traces is no longer a meaningful notion due to the existence of trace relations that relate words with different number of traces. For this reason, the spin chain description is no longer valid and one has to rely on other approaches, as we will review in Section 4. Our approach will be mostly numerical, following [56]. We study the diagonalization of $\mathcal{D}_2$ and test the LMRS criterion numerically in Section 4.

- $\frac{1}{16}$-**BPS sector**. This sector preserves the least supersymmetries and is the only sector that contains an order $N^2$ entropy that match with the entropy of a macroscopic black hole [57–59]. In the free theory, one should include the following letters that saturate the BPS condition $E = J_1 + J_2 + R_1 + R_2 + R_3$,

$$\bar{\Phi}^{4m}, \quad \psi_{m+}, \quad f_{++} = (\sigma^{\mu\nu})_{++}F_{\mu\nu}, \quad \bar{\lambda}_{\dot{\alpha}} = \bar{\Psi}^4_{\dot{\alpha}}, \quad \partial_{+\dot{\alpha}}, \quad m = 1,2,3.\tag{14}$$

The supercharges as well as the one-loop dilatation operators can be written in a compact form using a superspace formulation in [60]. The one-loop problem has been studied numerically in recent work including [61,62]. The increasing number of letters in the 1/16-BPS sector makes the numerical problem more challenging than the quarter-BPS sector, which limits the potential to extract meaningful data about the LMRS criterion. Instead, in Section 5, we present an indirect argument for the existence of chaos that utilizes the existence of fortuitous BPS operators in this sector [47].

In order to test the LMRS criterion, we need to choose a specific simple operator $O$. In the large $N$ limit of $\mathcal{N} = 4$ SYM, one can practically choose the simple operator to be an operator that only involves an order one number of letters. In principle, one would like to choose $O$ to be itself a conformal primary at one-loop and in such cases, the matrix element will be proportional to an OPE coefficient

$$\langle O_{\text{BPS},i} | O | O_{\text{BPS},j} \rangle \propto C_{O,i,j},\tag{15}$$

and the LMRS problem translates into studying the statistical properties of OPE coefficients, viewed as a matrix in an orthonormal basis of BPS states. Once we have the explicit forms of the BPS operators from the degenerate perturbation theory, the leading answer to the OPE coefficient can be simply evaluated using the free field Wick contractions. In other words, the only place the interaction plays a role is when we diagonalize the one-loop dilatation operator.

We will consider $P_{\text{BPS}}$ to be a projection into BPS subspace with fixed charges. Therefore, in order for $P_{\text{BPS}}OP_{\text{BPS}}$ to be nonzero, $O$ should carry zero net charge.[15] Some natural choices for such operators are

- Bound states of gravitons, such as $\frac{1}{N^2}$ :$\text{Tr}[\bar{Z}^2]\text{Tr}[Z^2]$: .

- Operators involving massive string states, such as $\frac{1}{N}$ :$\text{Tr}[Z\bar{Z}\bar{X}...\bar{Z}XZ]$: .

We require the operators to only include an order one number of letters, which can be viewed as a practical definition of "simple". Both these choices are generically non-BPS and a conformal primary operator in the interacting theory is generally a mixture of many such operators.

Consider the first option, a bound state of two gravitons that carry opposite angular momentum in $S^5$ such as $\frac{1}{N^2}$ :$\text{Tr}[\bar{Z}^2]\text{Tr}[Z^2]$:. This operator is *non*-BPS as the two gravitons interact and develops binding energy. This is reflected in the fact that the simple operator itself also gets renormalized at one loop. At one loop, the composite operator $\frac{1}{N^2}$ :$\text{Tr}[\bar{Z}^2]\text{Tr}[Z^2]$: can mix with other operators such as $\frac{1}{N}$ :$\text{Tr}[\bar{Z}^2 Z^2]$:, $\frac{1}{N}$ :$\text{Tr}[Z\bar{Z}Z\bar{Z}]$:, etc.[16] This mixing problem was studied in details in [63]. We note that, however, such mixings will be further suppressed by $1/N^2$ in the large $N$ limit since the two gravitons only interact weakly at small $G_N \sim 1/N^2$. Therefore, when we study operators of this kind, we will work in the approximation that we ignore the $1/N^2$ corrections terms, which simplifies the analysis significantly and sometimes allows us to find the spectrum of the LMRS operator analytically. We will comment more on the effect of these suppressed terms in Section 3.2. Similarly, in the discussion of the 1/4-BPS case in Section 4, we will study the following double trace operator as an example for the simple operator,

$$O = \frac{1}{N^2}\left(:\text{Tr}[Z^2]\text{Tr}[\bar{Z}^2]: + :\text{Tr}[X^2]\text{Tr}[\bar{X}^2]: +2 :\text{Tr}[ZX]\text{Tr}[\bar{Z}\bar{X}]:\right). \tag{16}$$

We will explain further the motivation for this particular choice in Section 4.

We also study operators of the second kind, i.e. those that correspond to massive string states, and give an argument for why they should at most lead to weak chaos. As an explicit toy example, in Section 3.3, we study the operator $\frac{1}{N}$ :$\text{Tr}[\bar{Z}^2 Z^2]$: projected into the 1/2-BPS sector.

# 3 Lack of strong chaos for $\frac{1}{2}$-BPS states

## 3.1 Review of the fermion description of $\frac{1}{2}$-BPS states

In this section we will study the LMRS criterion in the 1/2-BPS sector of $\mathcal{N} = 4$ SYM. To simplify the analysis, in this section we will consider gauge group U($N$) instead of SU($N$). The differences between the two are negligible in the large $N$ limit.

---

[15]One can generalize the discussion to cases that we sandwich $O_{\text{simp}}$ between two different projection operators $P_{\text{BPS}_1}$, $P_{\text{BPS}_2}$ to compensate for the charge of $O$. In such cases, $\hat{O} = P_{\text{BPS}_1}OP_{\text{BPS}_2}$ would be a rectangular matrix, but we can study its singular values.

[16]The actual one-loop primary also contains terms that involve other letters [63], but since at tree level, they do not Wick contract with $Z, \bar{Z}$ and they cannot contract among themselves, we can safely ignore them when we evaluate the matrix element of $O$ between two words built out of only $Z$ and $\bar{Z}$.

As was mentioned in Section 2, the 1/2-BPS operators in free theory do not develop anomalous dimensions and therefore $P_{\text{BPS}} = P_{\frac{1}{2}\text{-BPS}}$ is independent of the gauge coupling. We can further focus on the subspace that carry a specific amount of $R$-charge, $R_3 = L$. In the word basis, the $R$-charge is given by the eigenvalues of $:\text{Tr}[Z\check{Z}]:$.

In order to diagnose the properties of the BPS subspace via the LMRS prescription, we would like to find a convenient basis of the BPS subspace to compute matrix elements $\langle \text{BPS}_i | O | \text{BPS}_j \rangle$. A natural basis for the BPS subspace with $R$-charge $L$ is given by the set of all possible multi-trace words $\{w_i(Z)\}$, where each $w_i$ contains exactly $L$ $Z$'s. However, in the regime $L \sim N^2$, the multi-trace basis becomes highly overcomplete due to trace relations. Any traces with more than $N$ $Z$'s can be reexpressed as linear combinations of multi-traces with less than $N$ $Z$'s in each trace. The naive number of multi-traces grows as $\sim e^{\mathcal{O}(\sqrt{L})}$, while the actual dimension of the subspace, in the regime where $L \gg N^2$, only grows as $\sim L^N/N! \sim \exp(N \log(L/N))$ which is much smaller. For this reason, the multi-trace basis is inconvenient for the purpose of studying the LMRS problem. One possible way to deal with this problem is to use the so-called Schur polynomial basis, which forms an orthogonal basis for the physical subspace [64]. Here we will instead use a different but equivalent formulation of the subspace, in terms of $N$ one-dimensional free fermions in a harmonic potential [65]. The fermion description removes the redundancy in the multi-trace basis. Another important benefit of the fermion description is that it makes the physical picture clear and helps us identify whether an operator is chaotic or not.

Without going into the derivation of the fermion description (see [66] for a detailed exposition), let us simply state the dictionary and how to utilize the fermion language to do calculation. Let's denote the fermionic creation and annihilation operators at the $n$-th level of the harmonic oscillator to be $c_n^\dagger, c_n, n = 0, 1, 2, \dots$. They satisfy the standard anti-commutation condition

$$\{c_n, c_m^\dagger\} = \delta_{n,m}, \quad \{c_n, c_m\} = \{c_n^\dagger, c_m^\dagger\} = 0, \tag{17}$$

and the Hamiltonian is given by

$$H_f = \sum_{n=0}^{\infty} \left(n + \frac{1}{2}\right) c_n^\dagger c_n. \tag{18}$$

We use the subscript $f$ to highlight that the Hamiltonian is for the auxiliary fermion system and is distinct from the Hamiltonian of the gauge theory. In terms of the fermion language, the CFT vacuum $|\text{vac}\rangle$ is mapped into the Fermi sea state $|\text{FS}\rangle$, where the lowest $N$ levels of the harmonic oscillator potential are occupied,

$$|\text{vac}\rangle \quad \leftrightarrow \quad |\text{FS}\rangle = c_{N-1}^\dagger c_{N-2}^\dagger \dots c_0^\dagger |0\rangle, \tag{19}$$

where $|0\rangle$ is the Fock vacuum of the fermion system. See Figure 1 (a) for an illustration. The Fermi sea state has energy

$$E_{f,\text{FS}} = \frac{1}{2} + \frac{3}{2} + \dots + \frac{2N-1}{2} = \frac{N^2}{2}. \tag{20}$$

A single trace operator $\text{Tr}[Z^k]$ is translated as[17]

$$\text{Tr}[Z^k] \quad \leftrightarrow \quad \sum_{l=0}^{\infty} \sqrt{\frac{(l+k)!}{l!}} c_{l+k}^\dagger c_l, \tag{21}$$

---

[17]In the first quantized language where we denote the creation and annihilation operators of the $i$-th harmonic oscillator as $a_i^\dagger, a_i$, we have $\text{Tr}[Z^k] = (a_1^\dagger)^k + (a_2^\dagger)^k + \dots + (a_N^\dagger)^k$.

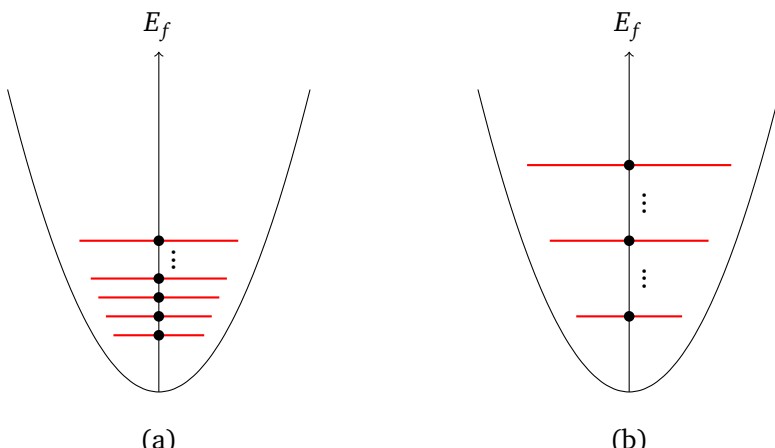

Figure 1: (a) In the fermion picture, the CFT vacuum is mapped to the Fermi sea state, where the lowest $N$ levels are occupied. (b) We are interested in a subspace of states with energy $L \sim N^2$ above the Fermi energy. In a typical state, all the fermions are highly excited and far separated in levels.

which raises the total energy of the fermion system by $k$ units. This translation extends to multi-traces of words involving only $Z$'s, by simply multiplying the corresponding fermionic operators. Therefore, in the fermion language, the subspace of 1/2-BPS states with total $R$ charge $L$ is simply the space of $N$ fermion states with fixed total energy $E_f = E_{f,\text{FS}} + L = N^2/2 + L$. We note that in a typical state with $L \sim N^2$, all the fermions will be highly excited and there doesn't exist a Fermi surface.

## 3.2 Projecting bound state of gravitons

As our main example, we study the projection of an operator $O = \frac{1}{N^2} :\text{Tr}[\bar{Z}^2]\text{Tr}[Z^2]:$ that corresponds to a bound state of two gravitons. To work out its projection into the 1/2-BPS sector, one could compute matrix elements of the form

$$
\begin{aligned}
\langle w_i(\bar{Z})| O |w_j(Z)\rangle &= \frac{1}{N^2} \langle w_i(\bar{Z})| :\text{Tr}[\bar{Z}^2]\text{Tr}[Z^2]: |w_j(Z)\rangle \\
&= \frac{1}{N^2} \langle w_i(\bar{Z})| :\text{Tr}[Z^2]\text{Tr}[\check{Z}^2]: |w_j(Z)\rangle .
\end{aligned}
\tag{22}
$$

Note that for the bra-state, we've applied charge conjugation to the operator $w_i(Z)$. From the first line to the second line, we have used that in the expression, the $\bar{Z}$ in $:\text{Tr}[\bar{Z}^2]\text{Tr}[Z^2]:$ can only contract with the $Z$'s in the ket state, and therefore mathematically it is equivalent to replace it by a derivative operator $\check{Z}$ acting on the words in the ket. We should emphasize that this replacement is only valid when we focus on the projection of $O$ in the BPS subspace. One can now view $(:\text{Tr}[Z^2]\text{Tr}[\check{Z}^2]: |w_j(Z)\rangle)$ on the second line of (22) as a new 1/2-BPS operator, and (22) can be readily evaluated by using standard Wick contractions between $Z$ and $\bar{Z}$.

The projected operator in (22) can also be translated into the fermion language and is given by

$$
\hat{O} = \frac{1}{N^2} \left( \sum_{l=0}^{\infty} \sqrt{\frac{(l+2)!}{l!}} c_{l+2}^\dagger c_l \right) \left( \sum_{l'=0}^{\infty} \sqrt{\frac{(l'+2)!}{l'!}} c_{l'}^\dagger c_{l'+2} \right),
\tag{23}
$$

with further restriction to a sub-block with $E_f = N^2/2 + L$. Physically, what the operator (23) does is to lower the energy of some fermion by two units and then transfer it to another fermion.

### 3.2.1 Finding the spectrum analytically

In this subsection we discuss the solution to the spectrum of $\hat{O}$. We start by noticing that $O$ is factorized into the product of two operators that are quadratic in the fermions

$$\hat{O} = \frac{4}{N^2} \ell_{-1} \ell_1 , \tag{24}$$

where

$$\ell_{-1} \equiv \frac{1}{2} \sum_{l=0}^{\infty} \sqrt{\frac{(l+2)!}{l!}} c_{l+2}^{\dagger} c_l , \qquad \ell_1 \equiv \frac{1}{2} \sum_{l=0}^{\infty} \sqrt{\frac{(l+2)!}{l!}} c_l^{\dagger} c_{l+2} . \tag{25}$$

We have

$$[\ell_1, \ell_{-1}] = H_f , \tag{26}$$

where $H_f$ is given in (18). It is easy to check that we have $[\ell_{\pm 1}, H_f] = \pm 2\ell_{\pm 1}$ and therefore $\{\ell_{-1}, H_f, \ell_1\}$ form a $\mathfrak{sl}(2)$ algebra.[18] Up to a constant factor, the Casimir of $\mathfrak{sl}(2)$ is given by

$$\mathcal{C} = \frac{1}{4} H_f^2 - \frac{1}{2} (\ell_1 \ell_{-1} + \ell_{-1} \ell_1) = \frac{1}{4} H_f^2 - \frac{1}{2} H_f - \ell_{-1} \ell_1 , \tag{27}$$

and therefore we can express $\hat{O}$ in terms of the Casimir and $H_f$ as

$$\hat{O} = \frac{1}{N^2} \left( H_f^2 - 2H_f - 4\mathcal{C} \right) . \tag{28}$$

Since we are interested in the spectrum of $\hat{O}$ in a subspace with fixed $H_f$, our only task is to find out the spectrum of the Casimir $\mathcal{C}$ in a subspace with fixed $E_f$. This can be done as follows. Let's first remove the constraint that $E_f = N^2/2 + L$ and consider the full Hilbert space with $N$ fermions. We can decompose the Hilbert space into irreducible representations of $\mathfrak{sl}(2)$. Since $\ell_1$ lowers the energy by two units while the total energy of the system is lower bounded by $N^2/2$, we know the representations belong to the discrete series. Each representation is formed by a module of states by acting with $\ell_{-1}$ arbitrary times on a lowest weight state that is annihilated by $\ell_1$. In other words, we have

$$\mathcal{H}_{\frac{1}{2}\text{-BPS}} = \oplus_k \mathcal{H}_{\mathcal{C}_k} , \tag{29}$$

where $k$ labels the energy of the lowest weight state in the corresponding representation, i.e. $E_{f,\text{lowest}} = N^2/2 + k$, and the Casimir $\mathcal{C}_k$ is determined by

$$\mathcal{C}_k = \frac{1}{4} E_{f,\text{lowest}}^2 - \frac{1}{2} E_{f,\text{lowest}} = \frac{1}{16} (N^2 + 2k)(N^2 + 2k - 4) . \tag{30}$$

We can get the number of times that a representation with a given $k$ appears by taking the difference of the number of states of total energy $N^2/2 + k$ with that of energy $N^2/2 + k - 2$, since that counts the number of states which are not descendants from lower levels. In the free fermion system, the number of states with total energy $E_f$ is given by the number of different ways to divide an integer $(E_f - N/2)$ into a sum of $N$ non-repeating non-negative integers, which we denote by $p_N(E_f - N/2)$. Note that the non-repeating requirement comes from the Pauli exclusion between the fermions, and we've also subtracted the contribution $N/2$ from the zero point energy. In conclusion, the number of representations with labeled by $k$ is given by

$$n_k = p_N \left( \frac{N(N-1)}{2} + k \right) - p_N \left( \frac{N(N-1)}{2} + (k-2) \right) . \tag{31}$$

---

[18] $H_f$ is related to $\ell_0$ in the usual notation by $H_f = 2\ell_0$.

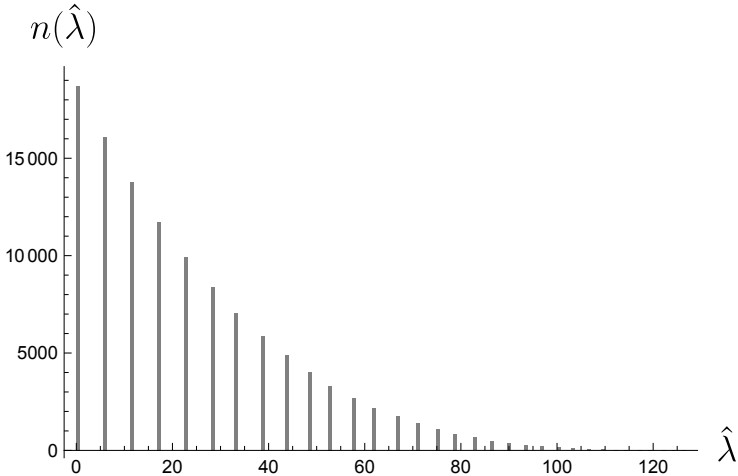

Figure 2: We show a histogram of the eigenvalues of the LMRS operator, in an example with $N = 8$, $L = 64$. We see that the eigenvalues are regularly spaced and has large degeneracies.

Each representation with $k \leq L$ and $(L - k)$ being an even integer contains one and exactly one state with energy $E_f = N^2/2 + L$ that we are interested in. The requirement that $L - k$ being even comes from the fact that $\ell_{-1}$ raises the energy by two instead of one. By plugging (30) into (28) and simplify, we are able to conclude that $\hat{O}$ has eigenvalues

$$\hat{\lambda}_k = \frac{1}{N^2}(L - k)(L + N^2 + k - 2), \tag{32}$$

where $k = 0, 2, ..., L$ or $k = 1, 3, ..., L$ depending on whether $L$ is even/odd, and the degeneracies are given by (31).

From (32) we see that the dependence of the eigenvalues depends on a simple parameter $k$ quadratically and therefore they are quite regularly spaced. We also note that the degeneracies in (31) becomes large when $k$ is large. For $k \sim L \sim N^2$, the degeneracies grow as $L^N$. Figure 2 shows an example of the distribution of eigenvalues. In conclusion, we find that the spectrum of the LMRS operator is in sharp contrast with the random matrix behavior.

As mentioned in Section 2, in principle one should consider the subleading terms such as $\text{Tr}[Z^2 \bar{Z}^2]$ in the simple operator that is suppressed by $1/N^2$. One could check that adding such terms split each peaks in Figure 2 into individual clusters of eigenvalues. This means that even if each cluster is governed by a random matrix, the Thouless energy must be smaller than the spacing between the peaks, which is $1/N^2$ suppressed compared to the overall span of the spectrum. This implies a Thouless time at least of order $N^2$ and therefore corresponds to weak chaos. We will not discuss this problem in detail here since we will study a more general class of operators in Section 3.3, where in fact we would argue for a better estimation of the Thouless time.

### 3.2.2 Moments of the LMRS operator

In this section we discuss an alternative way of showing that the LMRS operator $\hat{O}$ does not have random matrix behavior. The idea is to compute higher moments of $\hat{O}$, i.e. $\text{Tr}[\hat{O}^n]$, from which one can extract information about its coarse-grained eigenvalue density. If $\hat{O}$ were well-described by a random matrix, it should have a square-root edge in the spectrum, which would imply that the higher moments grow at most exponentially in $n$ when $n \gg 1$. We shall see that this is not the case for our $\hat{O}$, where we instead find factorial growth in moments. The

method we are describing here is not as precise as Section 3.2 in that it cannot distinguish the fine-grained details of the spectrum. However, it has the benefit that it can be generalized more easily to other situations.

It is convenient to consider the limit in which $E_f \approx L = qN^2$, with $q \gg 1$. In such a limit, the fermions are highly excited as in Figure 1 (b) and they are far separated in levels. Therefore, one can safely ignore the Pauli exclusion and simply treat them as $N$ classical harmonic oscillators. The operator $\hat{O}$ in (22) can be expressed conveniently in terms of the phase space variables of the $N$ harmonic oscillators, $z_i = (q_i + ip_i)/\sqrt{2}$, $\bar{z}_i = (q_i - ip_i)/\sqrt{2}$, $i = 1, ..., N$

$$\hat{O} = \frac{1}{N^2}(z_1^2 + ... + z_N^2)(\bar{z}_1^2 + ... + \bar{z}_N^2). \tag{33}$$

Instead of considering a microcanonical ensemble where we project $\hat{O}$ into fixed energy $E_f$, it is more convenient to work in a canonical ensemble given by inverse temperature $\beta_f$. The answer for the moments of the LMRS operator should not be sensitive to the ensemble in the classical limit we are considering. We can determine $\beta_f$ in terms of $E_f$ by considering the partition function of $N$ harmonic oscillators

$$Z(\beta_f) \propto \int \mathrm{d}^2z_1 \cdots \mathrm{d}^2z_N \exp\left[-\beta_f \sum_{i=1}^{N} |z_i|^2\right] = \frac{1}{\beta_f^N}, \tag{34}$$

from which we get

$$E_f = -\frac{\partial \log Z}{\partial \beta_f} \sim \frac{N}{\beta_f}. \tag{35}$$

We see that in order for the energy to be order $N^2$, we need to take $\beta_f \sim 1/N$. Demanding $E_f \sim qN^2$ we get

$$\beta_f = \frac{1}{qN}. \tag{36}$$

Before considering general moments of $\hat{O}$, let's first look at its expectation value. We have

$$\begin{aligned}\langle\hat{O}\rangle_{\beta_f} &\approx \frac{1}{Z}\int \mathrm{d}^2z_1 \cdots \mathrm{d}^2z_N \frac{1}{N^2}(z_1^2 + ... + z_N^2)(\bar{z}_1^2 + ... + \bar{z}_N^2)\exp\left[-\beta_f \sum_{i=1}^{N} |z_i|^2\right] \\ &= \frac{1}{N^2}\frac{1}{Z}\frac{2N}{\beta_f^{N+2}} = \frac{1}{N}\frac{2}{\beta_f^2} = 2q^2N.\end{aligned} \tag{37}$$

We see that the expectation value of $\hat{O}$ scales as $N$ in the large $N$ limit. This is because we are consider a high energy window where the expectation value of $|z_i|^2$ becomes large. We can introduce an operator $\widetilde{O} = \hat{O}/N$ such that the expressions below are finite in the large $N$ limit. Now, consider the $n$-th moment of $\widetilde{O}$

$$\begin{aligned}\langle\widetilde{O}^n\rangle_{\beta_f} &\approx \frac{1}{N^{3n}}\frac{1}{Z}\int \mathrm{d}^2z_1 \cdots \mathrm{d}^2z_N (z_1^2 + ... + z_N^2)^n(\bar{z}_1^2 + ... + \bar{z}_N^2)^n\exp\left[-\beta_f \sum_{i=1}^{N} |z_i|^2\right] \\ &= \frac{1}{N^{3n}Z}\frac{1}{\beta_f^{2n+N}}\sum_{k_1,...,k_N}\left(\frac{n!}{k_1!k_2!...k_N!}\right)^2 (2k_1)!(2k_2)!...(2k_N)!,\end{aligned} \tag{38}$$

where $k_1, k_2, ..., k_N$ are non-negative integers subjected to the constraint that they sum up to $n$. In the limit where $n$ is kept large but finite, while $N \to \infty$, the terms that dominate the sum are the ones where $n$ out of the $N$ $k_i$'s are one and the rest are zero.[19] There are $N!/((N-n)!n!)$

---

[19]One could check that, for example, the next term where one of the $k_i$ is two, the rest being zero or one is suppressed by $\mathcal{O}(1/N)$.

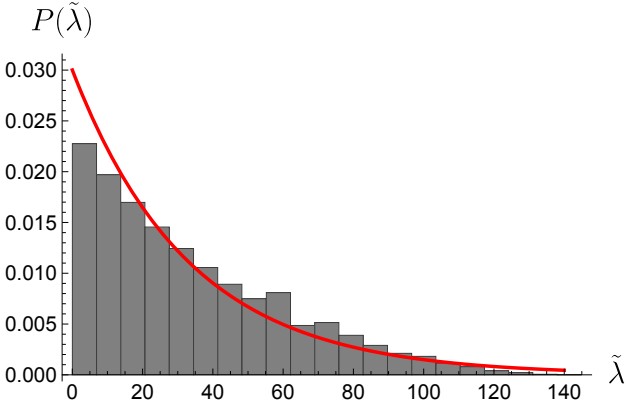

Figure 3: We plot the distribution of the exact eigenvalues $\widetilde{\lambda} = \hat{\lambda}/N$ for $N = 7$, $L = 200$ together with the large $N$ distribution (40) (red) with $q = L/N^2 \approx 4.08$.

such terms. This leads to

$$\langle \widetilde{O}^n \rangle_{\beta_f} \approx \frac{1}{N^{3n}Z} \frac{1}{\beta_f^{2n+N}} \frac{N!}{(N-n)!n!} (n!)^2 \times 2^n \approx n! \left(2q^2\right)^n. \tag{39}$$

Therefore, we find that the spectrum of operator $\widetilde{O}$ approaches an exponential distribution with probability density

$$P(x) = \frac{1}{2q^2} \exp\left(-\frac{x}{2q^2}\right), \tag{40}$$

in the $N \to \infty$ limit. This is in sharp contrast with a spectrum with a squared root edge that one would expect from a random matrix. In Figure 3, we show an example in which we compare the distribution (40) with the exact spectrum found in Section 3.2.

### 3.2.3 Comments on the gravity picture

As we discussed in the introduction, an intuition for the lack of strong chaos for horizonless geometries is that their phase space is embedded in the full phase space of supergravity is a "simple" way. In the 1/2-BPS case, the relevant supergravity solutions are well understood so let us elaborate this intuition further. The geometries are the Lin-Lunin-Maldacena (LLM) solutions [18], which are parametrized by a single function $u(x_1, x_2)$ on a two-dimensional plane that takes value 0 or 1. If we denote 0 by white and 1 by black, the moduli space of the solutions is therefore different ways of coloring the two dimensional plane by droplets of black regions. The Fermi sea state (AdS vacuum) corresponds to a distribution where the unit disk is filled by black while the outside is white. One could then consider excitations with relatively low energy. One can quantize such small fluctuations [67, 68] which leads to the partition function of 1/2-BPS states at $N = \infty$.

On the other hand, in the limit we were considering, where $E_f \sim qN^2$ with $q \gg 1$, the naive droplet picture becomes highly fragmented. In such a limit, a typical highly excited state of the fermion system does not correspond to a particular smooth classical solution.[20] Therefore, one might naively thought that the collective description using $u(x_1, x_2)$ completely breaks down in this limit. However, this is not entirely true as was discussed in [47] (see also related discussion in [70]). It is claimed that by suitably quantizing the full classical moduli space as opposed to small fluctuations, one can reproduce the microscopic description with $N$ free

---

[20]Nonetheless, it was proposed in [69] that typical states in this regime are well-approximated by certain singular geometries, where $u(x_1, x_2)$ is taken to be "grey", i.e. between 0 and 1.

fermions. Therefore, there is a sense that by treating the gravity moduli space exactly, one is able to reproduce the exact microscopic computation.

Classically, the LMRS criterion asks whether a simple supergravity mode, when restricted in the 1/2-BPS subspace, can be expressed in terms of $u(x_1, x_2)$ in a simple way. This problem was analysed in [71], for a wide range of simple operators. It was found that for a large class of supergravity modes, the expressions in terms of $u(x_1, x_2)$ remain simple and take the form as

$$\int dx_1 dx_2 f(x_1, x_2) u(x_1, x_2), \tag{41}$$

where $f(x_1, x_2)$ are some polynomials in $x_1, x_2$. The specific operator $O$ we studied in this section can be thought of as a product of two expressions of the form (41), one with $f = (x_1 + ix_1)^2$ and the other with $f = (x_1 - ix_2)^2$. The fact that supergravity modes translate into simple operators like (41) in the phase space of 1/2-BPS geometries reflects can be viewed as a sign that 1/2-BPS geometries are embedded in the full phase space of supergravity in a simple way.

## 3.3 Weak chaos from projecting stringy operators

We can consider projecting more general gauge theory operators into the 1/2-BPS sector. In the free limit, we can ignore most letters that do not Wick contract with $Z$ and $\bar{Z}$. A large family of operators is given by multi-traces that are built out of an order one number of both $Z$ and $\bar{Z}$. Such operators are generically unprotected and develop large anomalous dimensions at strong coupling. Roughly speaking, they correspond to massive single or multi-string states in the bulk.

In Appendix A we discuss the procedure of finding the form of the projected operator among this general family of operators. Here we focus on a particular simple example $O = \frac{1}{N} :\text{Tr}[\bar{Z}\bar{Z}ZZ]:$. As we derive in Appendix A, the projection of $O$ into the 1/2-BPS sector gives

$$\begin{aligned}
\hat{O} &= \frac{1}{N} P_{\text{BPS}} :\text{Tr}[\bar{Z}^2 Z^2]: P_{\text{BPS}} \\
&= \frac{1}{N} \sum_{l=0}^{\infty} l^2 c_l^\dagger c_l - \frac{1}{N} \sum_{l=0}^{\infty} \sqrt{l+1} c_l^\dagger c_{l+1} \sum_{l'=0}^{\infty} \sqrt{l'+1} c_{l'+1}^\dagger c_{l'} + \text{constant terms},
\end{aligned} \tag{42}$$

where the constant terms are terms that are proportional to identity once we project $\hat{O}$ into the subspace where the total energy of the fermions is fixed to be $E_f = \frac{N^2}{2} + L$. Since the constant terms do not affect the chaotic properties of $\hat{O}$, we will ignore them in the following and use $\hat{O}$ to denote only the first two terms in (42). For notational convenience, we define

$$\hat{O} = \hat{O}_1 - \hat{O}_2, \tag{43}$$

where

$$\hat{O}_1 \equiv \frac{1}{N} \sum_{l=0}^{\infty} l^2 c_l^\dagger c_l, \qquad \hat{O}_2 \equiv \frac{1}{N} \sum_{l=0}^{\infty} \sqrt{l} c_{l-1}^\dagger c_l \sum_{l'=0}^{\infty} \sqrt{l'+1} c_{l'+1}^\dagger c_{l'}. \tag{44}$$

Both $\hat{O}_1$ and $\hat{O}_2$ are "simple" operators. $\hat{O}_1$ measures the second moment of the energy of the $N$ fermions and is diagonal in the natural Fock space basis. $\hat{O}_2$ is nothing but a cousin of the operator that we studied in Section 3.2, where instead of displacing two units of energy from one fermion to another, here only one unit of energy is displaced.

Neither of these operators exhibit random matrix statistics by themselves. However, interestingly, once we consider their combination $\hat{O}$, we in fact find that the spectrum display random matrix statistics in the level spacings. In Figure 4, we show the distribution of the

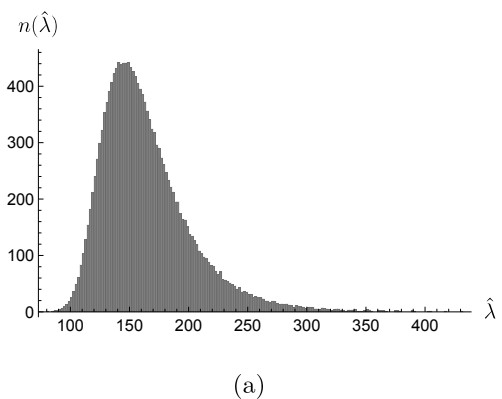
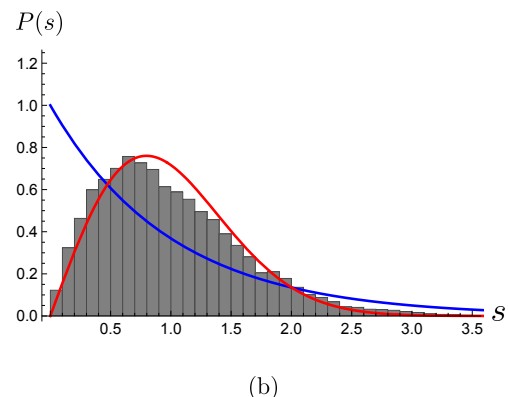

<p style="text-align:center;">(a)             (b)</p>

Figure 4: (a) We show the distribution of the eigenvalues of $\hat{O}$, for the case of $N = 7$, $L = 49$. (b) We show the distribution of the level spacings between unfolded eigenvalues (see Appendix B for the procedure of unfolding), where we removed the 10% lowest and highest eigenvalues.

eigenvalues of $\hat{O}$ and the level spacings, for the case of $N = 7, L = N^2 = 49$. We see from Figure 4 (b) that the level spacings mostly follow that of a Wigner surmise, with small deviations. We have not been able to decide whether the deviations disappear in the large $N$ limit, or it in fact signals systematic deviations from random matrix statistics. In the following, we will assume that in the large $N$ limit, the distribution converges to the Wigner surmise.

Despite the existence of random matrix statistics at the scale of adjacent eigenvalues, we will now explain that the chaos is only *weak*, namely the operator $\hat{O}$ has a long Thouless time, in a precise way we will discuss later. This therefore agrees with our conjecture in Section 1.3. We will later adopt a similar argument in the case of the two charge fuzzball solutions in Section 6, where we also present some further numerical evidence for a long Thouless time. In both cases, the argument relies on the feature that there exists certain ways to order the BPS states such that the action of the LMRS operator becomes local.

In our case here, one way of ordering the 1/2-BPS states that makes $\hat{O}$ local is to order them according to the value of $\hat{O}_1$. In other words, consider the Fock state basis of the fermions $\{|n_1, n_2, ..., n_N\rangle\}$, we can order them according to the size of

$$\langle n_1, ..., n_N | \hat{O}_1 | n_1, ..., n_N \rangle = \frac{1}{N}(n_1^2 + n_2^2 + ... + n_N^2). \tag{45}$$

The fact that we are using an operator $\hat{O}_1$ that appears in $\hat{O}$ is not essential. We could also choose to order the states using general $k$-th moment of the fermion energy and the same argument will apply. The benefit of ordering the states according to (45), or more generally higher moments of the energy, is that the action of $\hat{O}$ becomes *banded*, meaning that it only has non-zero elements in a narrow band near the diagonal.

To see the bandedness here, we only need to argue that $\hat{O}_2$ becomes banded. In a typical state with total energy $L \sim N^2$ above the Fermi sea energy, each fermion has energy that is of order $N$. When we act $\hat{O}_2$ on a Fock state, we change at most the energy of two fermions by (minus) one, and therefore

$$\delta \hat{O}_1 \sim \mathcal{O}(1). \tag{46}$$

We would like to compare this with the overall range of $\hat{O}_1$ in the subspace of states, which can be roughly characterized by its standard deviation. To estimate the standard deviation, we can follow a similar logic as in 3.2.2, that at very high energy, we can ignore the Pauli exclusion and consider a canonical ensemble of $N$ particles, each with energy of order $N$. Therefore, it is easy to see that the mean of $\hat{O}_1$ scales as $N^2$, while the standard deviation $\sigma(\hat{O}_1)$ is suppressed

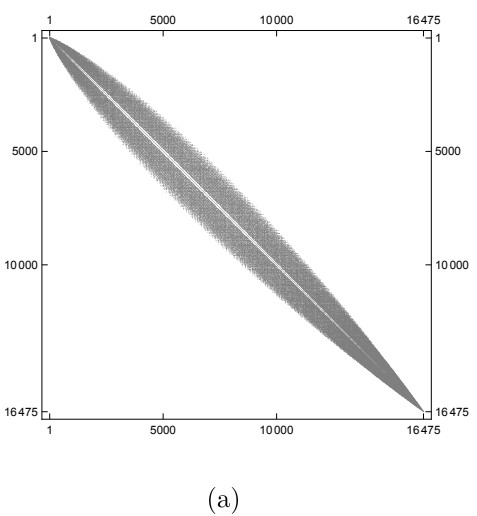

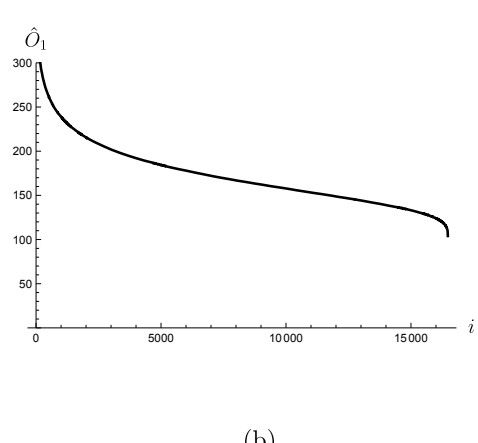

(a)

(b)

Figure 5: (a) Here we plot the matrix $\hat{O}$ for the case of $N = 7$, $L = 49$ where non-zero elements are in black. (b) The states are ordered according to their values of $\hat{O}_1$. Here we show how $\hat{O}_1$ varies from the first to the last state.

compared to the mean by a factor of $\sqrt{N}$ as we have $N$ particles. Therefore, we have

$$\sigma(\hat{O}_1) \sim N^{\frac{3}{2}}\,. \tag{47}$$

See also [72] for similar estimates. In Figure 5 (a), we show a plot of the matrix $\hat{O}$, in the basis that we order $\hat{O}_1$ from higher to lower (see Figure 5 (b)). As we can see from the plot, all the non-zero elements are very close to the diagonal of the matrix.

In [6] the Thouless time of a banded random matrix is discussed (see also Appendix C of [12] for a related numerical study). A one-dimensional banded random matrix $(M_{ij})_{K \times K}$ with width $w$ is a matrix in which all the elements with $|i - j| \leq w/2$ takes random value and all the other elements are zero. Such a matrix can be interpreted as the Hamiltonian of a particle hopping on a one-dimensional lattice, with random hopping strength within range $w$ of a site. It was shown in [6] that the Thouless time of such a matrix corresponds to the time for a particle to diffuse across the system,[21]

$$t_{\text{Thouless}} \sim t_{\text{diffuse}} \sim \left(\frac{K}{w}\right)^2\,. \tag{48}$$

In our case, as we can see from Figure 5 (b), $\hat{O}_1$ varies rather uniformly for the most part of the spectrum. Therefore, we can estimate the ratio between the size and the width of our banded matrix by taking the ratio of (47) and (46). Under the assumption that the Thouless time of the banded matrix we are studying can be bounded below by the Thouless time of a banded random matrix of a similar shape, we reach the conclusion that

$$t_{\text{Thouless}} \gtrsim N^3\,. \tag{49}$$

Therefore the operator $\hat{O}$ can only be weakly chaotic, with a long Thouless time at least scaling as $N^3$. In Appendix A, we explain that the projection of a generic simple operator with order one of letters in the free limit is always banded and therefore we expect the same estimation as in (49).

---

[21]The rigorous theorem in [6] (Theorem 2.2) relies on an assumption that $K \gg w^{7/6}$. The assumption is introduced for technical reasons to control edge effects of the diffusion process in a finite lattice [73]. The intuition of diffusion suggests that the result (48) should apply more broadly.

One might question our assumption that the banded matrix we have is less chaotic than a banded random matrix. After all, one can always transform a matrix, for instance a typical member of a random matrix ensemble, into a banded form. Such a banded matrix will then display stronger chaos than a banded random matrix.[22] However, we note that if we transform a typical random matrix into banded form, the resulting matrix is expected to carry distinctive features which separate them from the matrices we have. For example, consider the banded matrix being simply diagonal. The diagonal form of a random matrix has its eigenvalues as entries, which are highly random numbers satisfying level repulsion, which differs from $\hat{O}$ in (42) whose entries are given by simple expressions. As a more non-trivial example, we can consider the tridiagonal form $M_{\text{tri-diagonal}}$ of the Gaussian orthogonal ensemble. It was shown in [74] that in the ensemble of transformed matrices, the entries in the matrices are drawn independently from specific probability distributions. The diagonal elements of $M_{\text{tri-diagonal}}$ are drawn from normal distribution with the same standard deviation, while the $n$-th off-diagonal element is drawn from the $\chi_n$ distribution, resulting in a distinctive growing pattern along the off-diagonal. We expect the LMRS operators $\hat{O}$ in the 1/2-BPS subspace to not display such features in the cases where they are tri-diagonal. Even though here we only discussed the special cases where the banded matrix is diagonal or tri-diagonal, we expect the distinctions between our matrices and random matrices that were transformed into banded form to exist more generally.

Even though our discussion in this section is in the free limit, our general argument here suggests that one likely finds only weak chaos even at finite $\lambda$ coupling. This is because the Yang-Mills interaction itself only involves an order one number of letters, and can be thought of as a simple operator. Very schematically, at $n$-th loop in perturbation theory, we could have a perturbative correction to the matrix element of the simple operator of the form

$$\sim \frac{\lambda^n}{N^n} \langle w_i(\bar{Z})| \left( :\text{Tr}[Z,\bar{Z}]^2: \right)^n O |w_j(Z)\rangle \,, \tag{50}$$

and if we view the Yang-Mills interaction combined with $O$ as a new operator, our discussion of the bandedness still applies.

# 4 Evidence for lack of strong chaos for $\frac{1}{4}$-BPS states

## 4.1 Review of basic properties

In this section, we will study the LMRS criterion in the 1/4-BPS sector. Unlike the situation in the 1/2-BPS sector discussed in the previous section, here most of the BPS words in the free theory are lifted at one-loop. As a consequence, the projection operator $P_{\frac{1}{4}\text{-BPS}}$ changes discontinuously when going from the free theory to the weakly coupled theory.

The drastic difference between $P_{\text{BPS}} = P_{\frac{1}{4}\text{-BPS}}$ in the free theory and the weakly interacting theory can be easily seen by comparing the ranks of both. We will focus on a subspace with fixed $R$-charges

$$R_3 = L - M \,, \quad R_1 = M \,, \tag{51}$$

and $R_2 = 0$. In other words, we consider multi-trace operators with in total $L$ letters, where $L - M$ of them are $Z$'s, and $M$ are $X$'s. To count the number of BPS words in the free theory, one can simply count the number of multi-traces formed by $Z$ and $X$, modulo the trace relations among them. This problem can be analyzed in the large $N$ limit using an unitary matrix integral [49], and one finds an entropy $\sim N^2$ when $L, M \sim N^2$. Most of the entropy comes

---

[22]One should not confuse the notion of "banded random matrix" with a random matrix that is transformed into banded form.

from the different ways of ordering $Z$ and $X$ inside a trace. On the other hand, the number of the 1/4-BPS operators in the interacting theory can be counted in a similar way while further imposing the constraint that $[Z, X] = 0$ [75]. This means that as far as counting states is concerned, one is free to permute $Z$ and $X$ inside a trace and the ordering no longer matters. As a consequence, the entropy is cut down to $\sim N \log N$ in the interacting theory. Therefore,

$$\mathrm{rank}(P_{\text{BPS, interacting}}) \ll \mathrm{rank}(P_{\text{BPS, free}}). \tag{52}$$

In the following, we use $P_{\text{BPS}}$ to denote that in the interacting theory unless further specified.

Even though the number of 1/4-BPS states can be counted relatively easily (a generating function for the $U(N)$ case is given in (6.4) of [75]), the construction of the explicit wavefunctions $\psi(Z, X)$ of these states at finite $N$ is a more difficult task. We would need their wavefunction (or at least the projection operator) in order to test the LMRS criterion. As we discussed in Section 2, the equation that in principle determines the wavefunction for BPS states at one-loop is

$$\mathcal{D}_2 \psi(Z, X) = -\frac{1}{N} :\mathrm{Tr}\big[[Z, X][\check{Z}, \check{X}]\big]: \psi(Z, X) = 0. \tag{53}$$

There have been various different approaches to solve this problem in the literature. A natural approach used in [76, 77] is to work with the multi-trace basis and express $\psi(Z, X)$ in terms of superposition of them. The drawback of this approach is that the multi-trace basis is highly overcomplete and the results of [76, 77] assume the knowledge of the inner product matrix between multi-traces, which itself is difficult to compute explicitly. Another strategy would be to begin by constructing an orthonormal basis of operators at finite $N$ [78] and then express the action of $\mathcal{D}_2$ in this basis. We will not follow this approach but one can refer to e.g. [79–83] for further discussion along this line.

A yet different approach is to first consider the ungauged model, in which $Z$ and $X$ are $2N^2$ harmonic oscillators. In the ungauged model, a basis of solutions to (53) can be expressed in terms of special coherent states of these harmonic oscillators. One can then integrate them along the gauge orbits to get the BPS states in the gauged theory [84–86]. This approach has the benefit that it makes the physical picture of the BPS states clearer. We will come back to this approach in Section 4.4 and use it to argue that we can only have weak chaos for quarter BPS states.

In the next two subsections, we will mostly follow a different route by studying this problem through diagonalizing $\mathcal{D}_2$ numerically. Our approach is inspired by the results in [56]. A benefit of the numerical approach compared to all the other methods is that one also gets the wavefunction for all the non-BPS states that are lifted at one-loop. Therefore, one can contrast the properties of the BPS states with the non-BPS states within the same framework. In fact, it is exactly the non-BPS states that the reference [56] chose to focus on, where they found that in the regime $L, M \sim N^2$, the non-BPS states display random matrix statistics at one-loop. As part of our analysis, we will reproduce the results in [56] in some overlapping regime, but our main focus will be on the BPS states instead.

## 4.2 Spectrum of the one-loop dilatation operator $\mathcal{D}_2$

We numerically diagonalize the operator $\mathcal{D}_2$, whose eigenvalues (multiplied by $g^2 = \lambda/(8\pi)$) give the one-loop approximation to the anomalous dimensions of near-BPS states. We detail our numerical procedure in Appendix. B while focusing on the results here.[23]

In analyzing features of the spectrum, it is important to desymmetrize the system completely. This applies both to the spectrum of $\mathcal{D}_2$ [56] as well as that of the LMRS operator.

---

[23]The raw data that is used in generating the plots of this section is publicly accessible at [87].

In our current problem, we have two global symmetries. One is an SU(2) symmetry which rotates $X$, which can be thought of as $|\downarrow\rangle$, into $Z$, which can be thought of as $|\uparrow\rangle$. We can write the generators of SU(2) as

$$J_+ = \text{Tr}[Z\check{X}], \quad J_- = \text{Tr}[X\check{Z}], \quad J_z = \frac{1}{2}\left(\text{Tr}[Z\check{Z}] - \text{Tr}[X\check{X}]\right). \tag{54}$$

In a subspace with fixed $L$ and $M$, i.e. fixing the number of $Z$ and $X$'s, all the states have the same $J_z = (L - 2M)/2$, but they can belong to representations with different total spin, labelled by $j$. For example, some of the states are in fact descendants of words with only $Z$'s by applying with $(J_-)^M$ and belong to the same multiplet as the 1/2-BPS states. To desymmetrize, we will focus on the highest weight states, namely states that are annihilated by $J_+$

$$J_+ \psi(Z,X) = 0. \tag{55}$$

These states have the total spin $j$ equal to $J_z = (L - 2M)/2$.

Another symmetry that we should get rid of is a parity transformation $\mathcal{P}$, which in our convention acts on a single trace word as

$$\mathcal{P}\text{Tr}[a_1 a_2 ... a_L] = \text{Tr}[a_L a_{L-1} ... a_1]. \tag{56}$$

$\mathcal{P}$ has eigenvalues $\pm 1$ and we will be focusing on the states with eigenvalues $+1$.

In numerics, we have not been able to fully reach the regime where both the numbers of $Z$ and $X$ become comparable to $N^2$, but we are able to have the total number of letters $L \sim N^2$, while each of them being greater than $N$. In [56], it was shown that the random matrix statistics for the non-BPS states already kick in when $L$ is comparable to $N$, well before $N^2$. This provides confidence that our numerics will be qualitatively similar to the regime where both $L - M$ and $M$ get to order $N^2$. In our numerics, we find it relatively convenient to study the case where the gauge group is SU(4). This is because $N$ is not too large such that our numerical method described in Appendix B is efficient, while it is also not too small such that we still have a relatively large size of the Hilbert space.[24] Therefore, in the following we will focus on the SU(4) theory.

In Figure 6 (a), we show the spectrum of $\mathcal{D}_2$ (after a full desymmetrization), denoted by $\delta E$, in an example with $L = 21, M = 7$. In this particular example, there are in total $n = 671$ states of which 21 are BPS, corresponding to the spike at $\delta E = 0$ in Figure 6. Apart from the BPS states, there are no degeneracies in the spectrum because we have desymmetrized fully. There are some other features of the spectrum that is worth noticing. First, we notice that the overall span of the spectrum is quite large, with the largest eigenvalue being $\sim 55$. In fact, in the regime $L, M \sim N^2$, we have that the overall size of $\mathcal{D}_2$ also scales as $N^2$. Recall from the general discussion in Section 2 that the one-loop approximation is only trustworthy when we are in a regime where the splitting is much smaller than the gap in the free theory, which implies that $\lambda \delta E \ll 1$. What we are seeing here is that in order to trust the entire spectrum including the upper end, one is forced to consider 't Hooft coupling $\lambda$ that scales inversely with $L, M$. What we are saying here is reminiscent of the discussion by Festuccia and Liu [88], where they pointed out that the $\lambda$-perturbation theory at high energy $E \sim N^2$ is badly behaved in the $N \to \infty$ limit while keeping $\lambda$ fixed, no matter how small $\lambda$ is. Of course, for the BPS states and the low lying states in the spectrum, one can presumably trust the results for larger values of $\lambda$.

A second curious feature of the spectrum is the existence of low lying states that are separated from the "continuum". This appears to be a feature that is stable as we vary $L$ and $M$.

---

[24]In the case of SU(2), the trace relations are so powerful so that there aren't many states left, and the spectrum of $\mathcal{D}_2$ in fact becomes integrable [56].

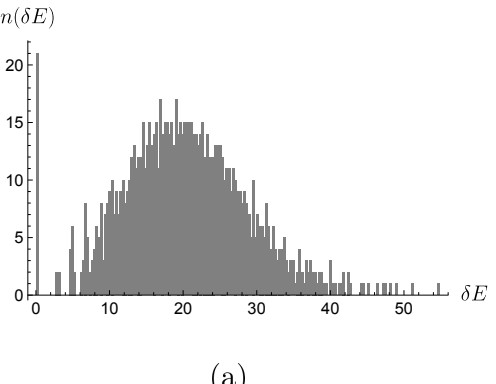
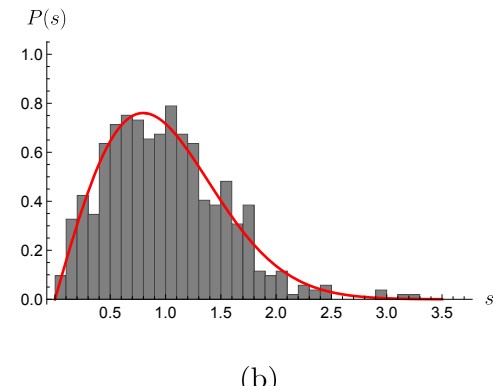

(a)                                                      (b)

Figure 6: (a) We display the eigenvalues of $\mathcal{D}_2$ in the case of SU(4), $L = 21, M = 7$. (b) We plot the probability distribution of the unfolded level spacings among non-BPS states. We didn't include the lowest and highest 10% of the eigenvalues in the plot. For comparison, we also plotted in the red line the Wigner surmise of the GOE ensemble $P_{\text{GOE}}(s) \approx \frac{\pi}{2} s e^{-\frac{\pi}{4} s^2}$.

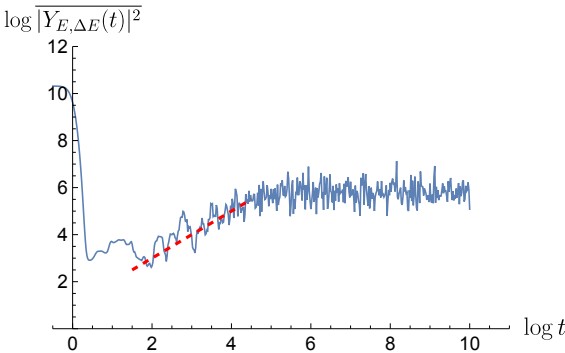

Figure 7: We plot the spectral form factor for the non-BPS states, with parameters $E = 20$ and $\Delta E = 8$. To suppress the noise in the plot, we time-averaged the spectral form factor in a time interval $[t - 1, t + 1]$ around each time $t$. We overlay a red dashed line with unit slope for comparison.

This is in sharp contrast with the feature of the spectrum if the near-BPS states are governed by the super-Schwarzian dynamics [89, 90], where one expects a continuum to form immediately at the gap. We will comment more on this feature in Section 4.4. It would be interesting to contrast this feature with the one-loop anomalous dimensions and the spectral gap in the 1/16-BPS sector [61, 62].[25]

The level statistics of the non-BPS states was studied in [56]. Here we reproduce the features they found as a sanity check. In Figure 6 (b) we plot the distribution of the nearest level spacing $s_i = \delta\varepsilon_{i+1} - \delta\varepsilon_i$, where $\delta\varepsilon_i$ are the eigenvalues, "unfolded" in order to remove the overall density of state. In Appendix B.3 we give a summary of how the unfolding is done. In Figure 6 (b), we also show how the distribution compares with the Wigner Surmise of the Gaussian Orthogonal Ensemble. We see that the distribution of the level spacing fits well with the random matrix statistics [56]. The level spacing only probes the random matrix behavior at the finest scale. One can quantify the "quality" of the random matrix property by studying

---

[25]See also [91, 92] for recent attempts to understand the gap in the boundary theory.

the spectral form factor [3], filtered with a Gaussian window,

$$|Y_{E,\Delta E}(t)|^2 = \left| \sum_i e^{-\frac{(\delta E_i - E)^2}{2(\Delta E)^2}} e^{-i\delta E_i t} \right|^2 . \tag{57}$$

Random matrix universality implies a linear growth in the spectral form factor at intermediate time, $|Y_{E,\Delta E}(t)|^2 \propto t$. For Yang-Mills theory at high energy, we would expect strong chaos for which the Thouless time is expected to be order one (independent of $N$) even at weak 't Hooft coupling. In Figure 7, we plot the spectral form factor associated to the spectrum of the example in Figure 6. In our numerics, since we are only dealing with $N = 4$, we cannot really meaningfully tell whether the Thouless time scales with $N$ or not. Nonetheless, one can still see an extended region in the plot where the spectral form factor grows linearly with time. Note that in GOE ensemble, there are corrections to the linear ramp that become important before reaching the plateau. We have not studied the correction terms using our data carefully.

These results suggest that the non-BPS states are chaotic. We would now like to turn our focus to the BPS states and understand whether they are chaotic under the LMRS criterion.

## 4.3 Numerical results for the spectrum of $\hat{O}$

We numerically studied the LMRS problem for the simple operator (16), which for convenience we reproduce here

$$O = \frac{1}{N^2} \left( :\mathrm{Tr}[Z^2]\mathrm{Tr}[\bar{Z}^2]: + :\mathrm{Tr}[X^2]\mathrm{Tr}[\bar{X}^2]: +2 :\mathrm{Tr}[ZX]\mathrm{Tr}[\bar{Z}\bar{X}]: \right) . \tag{58}$$

When evaluating its matrix elements between states built out of purely $Z$ and $X$ (and their conjugates), we can replace $\bar{Z}$ and $\bar{X}$ by the derivative operators $\check{Z}$ and $\check{X}$, similar to (22). We chose this operator because it commutes with the SU(2) generators as well as parity $\mathcal{P}$. Therefore, we can decouple all the potential effects due to symmetries by studying it in a subspace where the system is fully desymmetrized.

Before discussing the results for the spectrum of LMRS operator $\hat{O}$ in the weakly coupled theory, we can first consider its spectrum in the free theory, namely we use the free theory projection $P_{\text{BPS, free}}$. In Figure 8 (a), we plot the spectrum corresponding to the case with $L = 21, M = 7$ (we focus on the case of SU(4) here and below). Similar to the case in the 1/2-BPS sector in Figure 8, the spectrum has large degeneracies and are regularly spaced, though here it is more complex as all three terms in (58) become non-trivial.

Figure 8 (a) is the free theory result and no dynamics is involved. Once we turn on interaction and use the one-loop $P_{\text{BPS}}$, all the degeneracies of the LMRS operator are broken, as can be seen in the example shown in Figure 8 (b). As the same time, the number of eigenvalues decrease significantly once we go to one-loop. In the regime of our numerical analysis, for a fully-desymmetrized sector with fixed $L$ and $M$, there are only $\sim 20 - 30$ BPS states. This makes it challenging to study level statistics of $\hat{O}$ within a single sector. To deal with this issue, we consider an ensemble of 10 sectors with nearby values of $L, M$, compute level spacings of unfolded eigenvalues of $\hat{O}$ in each of them separately, and collect all the spacings in the ensemble. In Appendix B.3 we list all the sectors in the ensemble and the number of BPS states in each. As a way to make sure that the features we see in the results are not due to statistical fluctuation, we also consider the projection of $O$ into different finite energy windows, each containing exactly the same number of states as the BPS subspace.

The results of this analysis are summarized in Figure 9. In Figure 9 (c), the level statistics of $O$ projected into a high energy window is displayed.[26] We see that the distribution of the

---

[26]In each sector, we always consider a window starting from the $(\lfloor \frac{n}{2} \rfloor + 1)$-th eigenstate to the $(\lfloor \frac{n}{2} \rfloor + n_{\text{BPS}})$-th eigenstate, where $n$ and $n_{\text{BPS}}$ are the total number of states and the number of BPS states, respectively. The states are ordered from higher to lower energy.

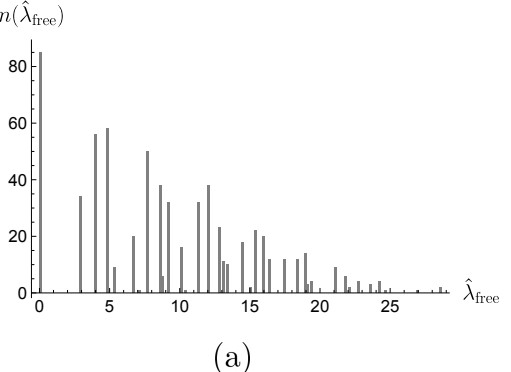
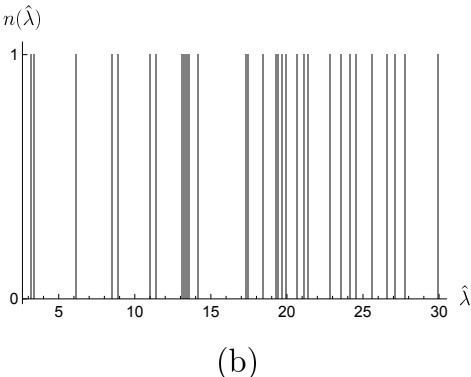

$$\text{(a)} \qquad\qquad\qquad\qquad\qquad \text{(b)}$$

Figure 8: (a) We plot the spectrum of the LMRS operator in the degenerate subspace of quarter-BPS words in the free theory, for the case of $L = 21, M = 7$. The spectrum has large degeneracies, similar to the 1/2-BPS case in Figure 2. (b) The spectrum of the LMRS operator in the one-loop theory, for the case of $L = 22, M = 6$. The degeneracies are completely broken.

level spacings aligns with the Wigner surmise, in agreement with the expectation that the finite energy states are chaotic. This is in sharp contrast with the level statistics of $\hat{O}$, which are displayed in Figure 9 (a). Instead of a Wigner surmise, the distribution of the level spacing aligns more with the Poisson distribution

$$P_{\text{Poisson}}(s) = e^{-s}, \qquad\qquad (59)$$

indicating the lack of repulsion between eigenvalues. There are also many more instances with large spacings, which again differs from the expectation from random matrix statistics. Even though the distribution mostly aligns with Poisson, there is a significant excess of spacings at intermediate values. We expect this to be a remnant of the regular spacings of the spectrum in the free theory, as was seen in Figure 8 (a), which has not been completely removed through the projection.[27] We expect that this feature would be washed away once the system size is further enlarged.

Apart from the level spacing, we can further look at the spectral form factor of the LMRS operator and check that a linear ramp is indeed absent. In Figure 10 (a), we plot the spectral form factor (57) averaged over the sectors we consider. From the plot one cannot see the existence of a linear ramp. On the other hand, the spectral form factor plateaus immediately after the initial dip. As a comparison, in Figure 10 (b) and (c) we plot the spectral form factor of $\hat{O}$ projected into a window of non-BPS states, where one can observe the ramp feature.

Therefore, numerically we see evidence that with the choice of the simple operator in (58), the 1/4-BPS states do not exhibit chaos. Of course, from our discussion of the 1/2-BPS sector, we expect that a more generic choice of operator can lead to weak chaos. We discuss this in Section 4.4. In Figure 9 (b), we further plotted the spectrum of $O$ projected into a small window that is right above the BPS bound. We see that (b) interpolates between the behavior in (a) and (c). In other words, in the one-loop problem, as we gradually lower the energy from the high energy window into the BPS subspace, the chaos gets weakened.

The weakening of chaos is analogous to the discussion of the transition between black holes and strings [93,94] and more specifically the interpolation between two-charge Sen black holes [95] and BPS configurations of heterotic strings [46]. Even though our computation here is only for small $N$ and weak 't Hooft coupling, we can ask what the analogous transition/crossover is in the large $N$ and strong coupling limit, where the gravitational description is accurate.

---

[27]Note that the scale of $s$ in Figure 9 cannot be compared with the scale in Figure 8 due to the unfolding procedure, see Appendix B.3 for discussion.

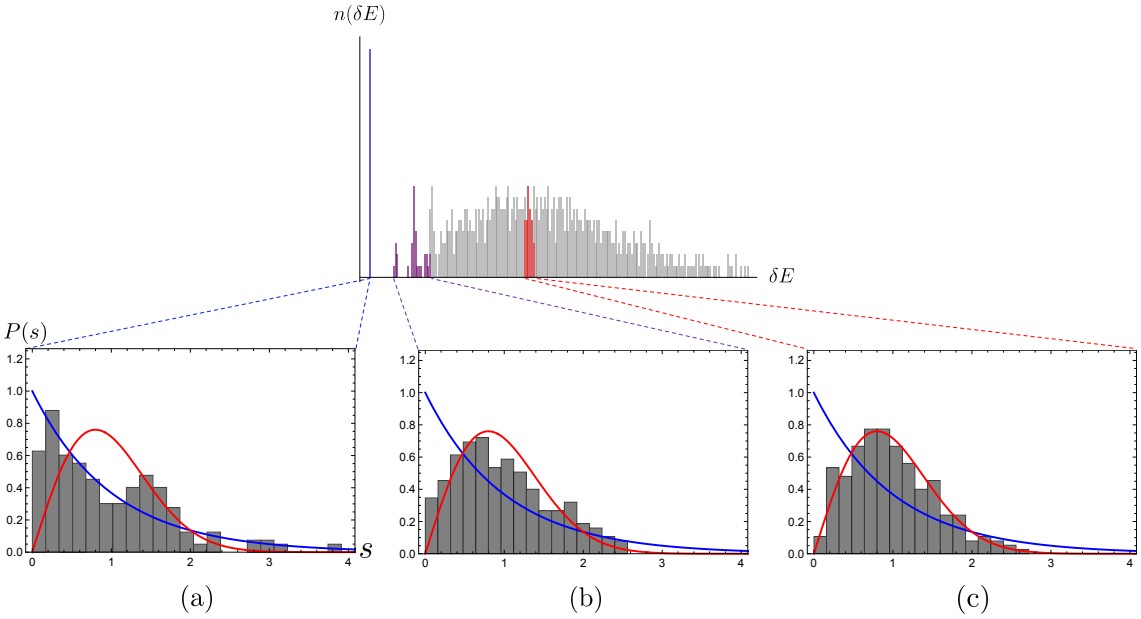

Figure 9: We display the level statistics of *O* projected in three different energy windows: (a) the BPS subspace, i.e. $\hat{O}$, (b) the low lying near-BPS states, (c) highly excited states. In each plot, we also plot the Wigner surmise and Poisson distribution in red/blue dashed lines for comparison. Each plot in fact contains data from several nearby sectors, see main text for discussion.

In AdS$_5$, there exist non-supersymmetric black hole solutions carrying the same *R*-charges described here [96]. These black holes become singular as approaching the extremal limit. On the other hand, there exists smooth and horizonless 1/4-BPS geometries that are similar to the LLM geometry, see [19–23]. Therefore, in gravity, we expect a transition or cross-over between the *R*-charge black holes and the horizonless BPS configurations, where the chaos slows down. In [97], certain instability of these black holes before reaching the BPS bounds was revisited. It was suggested that the black holes become unstable towards emitting a single dual giant graviton that surrounds it far away. There, an entropic argument suggests that it is favorable to emit only one dual giant instead of many of them. However, as one continues to decrease the energy of the black hole towards the BPS bound, we expect the entropic argument to eventually break down, and the black hole core undergoes a sequence of emission of dual giants and eventually disappears. This provides a picture of interpolation between the *R*-charge black holes and LLM-type geometries. It would be interesting to understand the validity of this picture in more detail.[28]

The comments above about the transition in gravity are not specific to the 1/4-BPS geometries and applies to the 1/2-BPS case as well. The main reason that motivated this discussion in the 1/4-BPS case is that on the field theory side we were able to access some of the non-BPS states in the degenerate perturbation theory. Of course, this is a tiny fraction of the actual near-BPS states at strong coupling, since generic near-BPS states will also involve other letters.

## 4.4 Weak chaos from coherent state description for the BPS states

For the special choice of operator *O* in (58), we found numerical evidence for a non-chaotic behavior under the LMRS prescription. Since we already see that in the 1/2-BPS sector, choosing a more general operator can lead to weak chaos, we would expect at least weak chaos for

---

[28]See also [98,99] for earlier discussions in the 1/2-BPS case from other perspectives.

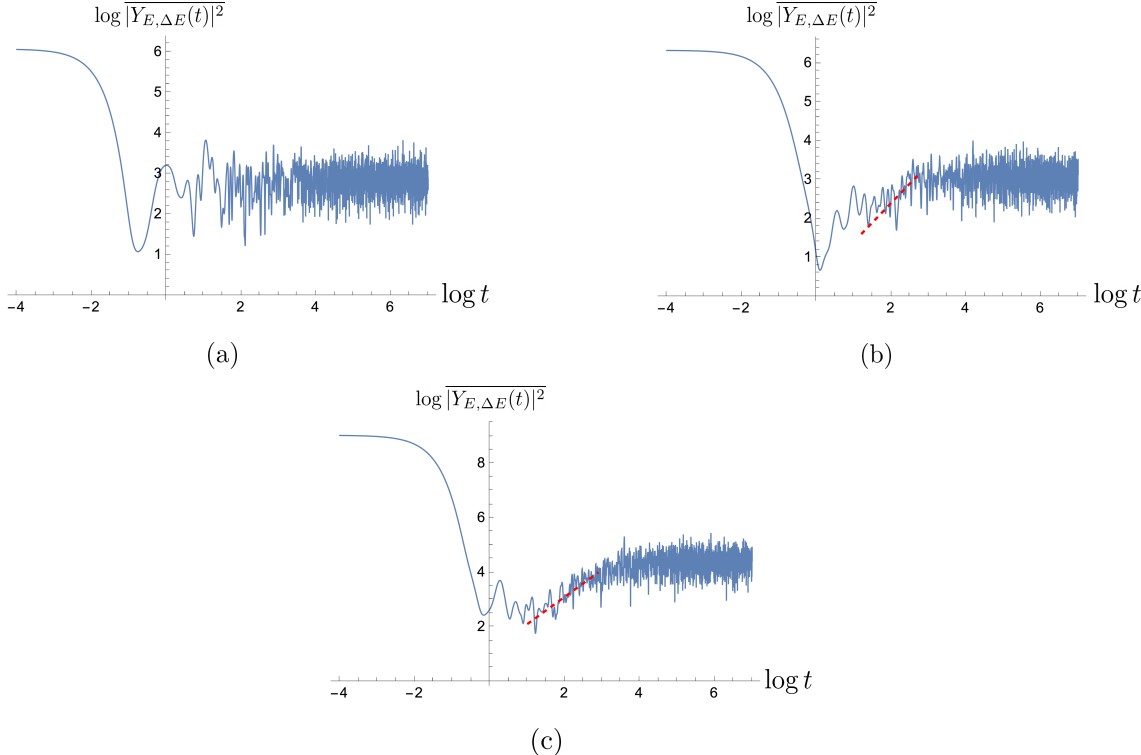

Figure 10: We show the spectral form factor of the projected operator, averaged over our ensemble of sectors. Within each sector, we choose $E$ and $\Delta E$ to be the mean and half of the width of the operator spectrum. In (a), the operator is projected into the BPS subspaces. In (b), the operator is projected into a highly excited window, the same ones depicted in Figure 9 (c). To improve the quality of the plot, in (c), we project the operator into a larger high energy window containing 100 states in each sector. In (b) and (c), we overlay a straight line with unit slope for comparison.

generic choices of simple operator in the 1/4-BPS sector. We would like to understand whether one can establish this analytically. In the 1/2-BPS case discussed in Section 3, we have good analytic control using a free fermion description of the subspace. On the other hand, to the best of our knowledge, a similarly simple description is not established for the 1/4-BPS states.[29] In this section we will review a description for the BPS states in terms of coherent states of commuting matrices [84–86]. Despite that we have not been able to use this description to perform explicit computations, we will try to use it to argue that 1/4-BPS states are only weakly chaotic, following a similar strategy as in Section 3.3.

So far we've been formulating the one-loop problem by starting with the gauge invariant words (superposition of multi-traces) and writing the one-loop dilatation operator $\mathcal{D}_2$ in this basis. To introduce the coherent state description, it is more convenient to start with a slightly different formulation, where we start by working with the basic degrees of freedom in the theory before we gauge $U(N)$ (or $SU(N)$), namely the matrix elements of $Z$ and $X$. More concretely, consider the $U(N)$ case for simplicity, we can replace $Z$ and $X$ by creation operators of $2N^2$ harmonic oscillators

$$Z \leftrightarrow a_Z^\dagger, \quad X \leftrightarrow a_X^\dagger. \tag{60}$$

---

[29]For the purpose of counting the number of 1/4-BPS states, one could use a simple description in terms of $2N$ free harmonic oscillators (in the case of $U(N)$) [75]. However, it is not clear to us how to extend this intuition to address questions about the LMRS operator.

Note that here $a_Z^\dagger, a_X^\dagger$ are $N \times N$ matrices. Similarly, we can replace $\check{Z}, \check{X}$ by the annihilation operators. Therefore, the $R$-charges are simply given by $\mathrm{Tr}[a_Z^\dagger a_Z], \mathrm{Tr}[a_X^\dagger a_X]$, while the one-loop dilatation operator in (12) can be rewritten as

$$\mathcal{D}_2 = -\frac{1}{N} \mathrm{Tr}\big[[a_Z^\dagger, a_X^\dagger][a_Z, a_X]\big]. \tag{61}$$

At one-loop, this takes the same form as the BMN quantum mechanics [100], though they differ at higher loops [101]. Since $\mathcal{D}_2$ is the square of $[a_Z, a_X]$, BPS states in the *ungauged* model are simply determined by the equation

$$[a_Z, a_X]\psi_{\mathrm{BPS,ungauged}} = 0. \tag{62}$$

A basis for the solutions to this equation is provided by the coherent states $|\mathcal{Z}, \mathcal{X}\rangle$, where $\mathcal{Z}, \mathcal{X}$ are each $N \times N$ complex matrices, satisfying the constraint

$$[\mathcal{Z}, \mathcal{X}] = 0. \tag{63}$$

Therefore, a basis for the BPS states in the actual gauge theory can be given by integrating these coherent states along the gauge orbit [84]

$$\int dU \, |U\mathcal{Z}U^\dagger, U\mathcal{X}U^\dagger\rangle, \quad \text{where} \quad [\mathcal{Z}, \mathcal{X}] = 0. \tag{64}$$

The space of 1/4-BPS states is therefore parametrized by two commuting complex matrices, up to common unitary transformations. This is a quantum version of the constraint from the classical superpotential that adjoint fields $X$ and $Z$ should commute.

The coherent state picture makes it clear that there is a certain notion of "locality" in the BPS subspace, similar to what we've seen in Section 3.3 of the 1/2-BPS case. We note that since $[\mathcal{Z}, \mathcal{X}] = 0$, we can use a common unitary transformation to bring both $\mathcal{Z}$ and $\mathcal{X}$ into upper triangular form. In the following, we consider a family of states where both $\mathcal{Z}$ and $\mathcal{X}$ are diagonal, denoted as $\Lambda_Z$ and $\Lambda_X$ [84].[30] With slight abuse of notation, we also use $\vec{\Lambda}_Z$ and $\vec{\Lambda}_X$ to denote the two vectors of diagonal elements. Given two BPS coherent states

$$|\psi\rangle = \int dU \, |U\Lambda_Z U^\dagger, U\Lambda_X U^\dagger\rangle, \quad |\psi'\rangle = \int dU \, |U\Lambda_Z' U^\dagger, U\Lambda_X' U^\dagger\rangle, \tag{65}$$

the inner product between them is given by

$$\langle\psi'|\psi\rangle \propto \int dU \, \langle\Lambda_Z', \Lambda_X'|U\Lambda_Z U^\dagger, U\Lambda_X U^\dagger\rangle. \tag{66}$$

In the case with only one matrix $Z$, the integral over $U$ can be evaluated exactly using the Harish-Chandra-Itzykson-Zuber formula [102,103]. However, in the case with two matrices as we have here, no exact formula is known in general [84]. Despite this, it is clear that the integral over the unitary group only becomes large when $\{\vec{\Lambda}_Z', \vec{\Lambda}_X'\}$ are close to $\{\vec{\Lambda}_Z, \vec{\Lambda}_X\}$ up to simultaneous permutations of the eigenvalues. Assuming that $\{\vec{\Lambda}_Z', \vec{\Lambda}_X'\}$ and $\{\vec{\Lambda}_Z, \vec{\Lambda}_X\}$ have been ordered in such a way that the integral over $U$ is dominated by the contribution near identity, we have

$$|\langle\psi'|\psi\rangle|^2 \sim e^{-|\vec{\Lambda}_Z - \vec{\Lambda}_Z'|^2 - |\vec{\Lambda}_X - \vec{\Lambda}_X'|^2}, \quad \text{when} \quad |\vec{\Lambda}_Z - \vec{\Lambda}_Z'|, |\vec{\Lambda}_X - \vec{\Lambda}_X'| \gg 1. \tag{67}$$

---

[30]We expect from the results of BPS state counting that these diagonal coherent states are already enough to span the entire BPS sector and the off-diagonal elements are redundant, but we have not understood the precise treatment of this problem.

In other words, the overlap becomes exponentially small if $\{\vec{\Lambda}'_Z, \vec{\Lambda}'_X\}$ and $\{\vec{\Lambda}_Z, \vec{\Lambda}_X\}$ differ by an order one amount, and they can be treated as approximately orthogonal. Similarly, we can evaluate the matrix element of a simple operator $O$ in the BPS coherent state basis. For example, for the operator (58) that we've discussed in the previous section, we have

$$
\begin{aligned}
\langle \psi' | O | \psi \rangle &\propto \int \mathrm{d}U \, \langle \Lambda'_Z, \Lambda'_X | O | U\Lambda_Z U^\dagger, U\Lambda_X U^\dagger \rangle \\
&= O(\Lambda'_Z, \Lambda'_X, \Lambda_Z, \Lambda_X) \int \mathrm{d}U \, \langle \Lambda'_Z, \Lambda'_X | U\Lambda_Z U^\dagger, U\Lambda_X U^\dagger \rangle \,,
\end{aligned}
\tag{68}
$$

where

$$
O(\Lambda'_Z, \Lambda'_X, \Lambda_Z, \Lambda_X) \equiv \frac{1}{N^2} \left( \mathrm{Tr}[\bar{\Lambda}'^2_Z]\mathrm{Tr}[\Lambda^2_Z] + \mathrm{Tr}[\bar{\Lambda}'^2_X]\mathrm{Tr}[\Lambda^2_X] + 2\mathrm{Tr}[\bar{\Lambda}'_Z\bar{\Lambda}'_X]\mathrm{Tr}[\Lambda_Z\Lambda_X] \right). \tag{69}
$$

We see that the effect of inserting operator (58) is to simply multiply the overlap (66) by an overall factor. Therefore, a simple operator does not connect coherent states that are far away. The size of the matrix element $\langle \psi' | O | \psi \rangle$ still decays exponentially as in (67) when the two BPS states get further away. This feature does not rely on the particular simple operator we choose. For a generic operator that only contains an order one number of letters, we expect the property of exponentially decaying matrix elements to hold.

What we argued is that a simple operator cannot connect coherent states that are far away. A small improvement of this argument, following the same strategy as in Section 3.3, suggests that a simple operator will be *banded* once projected in the 1/4-BPS subspace. We consider a subspace in which $\mathrm{Tr}[\bar{\Lambda}_Z\Lambda_Z]$ and $\mathrm{Tr}[\bar{\Lambda}_X\Lambda_X]$ are approximately constant and of order $N^2$. Within this subspace, we can form an order the BPS coherent states in a way that both

$$
\frac{1}{N}\mathrm{Tr}[\bar{\Lambda}^2_Z\Lambda^2_Z], \qquad \frac{1}{N}\mathrm{Tr}[\bar{\Lambda}^2_X\Lambda^2_X], \tag{70}
$$

are decreasing. Both of these quantities are of order $N^2$ and each has a standard deviation of order $N^{3/2}$ in the subspace. A simple operator can only connect states whose $\Lambda_Z$ and $\Lambda_X$ differ by an order one amount and can thus only shift (70) by an order one amount. Therefore, the projection of a simple operator becomes a two-dimensional banded operator. Following the results of [6] and similar arguments as in Section 3.3, we expect a Thouless time that is long, at least of order $N^3$.

After establishing that the BPS states are only weakly chaotic as seen by the LMRS criterion, we wish to make some further observations regarding the near-BPS states. In the spectrum as seen in Figure 6 (a), we see a curious feature that the continuum does not start immediately at the gap. On the other hand, the low energy spectrum, to the extent that can be told from the numerics, seems to be more similar to a situation with quasiparticle excitations, see Figure 6, or Figure 11 for a zoomed-up version for a different choice of charges. One can gain a bit more insight by comparing the amount different states are excited by acting $O$ on the BPS subspace. In other words, we could consider the quantity

$$
\frac{1}{Z}\mathrm{Tr}\left[ P_{\delta E_i} O P_{\mathrm{BPS}} O \right], \tag{71}
$$

where $P_{\delta E_i} = |\delta E_i\rangle \langle \delta E_i|$ is the projection operator into the $i$-th eigenstate. We normalize this quantity through dividing it by $Z \equiv \mathrm{Tr}[P_{\delta E>0} O P_{\mathrm{BPS}} O]$ such that summing over $i$ in (71) would give unity. In a case with a continuum right at the gap, one would expect this quantity to be roughly of the same order for the states near the gap. Instead, in our numerics, we find

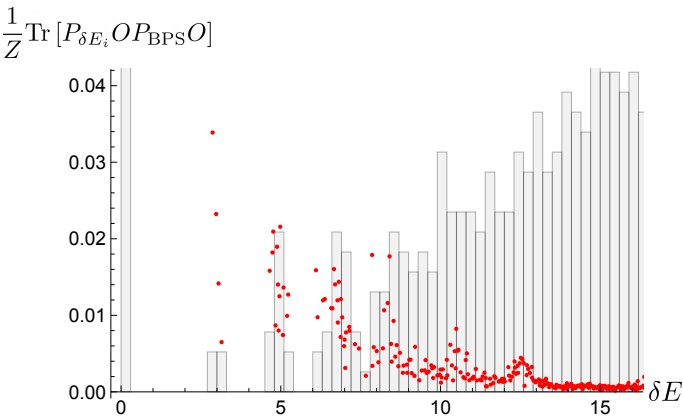

Figure 11: In red, we show the dependence of the quantity (71) on the energy for the case of $L = 22, M = 7$. We also plot the distribution of anomalous dimensions in the background.

the quantity (71) to be concentrated mostly on the low lying states that are away from the continuum, see Figure 11 for an example.[31]

In [84, 85], a set of low energy excitations are proposed by using the coherent state basis. The bulk analogy is that the low energy excitations correspond to strings stretching between D-branes. We suspect that these "open-strings" might provide a physical picture for the low lying states as seen in Figure 11. An illustration of the physical picture for BPS and excited states can be seen in Figure 12.

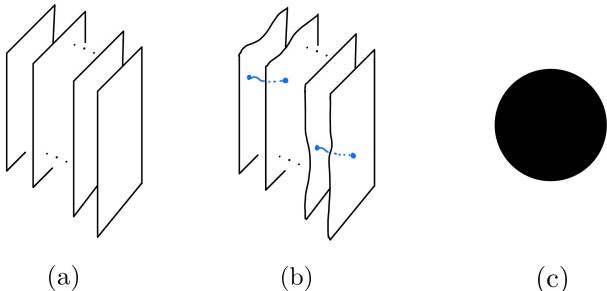

Figure 12: We offer a cartoonish depiction for (a) the BPS coherent states (collections of BPS branes), (b) the near-BPS states (open strings excitations between branes), and (c) the states far above the BPS bound (black hole states).

# 5  Fortuity and the invasion of chaos for $\frac{1}{16}$-BPS states

In the previous sections, we have studied the LMRS problem for the 1/2 and 1/4-BPS sectors, and we found lack of strong chaos for the BPS states in both cases. Both of these sectors are described by horizonless configurations and have small entropy. On the other hand, for the 1/16-BPS sector, developments in studying the superconformal index [75, 104] has uncovered an $\mathcal{O}(N^2)$ entropy in the regime where various charges scale as $N^2$ [57–59]. The entropy from

---

[31]One might be tempted to simply associate the decay to the fact that the simple operator cannot raise the energy too much. However, we should emphasize that the energy being plotted in Figure 11 is the anomalous part of the energy and the bare part of the energy carried by $O$ has already been taken care off. For this reason, the concentration at low lying states appears to be a genuinely interesting feature.

the index analysis agrees with the Bekenstein-Hawking entropy of supersymmetric AdS$_5$ black holes [30–33] that carry the same charges. Furthermore, progress has been made in matching the structure of the large $N$ saddle points for the superconformal index to the sum over bulk geometries, see [105, 106] and references therein. It was further proposed in [35] that the effective dynamics of the near-extremal limit of these black holes are governed by the $\mathcal{N} = 2$ JT supergravity, and therefore governed by a random matrix ensemble [44]. Particularly, the original analysis of LMRS [36, 37], which we reviewed in the introduction, applies precisely to this context. We view the collection of these results as strong evidence from the gravity side that the 1/16-BPS states should be strongly chaotic based on the LMRS criterion.

It is therefore natural to ask, on the gauge theory side, what is the main ingredient that resulted in the different behavior in the 1/16-BPS sector. In this section, we discuss an "invasion" mechanism through which the 1/16-BPS sector can behave qualitatively differently compared to other sectors. The argument is indirect and more qualitative compared to the previous sections, but we believe it could serve as an useful clue towards a better understanding of this problem.

## 5.1 Review of the fortuity phenomenon

Our observation is motivated by recent developments in the finite $N$ $Q$-cohomology of the 1/16-BPS operators [60, 61, 107–110] and particularly the results in [47, 62]. We refer the readers to these references for a more detailed and precise discussion, here we just sketch the main idea. In studying the $Q$-cohomologies, one considers the supercharge $Q$ of the 1/16-BPS sector and searches for all possible solutions to the equation

$$QO = 0, \tag{72}$$

up to the equivalence relation where two solutions $O_1$ and $O_2$ are equivalent if they differ by an $Q$-exact term, i.e. $O_1 - O_2 = QO'$. Crudely speaking, one could view the study of $Q$-cohomology as the midway point between studying the index and finding the precise BPS operators, in that one tries to look for the operator but does not try to simultaneously solve $QO = 0$ and $Q^\dagger O = 0$ (or equivalently $\mathcal{D}_2 O = 0$ at one-loop). By standard argument, the $Q$-cohomologies are in one-to-one correspondence to the actual BPS operators, i.e. the representative in the cohomology differs from the actual BPS operator by an $Q$-exact term.[32]

The recent development, nicely synthesized in [47], is the realization that there are two different types of solutions to (72). This is clearest formulated in the multi-trace basis where the $Q$-action is simple. The precise form of $Q$ will not be important in understanding the main idea, but what's important is that in the multi-trace basis, $N$ does not appear in equation (72) explicitly. Therefore, one can choose to first study (72) in the $N \to \infty$ limit, where the multi-traces are linearly independent. The solutions, up to equivalence relations, are called the "monotone" operators since $O_\text{monotone}$ when viewed as an operator in the finite $N$ theory still satisfies (72)

$$QO_\text{monotone} = 0, \quad \forall N. \tag{73}$$

Due to trace relations at finite $N$, the space of monotone operators will start to shrink as $N$ decreases, but the important feature is that they only disappear but not emerge as $N$ gets smaller. In the half and quarter-BPS sectors that we discussed in Section 3 and 4, all the BPS operators are of the monotone type. As a consequence, one can extrapolate these operators from small $N$ to the large $N$ limit in a precise sense specified in [47], where at $N = \infty$ they

---

[32]It would be desirable if one could formulate a notion of chaos at the level of $Q$-cohomology instead of at the level of actual wavefunctions. However, one should be cautious that the space of $Q$-exact operators is vast and two equivalent operators at the level of cohomology are not necessarily close in other ways. We thank Ying-Hsuan Lin for discussions related to this point.

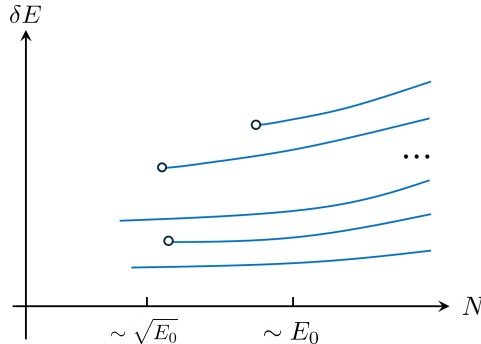

Figure 13: We sketch schematically how the one-loop anomalous dimensions vary with $N$. States start to exit the physical Hilbert space when $N \lesssim E_0$, as represented by the white circles in the sketch.

become light operators that correspond to multi-graviton states in the bulk. This motivates [47] to conjecture that the monotone operators are dual to smooth horizonless geometries in the bulk, formed by a large collection of gravitons or other light BPS particles.

On the other hand, in the 1/16-BPS sector, the number of monotone operators is not enough to match the prediction from the index [60]. In fact, it is now understood that in the regime where the charge of the operator scale as $N^2$, almost none of the 1/16-BPS operators belong to this type, but are instead "fortuitous". The fortuitous operators $O_{\text{fortuitous}}$ are not solutions to (72) at infinite $N$, but only becomes solutions when $N$ is small enough. The way this is achieved is to have

$$QO_{\text{fortuitous}} = \text{trace relation at } N \leq N_*, \tag{74}$$

meaning that the right-hand side would be non-zero when $N > N_*$, but becomes zero for all $N \leq N_*$ due to new trace relations at $N = N_*$. We note that a trace relation at $N_*$ remains a relation when $N < N_*$.

At least at the level of state counting, fortuitous operators are the ones that account for typical black hole microstates. Therefore, if our conjecture in Section 1.3 is true, one would like to have an argument for why fortuitous operators are chaotic. In the following, we provide a perspective for seeing why this might be the case.

## 5.2 Following $N$ and the invasion of chaos

A nice picture for understanding the fortuitous operators was provided in [62], where the authors discussed a way of interpolating between theories with different $N$ by viewing $N$ as a continuous parameter instead of discrete integers. Here we briefly review their idea. To be concrete, we can consider the one-loop problem for a degenerate subspace with classical dimension $E_0$, a basis of which at infinite $N$ is formed by the multi-traces $w_i$, $i = 1, 2, ..., n_w$. When $N$ is large compared to $E_0$, all the multi-traces are independent and one can determine unambiguously $\mathcal{D}_2$ as a matrix in this basis. An important point is that $\mathcal{D}_2$ is analytic in $1/N$ and can be continued away from integer $N$. By diagonalizing $\mathcal{D}_2$ one gets a set of anomalous dimensions $\{\delta E_i(N) | i = 1, ..., n_w\}$, and corresponding wavefunctions $\{\psi_i(N) | i = 1, ..., n_w\}$ that are analytic in $N$ and interpolates between the actual physical answers at different $N$. Generally we do not expect level crossings and the eigenvalues should vary smoothly with $N$ without crossing each other, see Figure 13 for a schematic illustration.

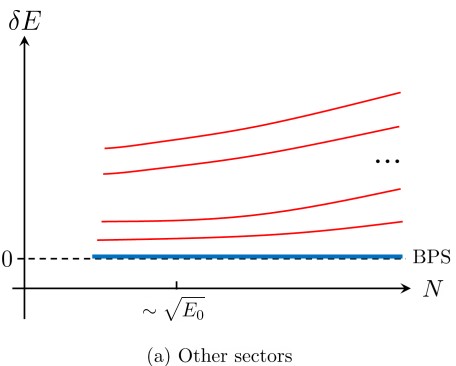

(a) Other sectors

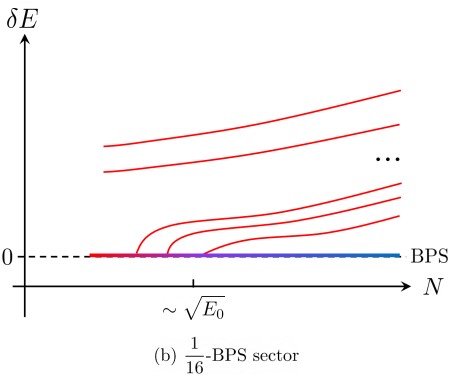

(b) $\frac{1}{16}$-BPS sector

Figure 14: (a) In sectors with more supersymmetries, the non-BPS states and the BPS states stay separated as one follows $N$. (b) In the 1/16-BPS sector, the non-BPS states "invade" the BPS subspace as one decreases $N$. We use red/blue to represent strong/weak chaotic behavior.

We can follow these states and their energies perfectly as described when $N \gtrsim E_0$. However, when $N \lesssim E_0$, the trace relations between the multi-traces start to kick in and not every state continues to be physical at smaller $N$. In order to get rid of the unphysical states, one would have to also keep track of the inner product matrix between the multi-traces

$$W_{ij} = \langle w_i | w_j \rangle , \quad i, j = 1, ..., n_w . \tag{75}$$

The matrix elements of this matrix are polynomials of $N$ and therefore it can also be continued away from integer $N$. At integer $N$, the states that are in the null space of $W$ are no longer physical and we can choose to stop following them. At $N \sim \sqrt{E_0}$ (which simply means $E_0 \sim \mathcal{O}(N^2)$), trace relations are prevalent and most of the states exit the physical Hilbert space. If one follows them further, they may have negative norm and their energy may become complex, see [62]. We will instead focus on those states that we can follow all the way down to small $N$. The idea of following $N$ provides a way to interpolate smoothly between the string-like integrable dynamics at large $N$ and the black hole-like chaotic dynamics at small $N$.

Combining these with the fortuity phenomenon reviewed in Section 5.1, one finds a qualitative difference between the 1/16-BPS sector and other sectors with more supersymmetry. We sketch the main idea in Figure 14. For the sectors other than the 1/16-BPS sector, for example the 1/2 and 1/4-BPS sectors studied in Section 3 and 4, since all the BPS operators are of the monotone type, one can always follow $N$ and trace them back to $N = \infty$, without leaving the BPS subspace. Therefore, as we follow $N$, the non-BPS states and the BPS states are always divided, see Figure 14 (a). On the other hand, in the 1/16-BPS sector, as one decreases $N$, the non-BPS states at the bottom of the spectrum can join the BPS subspace due to the fortuity phenomenon. In [62], an explicit example of this joining effect is discussed. In the regime $E_0 \sim N^2$, most of the BPS subspace in fact "comes from" non-BPS subspace.

If we assume that the non-BPS states are governed by the random matrix statistics, then the fortuity phenomenon provides an avenue for this non-BPS chaos to "invade" the BPS subspace and makes it also chaotic. We expect this chaos to be *strong* as the chaos for generic non-BPS states. For example, in Figure 9 (c) we saw that the projection of a simple operator into a finite energy band is chaotic and we are saying that such finite energy bands can migrate into the BPS subspace in the 1/16-BPS sector. Here we are using the fact that the wavefunctions of the states and OPE coefficients (matrix elements) vary continuously as we vary $N$, therefore their properties should not change suddenly as they migrate in to the BPS subspace. We are also assuming that the chaos does not weaken or slow down as we approach the bottom of

the continuum,[33] such that the non-BPS states still remain their chaotic properties before they migrate across the gap and join the BPS subspace. This is a non-trivial assumption since as we've seen in Section 4.3, this is in fact *not* true in the 1/4-BPS sector and there chaos slows down. We do not expect this will happen in the 1/16-BPS sector based on the Schwarzian description of the near-extremal limit [35]. Without the fortuity phenomenon, the BPS subspace is always separated from the non-BPS states by a gap, leaving room for the possibility that the BPS states are less chaotic compared to generic non-BPS states.

Some evidence for the picture we are proposing here is provided in [62], where the authors used a quantity called information entropy to quantify the chaotic property of a state (see also [56]). Given a basis of states $|w_1\rangle, |w_2\rangle, ..., |w_K\rangle$ that are orthonormal, one can expand a state $|\psi\rangle$ as

$$|\psi\rangle = \sum_{i=1}^{K} c_i |w_i\rangle, \tag{76}$$

and the information entropy is defined as

$$S = -\sum_{i=1}^{K} |c_i|^2 \log |c_i|^2. \tag{77}$$

We will comment further on this quantity and how it compares with the LMRS prescription in Section 7. In [62], the authors studied the information entropy at one-loop, for a total of 69 states that are in the same charge sector as the first known fortuitous operator. From their results, one can verify some key assumptions in our argument. First, the information entropy of the fortuitous operator varies smoothly as it migrates towards and eventually joins the BPS subspace (at $N = 2$). It saturates at a value that is close to the prediction from random matrix theory. Secondly, for relative small values of $N$, one finds that the information entropy is uniform and close to the random matrix value as we go from high energy to low energy. In other words, chaos does not slow down dramatically as we approach the BPS limit, it is always strong.

We have *not* given an argument for whether the monotone operators are only weakly chaotic. It would be interesting to see whether one can extend the coherent state formalism [84–86], reviewed in Section 4.4, to provide a physical picture for all the monotone operators. One might further use it to argue that general monotone operators can only be weakly chaotic under the LMRS criterion.

Ultimately, it would be desirable to develop analytical or numerical techniques to study the dynamics of the BPS and near-BPS states in the 1/16-BPS case directly, without relying on the picture of following $N$. An idea would be to study SYK-like toy models that exhibit both monotone and fortuitous states [111].

# 6 Lack of strong chaos for 2-charge fuzzballs

In this section, we compute the LMRS operator in a subset of the 2-charge fuzzball system. The "simple" operator that we will choose will be a particular twist operator, see [112–115] for some relevant discussion of the twist operator in this context. Since the states that we consider preserve 1/2 of the supersymmetries of the D1-D5 CFT (or equivalently 1/4 of the supersymmetries of Type IIB supergravity) the resulting matrix elements will be protected. Hence we may phrase the computation either in terms of the Hilbert space of the D1-D5 CFT at zero coupling (the free symmetric orbifold), or the gravitational Hilbert space [116] of the 2-charge fuzzballs. We comment briefly on more general choices of operators in Section 6.5.

---

[33]By "weakened", we mean in a dramatic way in which the Thouless time starts to scale with $N$.

## 6.1 Review of D1-D5 description of 2-charge fuzzballs

We consider $N_1$ D1 branes and $N_5$ D5 branes wrapped on $S_1 \times T^4$ in the decoupling limit where the system is described by a (4,4) SCFT. We will mostly consider LMRS matrix elements that are protected in the 2-charge case; hence we can work with the D1-D5 CFT at the free orbifold point which consists of massless bosons and fermions

$$X^{\dot{A}A}(\tau, \sigma), \quad \psi^{\alpha\dot{A}}(\tau + \sigma), \quad \widetilde{\psi}^{\dot{\alpha}\dot{A}}(\tau - \sigma). \tag{78}$$

Here the left/right-moving fermions carry a spinor index $\alpha, \dot{\alpha}$ under the $\mathfrak{su}(2)_L \times \mathfrak{su}(2)_R = \mathfrak{so}(4)$ R-symmetry group (associated with the $S_3$ of $AdS_3 \times S_3$). They also carry an index $\dot{A}$ which transforms under an $\mathfrak{so}(4)_{\text{outer}} = \mathfrak{su}(2) \times \mathfrak{su}(2)$ outer automorphism of the $\mathcal{N} = (4,4)$ algebra. This automorphism is related to the $\mathfrak{so}(4)$ symmetry associated with the tangent space of $T^4$. We consider the CFT on a cylinder with Ramond boundary conditions for the fermions. As we will review in Section 6.2, a basis for the vacuum sector[34] may be described in terms of "strands" of positive integer length $\ell$. The Hilbert space can be thought of as a Fock space, specified by the number of $N_\ell$ strands of various lengths, subject to the constraint that

$$\sum_{\ell,(s)} \ell N_\ell^{(s)} = N = N_1 N_5. \tag{79}$$

Since each strand contributes to the dimension $h = \ell/4$, the total dimension of the operator is $h = N/4$ (independent of $N_\ell^{(s)}$). Furthermore, each strand hosts a Hilbert space $\mathcal{H}_R$ of dimension $2^4 = 16$ corresponding to the Ramond ground states of the 4 complex fermions:

$$\mathcal{H}_R = \text{span}\{|\alpha\dot{\alpha}\rangle, |\alpha\dot{B}\rangle, |\dot{A}\dot{\alpha}\rangle, |\dot{A}\dot{B}\rangle\}. \tag{80}$$

We denote the collective index by $(s)$ in (79). As in (78), these indices transform in the fundamental of the various $\mathfrak{su}(2)$ algebras. As a first pass, we will restrict to the states built out of $|\alpha\dot{\alpha}\rangle = |+\dot{+}\rangle$. It would be more satisfactory to include all possible states; we comment on this in Section 6.4. We can follow these states to strong coupling, where they are dual to 2-charge fuzzball solutions [17, 117, 118]. If we restrict to the states $|\alpha\dot{\alpha}\rangle$ the corresponding geometries are homogeneous on the $T^4$. In the setup we are considering where the fuzzball has no excitations in the torus directions, the classical solution is specified by a profile $x_j = F_j(v)$, where $x_1, \cdots, x_4$ are 4 spatial coordinates transverse to the D5 branes:

$$F_j(v) = \sum_{\ell \neq 0} \frac{a_{\ell,j}}{\sqrt{2\ell}} e^{2\pi i \ell v / L}. \tag{81}$$

We group these as [118, 119]

$$a_\ell^{++} = a_{\ell,1} + i a_{\ell,2}, \quad a_\ell^{+-} = a_{\ell,3} + i a_{\ell,4}, \quad \ell > 0, \tag{82}$$

$$a_{|\ell|}^{--} = -a_{\ell,1} - i a_{\ell,2}, \quad a_{|\ell|}^{-+} = -a_{\ell,3} - i a_{\ell,4}, \quad \ell < 0. \tag{83}$$

In the phase space quantization [116], these modes become quantum harmonic oscillators with the constraint:

$$[a_k^{(s)}, a_\ell^{\dagger(s')}] = \delta^{s,s'} \delta_{k,\ell}, \qquad \sum_{\ell,s} \ell a_\ell^{\dagger(s)} a_\ell^{(s)} = N = N_1 N_5. \tag{84}$$

---

[34]In the NS-NS sector, these states would have different energies, but in the R-R sector, winding a string costs no additional energy. This can be understood simply from the fact that a periodic boson has vacuum energy $E = -\pi/(12\ell)$, an anti-periodic fermion has vacuum energy $-\pi/(24\ell)$, but a periodic fermion has vacuum energy $E = +\pi/(12\ell)$.

For later purposes, we also define

$$\alpha_\ell^{\dagger(s)} = \frac{1}{\sqrt{N_\ell^{(s)}}} a_\ell^{\dagger(s)}, \quad \alpha_\ell^{(s)} = a_\ell^{(s)} \sqrt{N_\ell^{(s)}}. \tag{85}$$

These operators simply shift the occupation numbers in the Fock basis. The holographic dictionary [118, 119] equates the Fourier momentum $\ell$ of the fuzzball profile with the length of the strand $\ell$ in the orbifold CFT description. Furthermore, the number $N_\ell$ of strands of a given length $\ell$ is related to the amplitude of the $\ell$-th Fourier mode in the gravity description.

## 6.2 The projected twist operator

If we take $N$ copies of a field theory, we can consider boundary conditions on a cylinder where the fields satisfy $X_1(\sigma = 2\pi) = X_{g(1)}(\sigma = 0)$, $X_2(\sigma = 2\pi) = X_{g(2)}(\sigma = 0)$, ..., $X_N(\sigma = 2\pi) = X_{g(N)}(\sigma = 0)$. For example, for $N = 4$ we can depict several possible states on the spatial cylinder:

$$\left| \begin{array}{c} 1 \\ 2 \\ 3 \\ 4 \end{array} \right\rangle \quad \longleftrightarrow \quad |\mathbf{1}\rangle \in \mathcal{H}_{\mathrm{aux}}, \tag{86}$$

$$\left| \begin{array}{c} 1 \\ 2 \\ 3 \\ 4 \end{array} \right\rangle \quad \longleftrightarrow \quad |(12)(34)\rangle \in \mathcal{H}_{\mathrm{aux}}, \tag{87}$$

$$\left| \begin{array}{c} 1 \\ 2 \\ 3 \\ 4 \end{array} \right\rangle \quad \longleftrightarrow \quad |(1342)\rangle \in \mathcal{H}_{\mathrm{aux}}. \tag{88}$$

Thus we can label the states of the $N$-fold tensor product theory by a group element $g \in S_N$. (We are assuming that on each strand, the fields are in the Ramond ground state $|++\rangle$.) So we may define an auxiliary Hilbert space $\mathcal{H}_{\mathrm{aux}}$ which is spanned by wavefunctions $\psi(g)$ where $g \in S_N$ with inner product

$$\langle \psi | \chi \rangle = \frac{1}{N!} \sum_{g \in S_N} \psi^*(g) \chi(g). \tag{89}$$

Now we are interested not in the tensor product theory but the *symmetric orbifold* theory, which involves a quotient by $S_N$. This quotient enforces that the state is invariant under arbitrary relabelings of the seed theories (e.g. the numbers 1,2,3,4 in (87) and (88)). When we quotient by $S_N$, we restrict to wavefunction $\psi(g)$ that are invariant under conjugation, e.g., $\psi(g) = \psi(hgh^{-1})$, $\forall h \in S_N$. This gives rise to a "physical" Hilbert space $\mathcal{H}_{\mathrm{phys}}$ of dimension $\dim \mathcal{H}_{\mathrm{phys}} = p(N)$ that is spanned by class functions, with an inner product inherited from (89):

$$\langle \psi | \chi \rangle_{\mathrm{phys}} = \frac{1}{N!} \sum_{[g]} |[g]| \psi^*([g]) \chi([g]). \tag{90}$$

Here $|[g]|$ is the number of group elements in the conjugacy class $[g]$ of $g$. If we like, we may embed $\mathcal{H}_{\mathrm{phys}} \to \mathcal{H}_{\mathrm{aux}}$ in a manner that preserves the inner products:

$$|[g]\rangle_{\mathrm{phys}} = \sum_{g \in [g]} |g\rangle_{\mathrm{aux}}. \tag{91}$$

Now we can consider the left group action on $g : \mathcal{H}_{\text{aux}} \rightarrow \mathcal{H}_{\text{aux}}$ which is defined by $U(g)|h\rangle = |gh\rangle$. For example, the group element $g = (23)$ acts on

$$U(g) \left| \begin{array}{c} 1 \\ 2 \\ 3 \\ 4 \end{array} \right\rangle = \left| \begin{array}{c} 1 \\ 2 \\ 3 \\ 4 \end{array} \right\rangle . \tag{92}$$

By averaging over all such operators in a particular conjugacy class, we can define an operator that acts on $\mathcal{H}_{\text{phys}}$. For example, let $\tau$ be the conjugacy class of all transpositions (2-cycles). We can define:

$$\Sigma_2 |\psi\rangle = \frac{1}{|\tau|} \sum_{t \in \tau} \psi(tg), \tag{93}$$

$$\sum_{t \in \tau} \psi(tg) = \sum_{t \in \tau} \psi(htgh^{-1}) = \sum_{t' \in \tau} \psi(t'hgh^{-1}) \in \mathcal{H}_{\text{phys}}. \tag{94}$$

Here $|\tau| = \binom{N}{2}$ is the number of transpositions. In the second line, we used the fact that $t' = hth^{-1}$ is still a transposition to show that if $|\psi\rangle \in \mathcal{H}_{\text{phys}}$ then $\Sigma_2 |\psi\rangle \in \mathcal{H}_{\text{phys}}$. We can use this formalism to derive some concrete expression (105) and (104) for the twist operator. Consider evaluating $\Sigma_2$ in a basis $|[g]\rangle \in \mathcal{H}_{\text{phys}}$, e.g., $\hat{\Sigma}_2 |[g]\rangle = \sum_{[h]} (\Sigma_2)_{[h],[g]} |[h]\rangle$. The only possible conjugacy classes $[h]$ that can appear are the ones that are obtained by multiplying a transposition, e.g., $[\tau g]$. Such conjugacy classes $[h]$ must have a very similar cycle decomposition as $[g]$.

More specifically, the cycle decomposition of $[h]$ should be obtained from $[g]$ by either joining two of the cycles (say of length $\ell_1, \ell_2$) in $[g]$, or by splitting one of the $[g]$ cycles say of length $\ell_3 = \ell_1 + \ell_2$ into 2 shorter cycles of length $\ell_1, \ell_2$. Indeed we have already seen an example of the joining interaction in (92). More generally, if $i$ and $j$ are part of different cycles, the twist operator stitches the two cycles together

$$(i,j)(i_1, \cdots, i_p = i, \cdots i_{\ell_1})(j_1, \cdots, j_q = j, \cdots, j_{\ell_2}) = (\cdots, i_{p-1}, j_q, j_{q+1}, \cdots, j_{q-1}, i_p, i_{p+1} \cdots), \tag{95}$$

and the splitting interaction is given by

$$(i,j)(i_1, \cdots, i_p = i, \cdots, i_q = j, \cdots, i_{\ell_3}) = (\cdots, i_{p-1}, i_q, i_{q+1}, \cdots)(i_p, i_{p+1}, \cdots, i_{q-1}). \tag{96}$$

This yields a cycle of length $\ell_1 = \ell_3 - \ell_2$ and $\ell_2 = q - p$. To compute the matrix elements of $\Sigma_2$, we must count all possible ways that cycles can join and split. For $\ell_1 \neq \ell_2$ this yields the combinatorial factors:

$$(\Sigma_2)_{[h],[g]} = \frac{|[g]|}{|[h]|} \times \begin{cases} \ell_1 \ell_2 N_{\ell_1} N_{\ell_2}, & [g] \xrightarrow{\tau} [h] \text{ by joining}, \\ \ell_3 N_{\ell_3}, & [g] \xrightarrow{\tau} [h] \text{ by splitting}, \\ 0, & \text{if } g, h \text{ unrelated by } \tau. \end{cases} \tag{97}$$

In the joining interaction, one chooses $\ell_1 \times \ell_2$ locations (corresponding to the choice of $p$ and $q$ in (95)) to join the two cycles together. In the splitting interaction, one must choose 1 of $\ell_3$ possible locations to split the cycle, corresponding to the choice of $p$ in (96). The choice of $q$ is then automatically determined if we demand that the cycle is split into smaller cycles of fixed lengths $\ell_1, \ell_2$. Finally, we have the factor $|[g]|/|[h]|$, which is a conversion factor between matrix elements in the auxiliary Hilbert space and matrix elements in the physical Hilbert space due to (91).

Using the orbit-stabilizer theorem, one can compute the number of permutation elements with a given cycle decomposition:

$$|[g]| = N! \times \prod_i \frac{1}{N_i! \, \ell_i^{N_i}} \,. \tag{98}$$

(To lighten the notation, we will write $N_{\ell_i} = N_i$.) For the joining interaction $(N_1, N_2, N_3) \to (N_1 - 1, N_2 - 1, N_3 + 1)$, giving $|[g]|/|[h]| = \frac{\ell_3(N_3+1)}{\ell_1 \ell_2 N_1 N_2}$, while the splitting interaction $(N_1, N_2, N_3) \to (N_1 + 1, N_2 + 1, N_3 - 1)$ yields $|[g]|/|[h]| = \frac{\ell_1 \ell_2 (N_1+1)(N_2+1)}{\ell_3 N_3}$. Altogether, the matrix elements for $\ell_1 \neq \ell_2$ are given by

$$(\Sigma_2)_{[h],[g]} = \begin{cases} (\ell_1 + \ell_2)(N_{\ell_1 + \ell_2} + 1), & [g] \xrightarrow{\tau} [h] \text{ by joining}, \\ \ell_1 \ell_2 (N_{\ell_1} + 1)(N_{\ell_2} + 1), & [g] \xrightarrow{\tau} [h] \text{ by splitting}. \end{cases} \tag{99}$$

The case $\ell_1 = \ell_2 = \ell_3/2$ requires special treatment.[35] We find:

$$(\Sigma_2)_{[h],[g]} = \frac{|[g]|}{|[h]|} \begin{cases} \binom{N_1}{2}(\ell_1)^2, & [g] \xrightarrow{\tau} [h] \text{ by joining}, \\ \frac{1}{2} \ell_3 N_3, & [g] \xrightarrow{\tau} [h] \text{ by splitting} \end{cases} \tag{100}$$

$$= \frac{1}{2} \begin{cases} \ell_3(N_3 + 1), & [g] \xrightarrow{\tau} [h] \text{ by joining}, \\ \ell_1^2 (N_1 + 2)(N_1 + 1), & [g] \xrightarrow{\tau} [h] \text{ by splitting}. \end{cases} \tag{101}$$

The above discussion applies generally to any symmetric orbifold theory; to specialize to the D1-D5 problem, a minor modification of the definition of the twist operator is required. In particular, we are interested in an operator that can split and join Ramond states. This introduces two subtleties. First, Ramond states carry $\mathfrak{su}(2)$ charge. So an operator that can change the number of strands must be a charged operator. Second, the twist operators should be defined so that the fermionic fields remain periodic after the interaction. Both of these subtleties were addressed by Lunin and Mathur [120], where the operators are defined (see also [121]):

$$\Sigma_2^{\alpha \dot{\alpha}} = S_2^\alpha \bar{S}_2^{\dot{\alpha}} \Sigma_2 \,, \tag{102}$$

where $S^\pm$ and $\bar{S}^\pm$ are spin fields which carry $m = \pm\frac{1}{2}$ and $\bar{m} = \pm\frac{1}{2}$ (and $h = 1/4$) which enforce the periodicity condition, see [120].[36] The chiral operator $\Sigma_2^{++}$ has dimension/charge $h = m = 1/2$ and $\bar{h} = \bar{m} = 1/2$, whereas the anti-chiral operator $\Sigma_2^{--}$ has $h = -m = 1/2$ and $\bar{h} = -\bar{m} = 1/2$. Note that this operator has the correct quantum numbers to merge the Ramond ground states, e.g., $\Sigma_2^{--} |++\rangle_{\ell_1} \otimes |++\rangle_{\ell_2} \sim |++\rangle_{\ell_1 + \ell_2}$ since the single strand has less $\mathfrak{su}(2)$ charge than two separate strands. Similarly, $\Sigma_2^{++} |++\rangle_{\ell_1 + \ell_2} \sim |++\rangle_{\ell_1} \otimes |++\rangle_{\ell_2}$. Note that the marginal operators $\Sigma_{\text{marginal}}$ which deform the theory away from the orbifold point are the superconformal descendants of $\Sigma_2$, i.e. $\Sigma_{\text{marginal}} \sim G_{-\frac{1}{2}} \bar{G}_{-\frac{1}{2}} \Sigma_2$, whose matrix elements between half-BPS states vanish due to non-renormalization theorems. This twist operator is closely related to the so-called "blow-up mode" in supergravity, which is a linear combination of the 10D Ramond-Ramond scalar and the 4-form, see [122, 123].

---

[35]The factor of $\frac{1}{2}$ in (100) originates from the fact that there are really only $\ell_3/2$ distinct choices of $p$, since there is a $q \to p$ symmetry.

[36]For higher twist operators involving cycles of longer lengths, one also needs to modify the definition by the insertion of fractionally moded $\mathfrak{su}(2)$ generators.

Putting all the ingredients together, we can write a compact expression

$$\widehat{\Sigma}_2^{--} = \frac{1}{2} \sum_{\ell_1,\ell_2} C_{\ell_1,\ell_2} (\ell_1 + \ell_2) N_{\ell_1+\ell_2} \alpha^\dagger_{\ell_1+\ell_2} \alpha_{\ell_1} \alpha_{\ell_2}, \tag{103}$$

$$\widehat{\Sigma}_2^{++} = \frac{1}{2} \sum_{\ell_1,\ell_2} C_{\ell_1,\ell_2} \ell_1 \ell_2 N_{\ell_1} \alpha^\dagger_{\ell_1} N_{\ell_2} \alpha^\dagger_{\ell_2} \alpha_{\ell_1+\ell_2}, \tag{104}$$

$$\widehat{\Sigma}_2 = \frac{1}{Z_2} (\widehat{\Sigma}_2^{--} + \widehat{\Sigma}_2^{++}), \quad C_{\ell_1,\ell_2} = \frac{\ell_1 + \ell_2}{2\ell_1 \ell_2}. \tag{105}$$

The creation/annihilation terms[37] in (103) and (104) implement the joining and splitting interaction (85), and the prefactors agree with (99) and (101). The sums over $\ell_1$ and $\ell_2$ are independent, and hence for $\ell_1 \neq \ell_2$ there is an overcounting which is fixed by the factor of $1/2$, whereas for $\ell_1 = \ell_2$ the factor of $\frac{1}{2}$ originates from (101). The operator ordering is chosen to reproduce (101). We also verified that with the inner product (90), the two terms (105) and (104) are Hermitian conjugates of each other; we will study the Hermitian combination $\widehat{\Sigma}_2$. The natural CFT normalization of this operator is to set $Z_2 = 1$, but to compare with random matrix theory, it is natural to choose $Z_2 = Z_2^{\text{RMT}}$ such that $\text{Tr}_{\text{2-charge}}(\widehat{\Sigma}_2)^2 = \dim \mathcal{H}_{\text{phys}} = p(N)$ such that the overall scale of the eigenvalues is order one.

The final ingredient computed in [114] is $C_{\ell_1,\ell_2} = \frac{1}{2}(\frac{1}{\ell_1} + \frac{1}{\ell_2}) \leq 1$, which has the interpretation of the amplitude that the state created after joining two cycles will be a vacuum state (as opposed to an excited state). Without the LMRS projection, the operator $\Sigma_2$ generically produces an excited state, see [113, 114]. Very similar expressions to (105) and (104) already appeared in [112–114, 123]; in deriving them more carefully here we were able to determine the operator ordering and the appropriate treatment of the $\ell_1 = \ell_2$ case.

Actually, the formulas that we use here (with $C_{\ell_1,\ell_2} = 1$) have also appeared already in the math literature on card shuffling [124]. In that context, an ordering of a deck of $N$ cards can be viewed as a permutation group element $g \in S_N$ and one considers probability distributions $P(g)$ on such configurations. These probability distributions can be viewed as elements of $\mathcal{H}_{\text{aux}}$. The operator $\widehat{\Sigma}_2^{\text{card shuffling}}$ which is the same as (105) but with $C_{\ell_1,\ell_2} = 1$ (acting on $\mathcal{H}_{\text{aux}}$) then has the remarkable interpretation as the Markov operator which generates a random walk on the permutation group $S_N$. Equivalently, $\Sigma_2$ encodes the incidence matrix of the Cayley graph on $S_N$, see [125] for a review.

It is interesting to plot the matrix in a basis ordered by the number of cycles $k$. Since a transposition can only change the number of cycles by $\pm 1$, the matrix is banded, see figure 15. Furthermore, we may estimate the size of the band as follows. Let us define $T(k, N)$ to be the number of partitions of $N$ into $k$ parts. In the basis depicted in Figure 15, $T(k, N)$ gives the size of each block. According to [126], one can estimate the maximum value

$$\max_k \frac{T(k, N)}{p(N)} = \frac{c_{\text{w}}}{\sqrt{N}} + o(N^{-1/2}), \quad c_{\text{w}} = e^{-\sqrt{6}/\pi}. \tag{106}$$

This predicts that the maximum width of the matrix in the basis plotted in Figure 15 divided by the dimension of the Hilbert space is $\propto 1/\sqrt{N}$ at large $N$. Furthermore, the typical number of cycles $k$ is of order $\frac{\sqrt{3N}}{\pi\sqrt{2}} \log N$.

The following comment is an aside: the states we are considering here are related to the highly excited BPS states of heterotic strings in a different duality frame. Although the BPS states are described by long strings that carry momentum and winding on extra circles, it has been proposed that a small string scale black hole could play a role in capturing some supersymmetric observables (see [127, 128] for a recent discussion and references therein). It would be interesting to understand the LMRS problem in that picture.

---

[37]we have suppressed the indices $(s) = ++$ on the $\alpha, \alpha^\dagger$. For a more general expression, see (107).

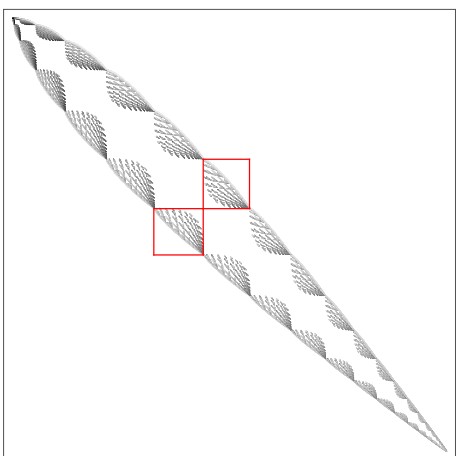

Figure 15: The non-zero matrix elements of $\hat{\Sigma}_2$ in a basis ordered by the number of cycles. The number of cycles $k$ increases from left to right, and also from top to bottom. We display $N = 30$. We highlight two rectangular blocks of dimensions $598 \times 638$ and $638 \times 598$. Here $638 = T(k = 8, N = 30)$ is the dimension of the largest subspace with a fixed number of cycles ($k = 8$ in this example). It determines the "width" of this matrix. In the large $N$ limit, the maximum width $w$ of this matrix satisfies $w/p(N) \sim 1/\sqrt{N}$, see (106).

## 6.3 Eigenvalue statistics of the projected operator

We computed the eigenvalues of $\widehat{\Sigma}_2$ numerically, by explicitly working with matrices of size $p(N) \times p(N)$. The resulting histogram is shown in Figure 16. The eigenvalues for the $N = 44$ operator can be found in the source code on arXiv.[38] The first qualitative feature to notice is that at large $N$ there are many peaks, with relatively large gaps separating the different peaks. We have not found a clear explanation of the pattern of peaks, but it seems to depend sensitively on the details of the coefficients $C = C_{\ell_1, \ell_2}$ in (105). As we review in Appendix C, the spectrum with $C = 1$ can be computed analytically [124], and contains a large number of degeneracies. While we do not understand this in detail, we believe that the structure of the peaks in the spectrum with $C = C_{\ell_1, \ell_2}$ should be related in some way to these degeneracies but we leave this for future work. We also consider a problem with random $C$'s and in that case we did not see multiple peaks in the spectrum.

A second qualitative feature of the spectrum is that there is clear evidence of eigenvalue repulsion for eigenvalues within each "peak" of the cluster, see Figure 17 and also Figure 19 in Appendix C. At $N = 44$ we see a remarkably good fit with the GOE prediction for the nearest-neighbor eigenvalue separations. The observed eigenvalue repulsion is strong evidence of chaotic behavior at very small separations in $\delta\widehat{\lambda}$. However, we would really like to know whether the operator is strongly chaotic or weakly chaotic.

To address this, we would like to estimate the Thouless time for the LMRS twist operator. In principle, it should be possible to estimate the Thouless time numerically from studying the spectral form factor (say in a microcanonical window that isolates just one peak). However, even for $N = 44$, we were not able to see a clear ramp in the microcanonical spectral form factor (even with time averaging), let alone infer the Thouless time as a function of $N$. Nevertheless, although far from a proof, we have collected four pieces of evidence that the Thouless time grows with $N$, most likely $\sim N^\alpha$ with $\alpha \geq 1$ (up to some logarithms).

---

[38]See the file `ev44.txt` in the source code.

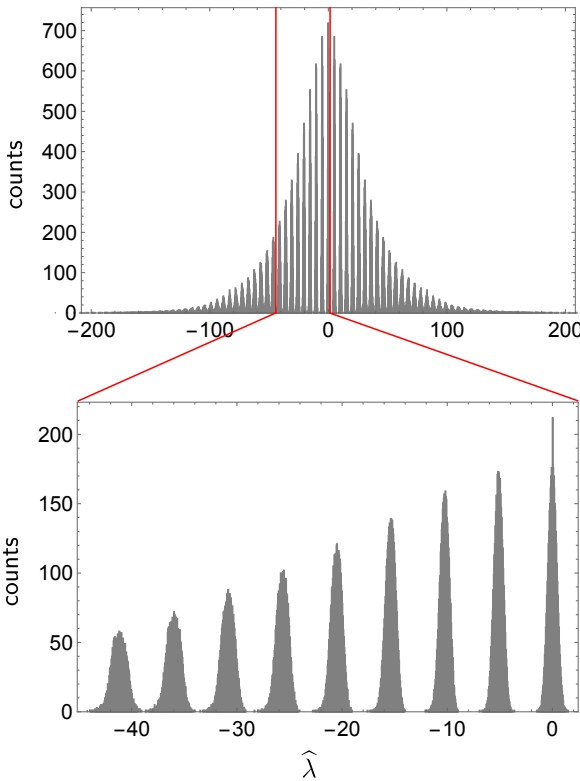

Figure 16: The histogram of the projected $\Sigma_2$ operator for $N = 44$, which corresponds to a Hilbert space dimension of $p(44) = 75175$. In the lower panel, we zoom in on the centermost 9 clusters. The spectrum is symmetric under $\hat{\lambda} \to -\hat{\lambda}$, so this is equivalent to the centermost $8 + 1 + 8 = 17$ clusters. Here $Z_2 = 1$.

1. The first evidence comes from the peaks in the spectrum. Empirically,[39] the separation between peaks[40] seems to scale like $\sim 1/N^p$, see Figure (18). Clearly we do not expect random matrix theory statistics on scales larger than the peak separation scale; this gives a rough Thouless time estimate of $t_{\text{Thouless}} \sim N^{0.57}$. We believe that this bound is not saturated. In particular, a tighter bound would be the "gap" between peaks, defined as the difference between the largest eigenvalue in one cluster and the smallest eigenvalue in the neighboring cluster to the right, but this did not have a clear power law scaling in the range of $N$ that we could access numerically. Furthermore, the spectral form factor of a single peak at $N = 44$ did not seem very close to random matrix theory.

2. A second estimate comes from assuming that the matrix (15) is similar to a random banded matrix with a width determined by (106). Combining this with equation (48) [6]; this gives a Thouless time estimate $t \gtrsim N$. Note that our matrix in the basis plotted in Figure 15 is sparse, so this comparison may not be totally justified.

3. As a simpler problem, we can again return to the $C = 1$ operator. One can consider the operator $(\widehat{\Sigma}_2^{\text{card shuffling}})^\tau$ and ask what for the timescale $\tau_{\text{diffuse}}$ to diffuse over the group. A celebrated result [124] is that this mixing/diffusing timescale is $\sim N \log N$. Relatedly, the diameter of the Cayley graph of $S_N$ (with the set of transpositions as the

---

[39]It would be interesting to analytically derive the scaling of this separation, perhaps by computing moments of the operator $\widehat{\Sigma}_2$.

[40]These results use the RMT normalization of the operator, $\text{Tr}(\widehat{\Sigma}_2)^2 = p(N)$.

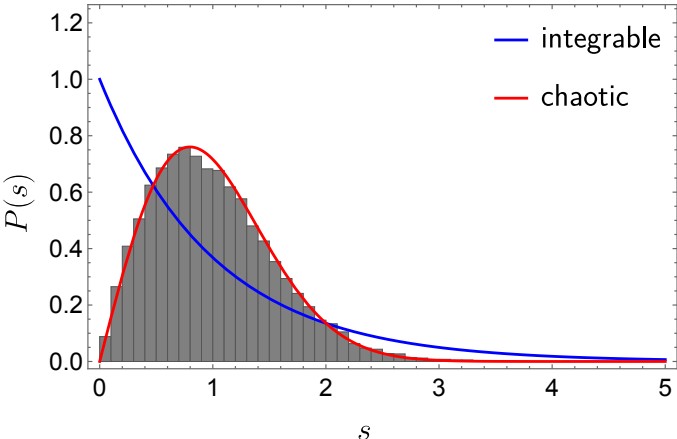

Figure 17: Nearest neighbor eigenvalue separation statistics for the 9 innermost clusters displayed in Figure 16. To make this figure, we separately unfolded each of the 9 clusters, computed the nearest-neighbor eigenvalue separations, and then combined all of them. See Figure 19 for the eigenvalue separations before combining.

generating set) is $N-1$. It seems rather implausible to us that including a factor of $C \leq 1$ (which preserves locality in $k$ space) would overcome the fact that the Cayley graph is very "big" in the large $N$ limit. Based on intuitions from spatially-local lattice models, we expect the Thouless time to be longer than the diffusion timescale; this suggests a Thouless time that grows at least linearly in $N$.

4. For random $C$'s we computed the Thouless time by estimating the spectral form factor for various $N$'s and the results were roughly consistent with a scaling $t_{\text{Thouless}} \sim N$. See Appendix C.3 and Figure 24 for more on this.

## 6.4 Generalizing to the complete LMRS problem

We have computed the matrix elements (and the corresponding eigenvalues) of the LMRS operator in the subspace of 2-charge states that correspond to strands that are all in the $|+\dot{+}\rangle$ states. In the gravity description (81), this corresponds only considering fuzzballs with a profile that rotates in some particular plane. A more complete treatment of the LMRS operator would consider matrix elements between all possible Ramond ground states (all 16 states listed in (80)). This would include fuzzballs with profiles that carry angular momentum in the $S^3$ directions as well as excitations in the $T^4$ or $K^3$ directions.

Although we have not attempted a thorough analysis of this more complete LMRS matrix, it seems implausible that including such states will dramatically alter our conclusions (e.g. lead to strong chaos). Let us comment briefly on including fuzzball states that are excited in the $S_3$ directions. In this case, strands can come in 1 of 4 flavors corresponding to the possible values of $|\alpha\dot{\alpha}\rangle$. Using the Wigner-Eckart theorem, the joining interaction (103) can be generalized to

$$\langle \alpha\dot{\alpha}|_{\ell_1+\ell_2} \Sigma_2^{\beta\dot{\beta}} |\gamma\dot{\gamma}\rangle_{\ell_1} |\delta\dot{\delta}\rangle_{\ell_2} = \delta_{\alpha+\beta+\gamma+\delta}\,\delta_{\dot{\alpha}+\dot{\beta}+\dot{\gamma}+\dot{\delta}} \langle -\dot{-}|_{\ell_1+\ell_2} \Sigma_2^{-\dot{-}} |+\dot{+}\rangle_{\ell_1} |+\dot{+}\rangle_{\ell_2}. \quad (107)$$

(Note here that $\langle -\dot{-}|$ is the Hermitian conjugate of $|+\dot{+}\rangle$.) A similar expression holds for the splitting interaction (104). We see that the more general matrix elements are simply decorated by some selection rules. From (107), we see that we will get banded matrices with similar coefficients to the one that we considered in (105). The main difference is that the blocks that appear in the matrices will have different sizes, since there are now many more states with a given cycle pattern (corresponding to the various different spins that one can fill each strand

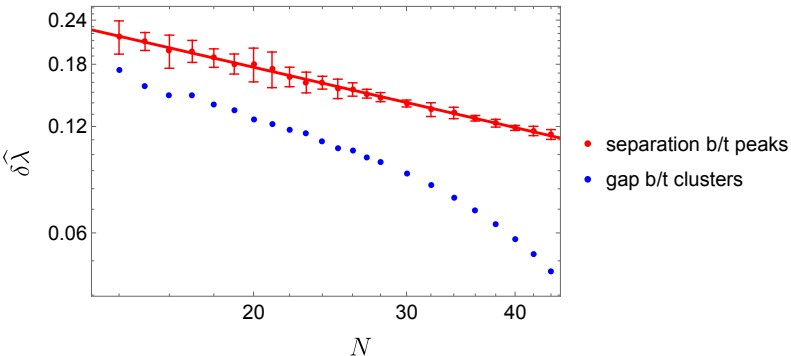

Figure 18: We consider the separation between peaks in the eigenvalue distribution as a function of $N$. For the separation between peaks, we report the mean and standard deviation for all peaks within the window $|\hat{\lambda}| \leq 2$, see Appendix C for more details. This defines an eigenvalue separation scale that upperbounds the scale at which random matrix theory statistics kick in $\delta\hat{\lambda}_{\text{RMT}} \ll \delta\hat{\lambda}_{\text{gap}}$. Hence we have a crude upper bound $t_{\text{Thouless}} \gtrsim 1/\delta\hat{\lambda}_{\text{gap}} \gtrsim 1/\delta\hat{\lambda}_{\text{separation}} \sim N^p$. The best fit curve for the mean separation between clusters $p \approx 0.57$.

with[41]). A preliminary numerical exploration suggests that the width of this banded matrix continues to satisfy (106) except that $c_{\text{w}}$ is a different constant, which depends on the number of colors that each strand can take.

Let us mention a refinement of the LMRS criterion when we have a non-Abelian global symmetry. In general, if we have an operator that transforms in some irrep $R$ of some group, the Wigner-Eckart theorem says that $\langle R_1, m_1, i_1 | \mathcal{O}_m^R | R_2, m_2, i_2 \rangle = \begin{pmatrix} R_1 & R & R_2 \\ m_1 & m & m_2 \end{pmatrix} (\mathcal{O}_{j_1,j_2}^j)_{i_1,i_2}$. Here $m$ runs from $1, \cdots, \dim R$ and $i_1, i_2$ are additional quantum numbers that are unrelated to the symmetry. The LMRS criterion in this situation states that $(\mathcal{O}_{j_1,j_2}^j)_{i_1,i_2}$ viewed as a rectangular matrix in $i_1, i_2$ behaves like a random matrix. In our situation, we can organize all the 2-charge states into irreps $R = (j, \tilde{j})$ of the $\mathfrak{su}(2)_L \times \mathfrak{su}(2)_R$ associated to the angular momentum in $S_3$. In the previous section, the projection into $|++\rangle$ Ramond states is equivalent to demanding that $N_{\text{strands}} = j = m = \tilde{j} = \tilde{m}$. In the more complete problem, we should instead project onto states with fixed $(j, m, \tilde{j}, \tilde{m})$. For example, consider the $(j, m, \tilde{j}, \tilde{m}) = 0$ sector; at large $N$ there will be many states with various strand numbers/cycle decompositions that satisfy this constraint; the splitting and joining interactions will still be local in a basis that is ordered by the number of strands.

In conclusion, we expect the twist operators projected in the full basis of 2-charge states to be weakly chaotic, with a long Thouless time. Supported by numerics and a collection of intuitive arguments, we conjecture that the Thouless time grows at least linearly in $N$. We believe that the most significant qualitative feature that is responsible for preventing strong chaos is the fact that the twist operator only changes the number of cycles by $\pm 1$ and is hence "local"; we will comment on this more in Section 7.2.

## 6.5 Other operators

We can consider testing the LMRS criterion with other operators besides the twist operator. One could consider more complicated twist operators that are based on other conjugacy classes besides the transpositions. After an appropriate dressing by fractionally-moded $\mathfrak{su}(2)$ charge operators and spin fields (see [120]), these are all chiral operators and should be thought of as

---

[41]If a strand of length $\ell$ appears with $N_\ell > 1$, one should consider only states that are invariant under $S_{N_\ell}$.

dual to supergravity modes in the bulk. We expect qualitatively similar features in the LMRS spectrum, at least for conjugacy classes involving relatively small cycle lengths $\ell_i \sim O(1)$ and small cycle numbers $N_i \sim O(1)$. In such cases, we again expect a similar notion of locality to hold and "banded" matrices to arise.

Another possibility is to consider operators which are dual to massive string modes in the bulk, see [129–132]. The LMRS matrix elements would not be protected for such operators, but one could try to study the problem in perturbation theory in the blow-up marginal deformation. An interesting possibility is that an operator that lacks the joining/splitting structure of the twist operators may not be chaotic at zero coupling, but mixes with the twist operators [133] at higher orders in perturbation theory, which would lead to weak chaos. This should be explored further.

# 7 Discussion

In this paper, we have conjectured that the BPS states which are described by horizonless geometries can be distinguished from those that are described by macroscopic black holes based on their chaotic properties. We studied various examples in the case of 4d $\mathcal{N} = 4$ Super-Yang-Mills theory and 2d D1-D5 CFT, using a diagnostic formulated by LMRS – project a simple operator into the BPS subspace and study its eigenvalue statistics. Interestingly, we find that, the horizonless BPS states can exhibit chaos. However, the highly structured features of the horizonless geometries allow us to show that the projected operator is banded, which leads to an estimate of the Thouless time that scales as a power of $N$. Therefore, the chaos is only weak, qualitatively different from the strong chaos one expects from a horizon.

We propose the LMRS criterion as a test for any potential proposals that replace horizons by horizonless objects. The benefit of this test is that it can be studied purely within the BPS context, where one has much greater control.

In the following, we discuss some possible extensions of our discussion as well as some open questions.

## 7.1 Other supersymmetric models

We should emphasize that we have not demonstrated the existence of strong chaos for BPS black hole states from a first-principles calculation on the boundary theory side. Our best argument for strong chaos comes from the super-Schwarzian description from the gravity side. In the 1/16-BPS sector of $\mathcal{N} = 4$ SYM, we've also presented some indirect arguments for strong chaos based on the fortuity idea. We expect the same argument to apply for the fortuitous 1/4-BPS states in the D1-D5 CFT, though more work is required to understand the details. An explicit computation of the projected operators in either of these cases seems challenging but perhaps might be possible numerically.

Another possible approach is to study toy models. For example, one could generalize the supersymmetric SYK model [134] to a system that exhibits fortuitous and monotone states at the same time [111]. Another interesting family [135] of models are the $\mathcal{N} = (2,2)$ Landau-Ginzburg theories with $N$ superfields $\Phi_i$ and superpotential $W = C_{ijk}\Phi_i\Phi_j\Phi_k$.[42] One can compute the chiral ring coefficients (a sphere 3-pt function) using localization. Note that even though localization was used, the resulting answer depends explicitly on the couplings. The resulting finite dimensional integrals are very similar to the toy path integrals introduced in [138] and can be analyzed by introducing $G, \Sigma$. Furthermore, in this model, there is also a

---

[42]See [135, 136] for more generalizations. This can be viewed as a variant of the Muragan-Stanford-Witten model [137].

"stringy exclusion principle" of sorts. In particular the chiral ring is given by

$$\mathcal{R} = C[\Phi_i]/[\partial_i W]. \tag{108}$$

Naively these would just be monomials of superfields $\Phi_i$, but we have to mod out by the relation $\partial_i W \equiv 0$, since this is a descendant according to the equations of motion. Modding out by these relations, one gets a finite number $3^N$ of chiral primaries, with maximum $R$-charge $N/3$. It would be interesting to explore whether there is a super-Schwarzian sector of this model, etc; we hope to return to this in the future.

It would also be interesting to understand the chaotic properties of BPS states in other supersymmetric models, such as the BMN matrix quantum mechanics [100,139] and the ABJM theory [140]. Indeed, the 1/2 BPS sector of the BMN matrix quantum mechanics is also described by Young Tableaux/free fermions [18], and more generally the problem of finding the black hole cohomologies is mathematically very similar to the problem in $\mathcal{N} = 4$ SYM at 1-loop.

## 7.2   Locality in the BPS subspace

When trying to bound the chaos for horizonless geometries, an important concept we used was a certain locality in the BPS subspace. By locality we mean that we can arrange the BPS states in some natural way such that the projection of a simple operator becomes banded.

The fact we are able to order the BPS states in a natural way is intimately connected to the extra structures of horizonless geometries at the level of supergravity. In the 1/2-BPS case, we used higher moments of the fermion energy to order the states. In the droplet picture, these higher moments map to charges that are conserved under the evolution of the droplets, which together generating a $W_\infty$ symmetry group [141]. As we go from non-BPS to the BPS limit, we lose the horizon but gain these simple operators that can be used to distinguish the BPS states.[43]

Although we focused on the 2-charge fuzzballs in Section 6, one can ask whether the known 3-charge fuzzball/horizonless solutions also have a similar notion of "locality." The quantization[44] of some 3-charge fuzzballs was carried out in [147]. Instead of a profile $F_i(v)$, the superstrata are characterized by 2 holomorphic functions of 3 complex variables [147], $G_1(\xi, \chi, \eta) = \sum_{k,m,n} b_{k,m,n} \xi^n \chi^{k-m} \eta^m$, $G_2(\xi, \chi, \eta) = \sum_{k,m,n} c_{k,m,n} \xi^n \chi^{k-m} \eta^m$. Here the Fourier modes of $G_1, G_2$ are again creation/annihilation operators of harmonic oscillators, see [147] for the details. Based on the discussion in [123], we believe that the appropriate notion of locality is related to the occupation numbers associated with these $b$ and $c$ modes. Note that the computation of the LMRS operators in the 3-charge case is more subtle since some of the 1/4 BPS states are lifted, see [115,148,149].

## 7.3   Other diagnostics of quantum chaos

We focused on a particular diagnostic of chaos in a degenerate subspace through projecting simple operators. One could ask whether there are other ways to probe the quantum chaos in the BPS subspace.

---

[43]A toy model that shares some vague similarities is SYK model with a $q = 2$ term plus a $q = 4$ term. In this model, as one increases the strength of the $q = 2$ term, or by simply lowering the energy, there is a transition from chaotic to integrable dynamics [142,143]. In the low energy limit where the $q = 2$ term dominates, one finds that the $G - \Sigma$ action develops an infinite number of soft modes other than the ordinary Schwarzian mode [144,145]. It would be interesting to understand whether there is an emergent symmetries that govern these soft modes and the possible connections to higher spin versions of JT gravity [146].

[44]As an aside, an interesting question is whether the super-Schwarzian mode is part of the quantization. It seems plausible that the 3-charge solutions with a long, nearly AdS$_2$ throat could develop a soft mode that requires a quantum treatment. Since the super-Schwarzian is responsible for strong chaos in the black hole case, it would be interesting to understand this issue for the horizonless geometries.

As we mentioned in Section 5.2, a quantity that one can use is the information entropy of quantum states. However, as defined, it only applies to individual microstates instead of a subspace of states. Unlike the LMRS criterion, the information entropy requires a choice of basis. In gauge theories with adjoint matter, one natural choice of the basis is the multi-traces [56,62], but they are highly redundant at high energy which leads to some ambiguity in defining the information entropy. Nonetheless, it seems plausible that the information entropy could still serve as a useful diagnostic. One might also consider other quantum information notions such as pseudorandomness [150,151].

At finite energies, another signature of chaos is operator growth [152–154]. A simple operator $O$ becomes highly complicated under Heisenberg evolution $e^{iHt}Oe^{-iHt}$. In our discussion, a simple operator becomes complicated not by a real time evolution, but by conjugating with a long Euclidean evolution $P_{\text{BPS}} = e^{-\infty H}$. It would be interesting to understand whether one can generalize the picture of operator growth to this situation and the relation to the Euclidean out-of-time-order correlator.

# Acknowledgments

We thank Ibrahima Bah, Chi-Ming Chang, Persi Diaconis, Matthew Heydeman, Shota Komatsu, Finn Larsen, Ji Hoon Lee, Hai Lin, Ying-Hsuan Lin, Raghu Mahajan, Juan Maldacena, Douglas Stanford, and Zhenbin Yang for discussions.

**Funding information** YC acknowledges support from DOE grant DE-SC0021085. HL is supported by a Bloch Fellowship and by NSF Grant PHY-2310429. SHS is supported in part by NSF Grant PHY-2310429. We would like to acknowledge the KITP program "What is String Theory? Weaving Perspectives Together", supported in part by grant NSF PHY-2309135 to the Kavli Institute for Theoretical Physics (KITP). SHS thanks the Aspen Center for Physics for hospitality, supported by NSF grant PHY-2210452.

# A   Details of projecting more general operators in the $\frac{1}{2}$-BPS sector

In this appendix, we give more details on how to derive the projection of a general operator into the 1/2-BPS sector. Our discussion will be only valid in the free limit, namely we can just use the free theory Wick contraction to evaluate the matrix element of an operator between two 1/2-BPS states. We will consider a family of operators that are only built out of $Z$ and $\bar{Z}$. We do not need to consider other letters such as other scalars $X, Y$ since they do not Wick contract with the ket or the bra operators.

Consider a simple operator $O(Z, \bar{Z})$ that is built out of traces of $Z$ and $\bar{Z}$. Examples that we studied in the main text are $O = \frac{1}{N^2}$ :$\text{Tr}[\bar{Z}^2]\text{Tr}[Z^2]$: and $O = \frac{1}{N}$ :$\text{Tr}[\bar{Z}^2Z^2]$:. More generally, we consider any operators simple if it contains only an order one amount of letters. We can compute the matrix element between two 1/2-BPS operators can be computed by a complex Gaussian matrix integral (see for example [66] for a detailed discussion)

$$\langle w_i(\bar{Z})| O(Z, \bar{Z}) |w_j(Z)\rangle \propto \int d^2Z \, w_i(Z^\dagger)O(Z, Z^\dagger)w_j(Z)e^{-\text{Tr}[Z^\dagger Z]}. \tag{A.1}$$

This is easy to understand - the Gaussian integral simply implements the Wick contractions between $Z$ and $Z^\dagger$. In (A.1), we did not take into account normal ordering of operator $O$ and $Z, Z^\dagger$, can freely contract. To implement normal ordering, one can subtract off the contribution

from various operators coming from contracting $Z, Z^\dagger$ within $O$, for example[45]

$$:\text{Tr}[\bar{Z}^2 Z^2] := \text{Tr}[\bar{Z}^2 Z^2] - 2\text{Tr}[\bar{Z}]\text{Tr}[Z] - 2N\text{Tr}[\bar{Z}Z] + N(N^2+1). \tag{A.2}$$

The integrand in (A.1) is invariant under unitary transformation of $Z$. We can use a unitary transformation to put $Z$ into *upper-triangular* form

$$Z = UTU^\dagger, \tag{A.3}$$

where $T_{ij} = 0$ for $i > j$ and $z_i = T_{ii}$ are the eigenvalues of $Z$. In the following we will use $T_{ij}$ to denote only the off-diagonal elements. One can then perform a change of variable and find [155]

$$\langle w_i(\bar{Z})| O(Z,\bar{Z}) |w_j(Z)\rangle$$
$$\propto \int \prod_{i<j} d^2 T_{ij} \prod_i d^2 z_i\, |\Delta(z)|^2 w_i(z^*) O(z,z^*,T_{ij},T_{ij}^*) w_j(z) e^{-\sum_i |z_i|^2 - \sum_{i<j}|T_{ij}|^2},$$
$$\text{where} \quad \Delta(z) = \prod_{i<j}(z_i - z_j). \tag{A.4}$$

Notice that $w_i$ and $w_j$ only depend on the eigenvalues and not the off-diagonal elements of $T$. This is because they are built out of traces of either only $Z$ or only $Z^\dagger$, so they receive no contribution from the off-diagonal elements. On the other hand, this is *not* true for a general operator built out of both $Z$ and $Z^\dagger$. For example, for $\text{Tr}[Z^{\dagger 2} Z^2]$, what we get is

$$\text{Tr}[Z^{\dagger 2} Z^2] = \text{Tr}[T^{\dagger 2} T^2] = \sum_i |z_i|^4 + \sum_{j>i}|z_i + z_j|^2 |T_{ij}|^2 + \text{terms with only } T_{ij}\text{'s}. \tag{A.5}$$

We do not keep track the terms with only $T_{ij}$'s since after integrating over them in (A.4), they only lead to a constant shift to the operator. On the other hand, the mixing terms between $T_{ij}$'s and $z_i$'s are important. In this example, after we integrate out $T_{ij}$, we get

$$\text{Tr}[Z^{\dagger 2} Z^2] \sim \sum_i |z_i|^4 + \sum_{j>i}|z_i + z_j|^2 = \sum_i |z_i|^4 + \sum_i z_i^* \sum_j z_j + (N-2)\sum_i |z_i|^2, \tag{A.6}$$

where in the second step, we expressed the expression in terms of the "power sums", namely each term is products of the form

$$O_{n,m} = \sum_i (z_i^*)^n z_i^m, \quad n, m \in \mathbb{Z}_{\geq 0}. \tag{A.7}$$

More explicitly, we have

$$\text{Tr}[Z^{\dagger 2} Z^2] \sim O_{2,2} + O_{1,0}O_{0,1} + (N-2)O_{1,1}, \tag{A.8}$$

up to a constant term. This is not special to the operator $\text{Tr}[Z^{\dagger,2}Z^2]$ but rather a general property. The function $O(z,z^*,T_{ij},T_{ij}^*)$ is invariant under arbitrary permutations of indices $i \to \sigma(i)$, $\sigma \in S_N$, and this property remains correct after we integrate out the off-diagonal $T_{ij}$. In other words, after we integrate over $T_{ij}$, the expression lives in the ring of multi-symmetric polynomials of $\{z_1,...,z_N; z_1^*,...,z_N^*\}$, which can be generated by the power sums in (A.7). Now, consider an operator that is a product of several power sums

$$O_{n_1,m_1}...O_{n_k,m_k} = \sum_{i_1,...,i_k} (z_{i_1}^*)^{n_1}...(z_{i_k}^*)^{n_k} z_{i_1}^{m_1}...z_{i_k}^{m_k}, \tag{A.9}$$

---

[45]Here we consider the case of U($N$), same as in Section 3.

in the first quantized language, it can be identified with an operator acting on the Hilbert space of the $N$ harmonic oscillators by replacing $z_i$ by $a_i^\dagger$ and $z_i^*$ by $a_i$,

$$O_{n_1,m_1}...O_{n_k,m_k} \sim \sum_{i_1,...,i_k} a_{i_1}^{n_1}...a_{i_k}^{n_k} a_{i_1}^{\dagger m_1}...a_{i_k}^{\dagger m_k}. \tag{A.10}$$

To understand this map, we notice an equivalence between levels of a one-dimensional harmonic oscillator and the maximal angular momentum wavefunctions of a two dimensional harmonic oscillator. In other words, we have

$$|n\rangle \longleftrightarrow \frac{1}{\sqrt{n!}} z^n e^{-\frac{|z|^2}{2}}, \tag{A.11}$$

and when we act $a^\dagger$ on the state $|n\rangle$, we simply multiply the wavefunction by $z$. We have a similar map for the bra states where we can interchange $a$ and $\bar{z}$. Following this mapping, expression (A.4) with $O$ being (A.9) can be reinterpreted as computing the matrix element of (A.10) in some states of $N$ harmonic oscillators.

Note that since the operator $O(Z, \bar{Z})$ only includes an order one number of letters, in (A.10) we have $k \sim \mathcal{O}(1)$ and also $n_i, m_i \sim \mathcal{O}(1)$. Therefore, such an operator chooses an order one number of harmonic oscillators and change their energies each by an order one amount. One can further write in (A.10) in terms of the second quantized fermionic operators $\{c_l, c_l^\dagger\}$,. Following the same discussion in Section 3, all such operators are "banded". For the case of the operator in (A.8), explicitly, we have

$$\text{Tr}[\bar{Z}^2 Z^2] \sim \sum_i a_i^2 a_i^{\dagger 2} + \sum_i a_i \sum_j a_j^\dagger + (N-2) \sum_i a_i a_i^\dagger$$
$$\sim \sum_{l=0}^\infty (l+2)(l+1) c_l^\dagger c_l + \sum_{l=0}^\infty \sqrt{l+1} c_l^\dagger c_{l+1} \sum_{l'=0}^\infty \sqrt{l'+1} c_{l'+1}^\dagger c_{l'} + (N-2) \sum_{l=0}^\infty (l+1) c_l^\dagger c_l, \tag{A.12}$$

up to constant terms. Combined with other terms in (A.2), we have

$$:\text{Tr}[\bar{Z}^2 Z^2]: \sim \sum_{l=0}^\infty (l+2)(l+1) c_l^\dagger c_l - \sum_{l=0}^\infty \sqrt{l+1} c_l^\dagger c_{l+1} \sum_{l'=0}^\infty \sqrt{l'+1} c_{l'+1}^\dagger c_{l'} - (N+2) \sum_{l=0}^\infty (l+1) c_l^\dagger c_l, \tag{A.13}$$

again up to constant terms. To restore the constant term, one can demand the normal ordered operator to have zero expectation value in the AdS vacuum.

# B  Details of the numerical simulation in the $\frac{1}{4}$-BPS sector

In this appendix, we give a description of our method of doing the numerical simulation of the 1/4-BPS in Section 4. One can access the numerical data used in generating the plots at [87].

## B.1  Constructing a basis of Hilbert space

We start by constructing a set of linearly independent multi-traces that span the entire Hilbert space. We do this by first constructing all the possible multi-traces $w_i(Z, X), i = 1, ..., K$. Due to trace relations at finite $N$, these operators are not linearly independent and $K$ is much larger than the actual dimension of the Hilbert space, denoted by $n$. We will like to therefore find a minimal set of multi-trace words that spanned the Hilbert space. Note that such a set of words is non-unique, but any such set will be equally good for our purpose.

We need a way to pick out $n$ linearly independent words out of $\{w_1, w_2, ..., w_K\}$. In principle, one could expand out each word into the matrix elements, $Z = (z_{ij})_{N \times N}$, $X = (x_{ij})_{N \times N}$, and view each word as a vector in the space of monomials $\{\prod_{ij}(z_{ij})^{n_{ij}}(x_{ij})^{m_{ij}} | n_{ij}, m_{ij} \in \mathbb{Z}\}$. Then one can simply look for $n$ linearly independent vectors among $K$ of them. However, this would be difficult for large systems since the number of monomials grow extremely fast.

We use a different method to look for linearly independent words [110]. The essential idea is that we will be able to tell the linear relations between the words as long as we know their values evaluated on enough samples of $(Z_i, X_i)$, where $Z_i$ and $X_i$ are $N \times N$ matrices (traceless in the case of SU($N$)) with numeric entries which can be chosen to be random integers. For example, we can randomly draw $M$ samples of the matrices, $\{(Z_1, X_1), (Z_2, X_2), ..., (Z_M, X_M)\}$, and form a $K \times M$ matrix

$$\mathcal{W} = \begin{pmatrix} w_{1,1} & w_{1,2} & ... & w_{1,M} \\ w_{2,1} & w_{2,2} & ... & w_{2,M} \\ ... & ... & ... & ... \\ w_{K,1} & w_{K,2} & ... & w_{K,M} \end{pmatrix}, \quad \text{where} \quad w_{i,j} \equiv w_i|_{Z=Z_j, X=X_j}. \tag{B.1}$$

Then when $M \gtrsim n$, the rank of the matrix $\mathcal{W}$ will saturate at $n$, i.e. the number of independent words. The matrix $\mathcal{W}$ then contains all possible information about the linear relation between the words and one can find a complete basis of words from here. In practice, since $K \gg n$, it is not very efficient to study (B.1) directly, where we include all $K$ words at once. In our numerics, it turns out to be more efficient to look for new independent words successively, meaning that we start with a set of linearly independent words and add in new independent words one by one. The basic idea is still the same as above so we will not go into the details.

## B.2 Finding the action of the one-loop dilatation operator

After completing the steps in Section B.1, we are left with a set of $n$ linearly independent multi-traces $w^T = \{w_1, w_2, ..., w_n\}$. They form a complete basis for the Hilbert space, but they are not orthogonal to each other. In principle, one could try to compute their inner product and find an orthonormal basis, though this is not necessary for our purpose, as we will comment later. The main goal we need to achieve is to find out how $\mathcal{D}_2$ acts in this particular basis, or in other words, we want to find the concrete matrix $D_2$ given by equation

$$\begin{pmatrix} \mathcal{D}_2 w_1 & \mathcal{D}_2 w_2 & ... & \mathcal{D}_2 w_n \end{pmatrix} = \begin{pmatrix} w_1 & w_2 & ... & w_n \end{pmatrix} D_2. \tag{B.2}$$

There are various different ways one might try to approach this. We will adopt a "numerical" approach to this problem, similar to what we did in Section B.1. Instead of computing action of $\mathcal{D}_2$ in general, we try to evaluate the value of $\mathcal{D}_2$ acting on the words at specific values of $Z$ and $X$, $Z = Z_*, X = X_*$. By doing this multiple time, we can then "reconstruct" how the operator acts in general. For example, numerically, we replace the action of a matrix derivative $(\check{X})_{ji} = \partial_{X_{ij}}$ on a word $w_k$ by

$$\frac{\partial w_k}{\partial X_{ij}}\bigg|_{X=X_*, Z=Z_*} \approx \frac{1}{\epsilon}\left[w_k|_{Z=Z_*, X=X_*+\epsilon S_{ij}} - w_k|_{Z=Z_*, X=X_*}\right], \tag{B.3}$$

where $S_{ij}$ is a shift matrix with only the $i, j$ element being one and other elements being zero.[46] In (B.3) one would introduce some error that depends on $\epsilon$. However, one can easily improve it and make it exact by choosing $Z_*, X_*$ to have integer entries, from which we know that the left hand side of (B.3) should also be an integer. Therefore by choosing $\epsilon$ small enough, we

---

[46]The shift matrix needs to be slightly modified in the case of SU($N$) to incorporate the $1/N$ piece of (13).

can simply round up the right hand side of (B.3) to the closest integer and get the exact answer for the derivative.

Following the same logic, we can then readily compute $\mathcal{D}_2 w_i|_{X=X_*,Z=Z_*}$, $i = 1, 2, ..., n$. In order to reconstruct the matrix $D_2$ in (B.2), we need to repeat the computation $n$ times, at $n$ randomly drawn values of the matrices $\{(Z_1, X_1), (Z_2, X_2), ..., (Z_n, X_n)\}$. We can collect the results into two matrices

$$D_* = \begin{pmatrix} (\mathcal{D}_2 w)_{1,1} & (\mathcal{D}_2 w)_{1,2} & ... & (\mathcal{D}_2 w)_{n,1} \\ ... & ... & ... & ... \\ (\mathcal{D}_2 w)_{1,n} & (\mathcal{D}_2 w)_{2,n} & ... & (\mathcal{D}_2 w)_{n,n} \end{pmatrix}, \quad \text{where} \quad (\mathcal{D}_2 w)_{i,j} \equiv \mathcal{D}_2 w_i|_{Z=Z_j, X=X_j}, \quad \text{(B.4)}$$

and

$$\mathcal{W} = \begin{pmatrix} w_{1,1} & w_{1,2} & ... & w_{1,n} \\ ... & ... & ... & ... \\ w_{n,1} & w_{n,2} & ... & w_{n,n} \end{pmatrix}, \quad \text{where} \quad w_{i,j} \equiv w_i|_{Z=Z_j, X=X_j}. \quad \text{(B.5)}$$

and the matrix $D_2$ in (B.2) is given by

$$D_2^T = D_* \mathcal{W}^{-1}. \quad \text{(B.6)}$$

We can compute the action of the simple operator $O$ in a similar way.

Now that we've gotten the explicit form of $D_2$, we can simply diagonalize it and get the one-loop anomalous dimensions. Note that even though the basis $w^T = \{w_1, w_2, ..., w_n\}$ is not orthonormal, it is related to one by a similarity transformation $V$. As a consequence, the explicit form of the $D_2$ in basis $w^T = \{w_1, w_2, ..., w_n\}$ is not a real symmetric matrix, but it is related to one via a similarity transformation $V D_2 V^{-1}$. The precise form of $V$ is not necessary for the purpose of finding eigenvalues of $D_2$ [56].

By diagonalizing $D_2$, we get a set of eigenvalues $\delta E_i$, as well as the associated eigenvectors $\psi_i$. Note that since $D_2$ is non-Hermitian in the basis we are using, it's important that we are looking at its right eigenvectors. The one-loop primaries are given by $w^T \cdot \psi_i$. Apart from the degenerate BPS subspace, these operators are orthogonal to each other though they are not necessarily normalized. For the purpose of studying the spectrum of the LMRS operator, the normalization is not needed, though it might be needed in order to study some other quantities.

As we discussed in Section 4.2, it is important to perform desymmetrization properly in order to study level statistics. In numerics, we in fact first perform the desymmetrization and then perform the steps discussed in this section.

## B.3 Details on the data analysis

In the analysis of level statistics, it is important to perform unfolding first in order to remove the effect of the overall density of state [156]. Here we adopt the same unfolding procedure as in [56]. Given a set of eigenvalues $\{\varepsilon_1, ..., \varepsilon_n\}$, which could either be the eigenvalues of $\mathcal{D}_2$, or the eigenvalues of the LMRS operator, one defines the cumulative level number

$$n(\varepsilon) = \sum_{i=1}^{n} \theta(\varepsilon - \varepsilon_i), \quad \text{(B.7)}$$

where $\theta(x)$ is the Heaviside theta-function. One then performs a polynomial fit (with degree $p$) for $n(\varepsilon)$, with the fitted result denoted by $n_{\text{average}}(\varepsilon)$. The unfolded spectrum $\{\delta_1, ..., \delta_n\}$ is then given by

$$\delta_i = n_{\text{average}}(\varepsilon_i). \quad \text{(B.8)}$$

Table 1: We record the sectors we included in Figure 9 and the number of BPS states in each.

|  | $L = 19$ | $L = 20$ | $L = 21$ | $L = 22$ |
|---|---|---|---|---|
| $M = 5$ | 20 | 22 | 27 | 30 |
| $M = 6$ | 18 | 24 | 26 | 32 |
| $M = 7$ | / | / | 21 | 24 |

In fitting $n(\varepsilon)$, it is important that one does not overfit. In practice, we generally choose $p$ to be 4 or 5, which is much smaller than the number of data points.

In our analysis of the spectrum of the LMRS operator, we considered an ensemble including 10 different sectors. We only included sectors that have relatively more BPS states. Since we focused on the case of $N = 4$, the number of BPS states is quite limited. It might be that for the purpose of seeing the difference between the BPS and the non-BPS states, one does not really need to go to regime $L, M \sim N^2$, and in that way one can get significantly more states, but we have not done this analysis. We performed unfolding in each sector separately, computed level spacings and eventually put the results together to generate Figure 9. In Table 1 we list the sectors we included in Figure 9 and their corresponding number of BPS states. We notice a curious feature that the number of highest weight BPS states does not grow monotonically as one varies $M$ from 0 to $L/2$, instead - it reaches maximum at some value in between. This feature can also be seen in the U($N$) case by expanding the index in [75].

# C   Details on the twist operator

## C.1   More numerical details

This subsection is a companion of Section 6, which includes more details on the numerical implementation.

First, we define peaks in the histogram distribution by looking for local maxima. The precise definition involves a choice of binning and regularizing by some Gaussian filter to reduce noise. We choose a bin size of $\delta\widehat{\lambda} = 5 \times 10^{-3}$ and a standard deviation for the Gaussian filter that of $5\delta\widehat{\lambda}$. The resulting peaks seem quite robust at moderate $N$, see Figure 21.

Let $\pi_i$ and $\pi_{i+1}$ the eigenvalues of two neighboring peaks. To define the gap between peaks, we consider all eigenvalues between $\pi_i < \lambda < \pi_{i+1}$ and computing their nearest-neighbor spacings. Then the gap between peaks $\pi_i$ and $\pi_{i+1}$ is defined as the maximum nearest-neighbor spacing for eigenvalues within this range. In practice, for peaks near the center of the spectrum, the difference between the maximum and second-maximum nearest-neighbor spacing is large, which signals that we can confidently identify peaks. For peaks near the outskirts of the distribution, it is more difficulty to reliably identify peaks.

## C.2   Card shuffling

Here we explain the computation of the eigenvalues of $\widehat{\Sigma}_2$ for the case $C = 1$ (We will denote the collection of $C_{\ell_1,\ell_2}$ by $C$). It is convenient to view $\widehat{\Sigma}_2$ as an operator on $\mathcal{H}_{\mathrm{aux}}$ as in 93. Notice that in this context, $[U(g), \widehat{\Sigma}_2] = 0$ where $U(g)$ can be either the left-action of the group or the right-action of the group. In the analogy to the quantization of a rigid body (or a particle on a group manifold), one of these symmetries is the "space frame" rotation, while the other is "body frame". This implies that for each irrep of $R$ of $S_N$, we expect an eigenvalue $\lambda_R$ with a $(\dim R)^2$ degeneracy.

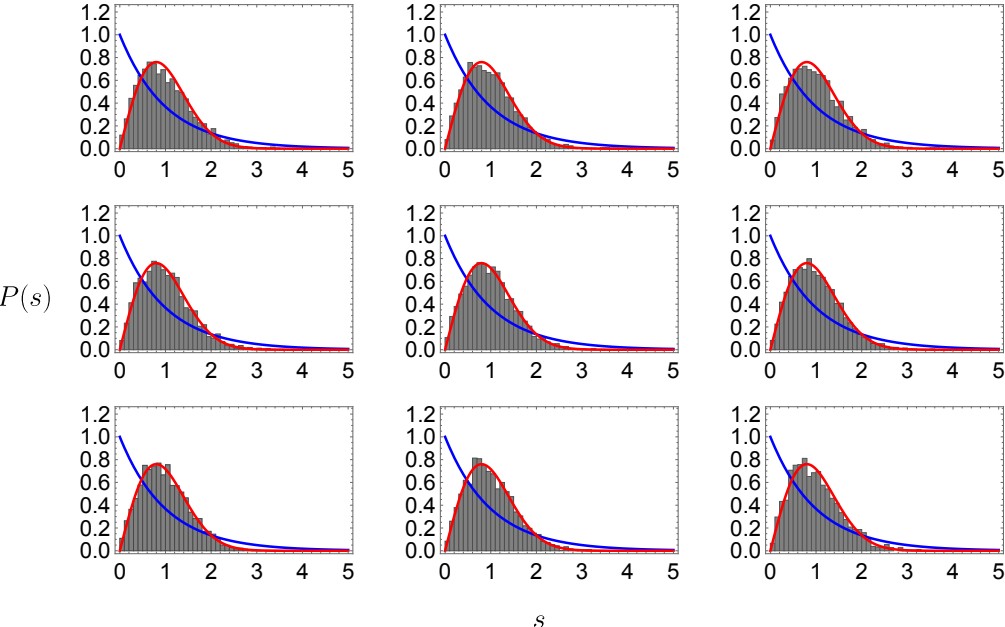

Figure 19: The eigenvalue separations for the 9 centermost clusters for $N = 44$.

Indeed, note that $\Sigma_2$ appearing in (93) is a convolution operator. Furthermore, let us define the Cayley graph of $S_N$: the vertices of the graph are the group elements of $S_N$, and there is an edge between $g, h$ iff $g = \tau h$ for some transposition $\tau$. The graph has a natural metric: the distance between $g_1$ and $g_2$ is the smallest number of edges that connect $g_1$ and $g_2$. Then the convolution operator (93) has support only on the shortest possible distance. It is therefore natural to view it as a discrete analog of a "kinetic term" for a particle on a group manifold.

Using the fact that (93) is a convolution, we can use the convolution theorem [124] to argue that its eigenvectors are the Fourier modes, or characters of the group $\chi_R(g)$. Furthermore, as we argued in the main text, we can project the matrix in to $\mathcal{H}_{\text{phys}}$, which removes the $(\dim R)^2$ degeneracy. The matrix then is

$$\widehat{\Sigma}_2([g], [h]) = \frac{1}{N!} \sum_R \frac{\chi_R(\tau)}{\dim R} \chi_R(g) \chi_R(h) |[h]| \,. \tag{C.1}$$

Here the sum is over irreducible representations $R$ of the symmetric group $S_N$. The factor $\binom{N}{2}$ is the number of transpositions. From the orthogonality of characters, we see that the eigenvalues of this matrix are just $\chi_R(\tau)/\dim R$. We checked that this matrix agrees with our expression (105), which is a separate check of the symmetry factors.

The eigenvalues that appear in this list have an additional degeneracy. We demonstrate this in Figure 22 for some moderate values of $N$. It would be interesting to understand this degeneracy, as it might provide a starting point to understanding the structure of the peaks that appear in the problem with $C \neq 1$, exhibited in Figure 16. (We checked however that the number of eigenvalues associated to a given "peak" are *not* in a 1-to-1 correspondence with the degeneracies that we computed here in the $C = 1$ problem.)

## C.3   A toy model with random $C$

We consider the operator $\widehat{\Sigma}_2$ defined in (105) and (104) but with each $C$ chosen at random from a Gaussian distribution over the reals with unit variance. We imposed the constraint

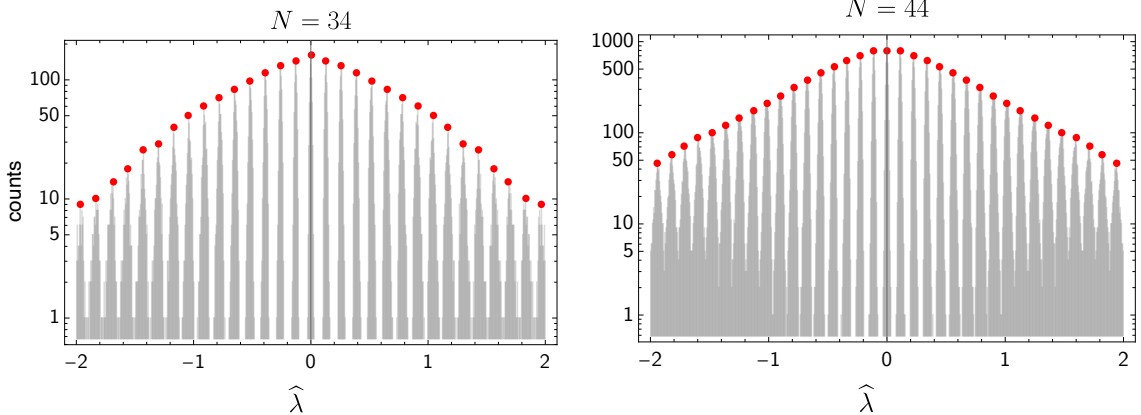

Figure 20: Identification of peaks in the $N = 34$ and $N = 44$ LMRS spectrum. For large values of $N$, and near the center of the spectrum, the peak detection algorithm is very robust, but for smaller values of $N \lesssim 10$, or near the edges of the spectrum, the peaks are not very well-defined.

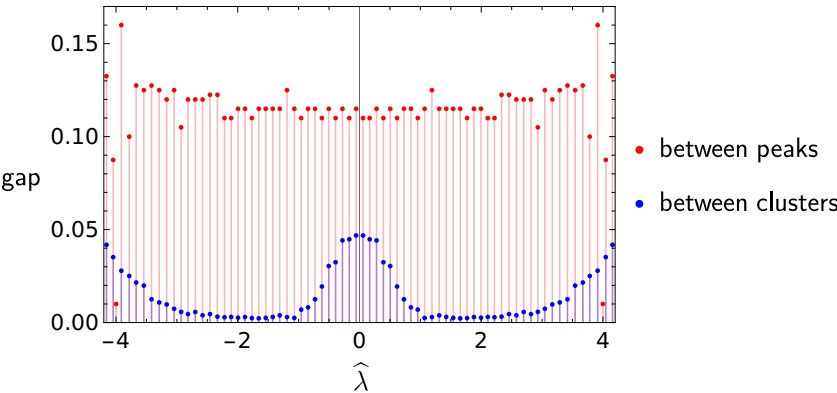

Figure 21: Identification of peaks in the $N = 44$ LMRS spectrum. We also show the distance between peaks (red) as well as the gap between clusters, defined as the biggest nearest-neighbor difference in eigenvalues between neighboring peaks. The distance between peaks is nearly constant in the $|\widehat{\lambda}| \lesssim 2$ range, whereas the separation between peaks varies significantly.

$C(\ell_1, \ell_2) = C(\ell_2, \ell_1)$. It seems plausible that by randomizing $C$'s, the operator should become more chaotic. We computed the Thouless time as follows.

We considered a Gaussian window $f(E)$ that defines a microcanonical ensemble for the spectral form factor. The GOE random matrix theory prediction for the ramp is:

$$\text{ramp} = \frac{t}{\pi} \int \mathrm{d}E \, f^2(E). \tag{C.2}$$

We normalized the operator so that the $\text{Tr}^2(C) = p(N)$. By averaging over a large number of random couplings, we computed the connected part of the spectral form factor, see Figure 23. After smoothing in time, we compared this to the predicted ramp and computed the time when the fractional error goes below some threshold (either 20% or 10%.) The points in Figure 24 were made by sampling $N_{\text{trials}} = 512$ different $C$'s for $N \leq 30$, 256 $C$'s for $N$ up to 36, and 128 $C$'s for $N$ up to 40.

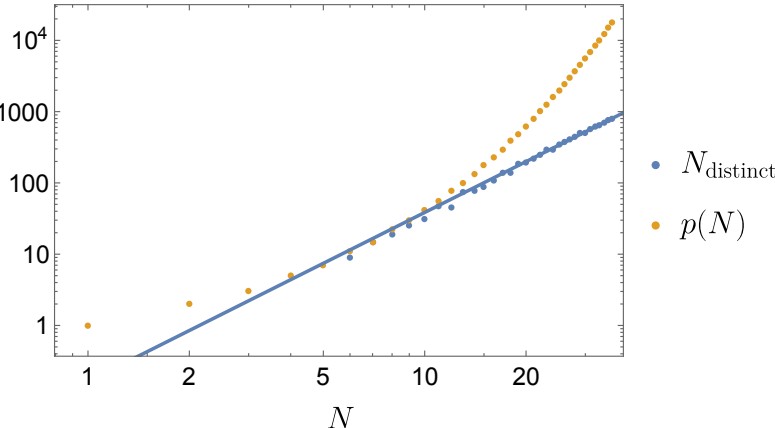

Figure 22: We plot both $\dim \mathcal{H}_{\text{phys}} = p(N)$ and the number of distinct eigenvalues $N_{\text{distinct}}$ as a function of $N$. We also show fit a power law to the $N \geq 10$ data which yields $N_{\text{distinct}} \sim N^{2.4}$, although it is unclear whether the exponent is reliable at these moderate values of $N$. This shows that at large $N$ there are many degeneracies in the spectrum.

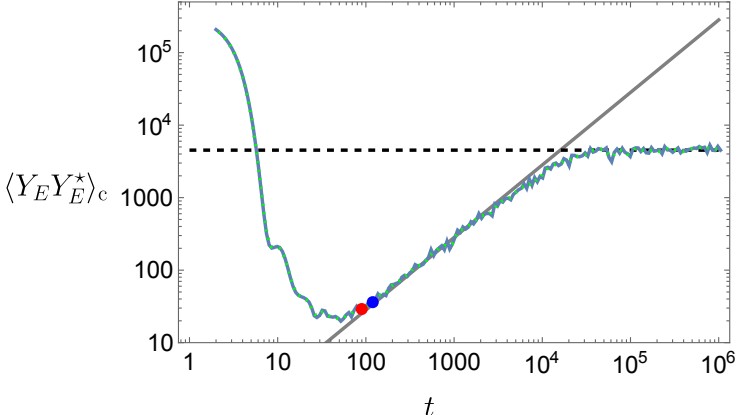

Figure 23: Microcanonical spectral form factor for $N = 34$ in the random $C$ problem. In dotted green, we plot the smoothed curve. We show the red and blue dots corresponding to the 10% and 20% error cutoffs that define the Thouless time estimate in Figure 24. The solid gray line is the predicted ramp, and the dashed black line is the predicted plateau from RMT. We averaged over 256 different instances of $\widehat{\Sigma}_2$.

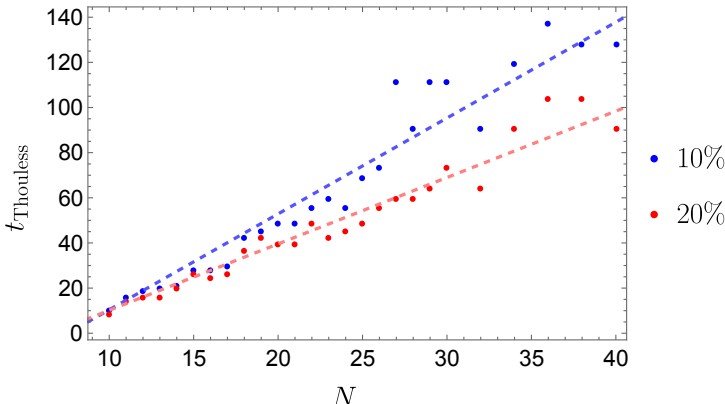

Figure 24: Computation of the Thouless time in the randomized $C$ problem. We show the best fit linear curves (with a constant offset). We considered a 10% error cutoff to define the Thouless time, as well as a less stringent 20% cutoff, which yields a shorter Thouless time.

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
