# Peer review of "BPS Chaos"

_SciPost Physics, doi:SciPost Phys. 18, 072 (2025)_

## Round 2 · Referee Report · Anonymous (Referee 1) · 2024-12-7

Report

This paper tackles an important problem about black hole microstates: the distinction between typical microstates of black holes and those of smooth horizonless geometries. It conjectures that within the BPS subspace, the former exhibits strong chaos while the latter is, at most, weakly chaotic. Supporting this conjecture, the paper presents evidence by applying the LMRS diagnostic of chaos to the 1/2 and 1/4 BPS sectors in N=4 SYM and the 1/2 BPS sector in the D1-D5 CFT. Furthermore, it suggests a potential connection between chaos in the BPS and non-BPS sectors via an analytic continuation of N.

I recommend the publication of this paper after the following points are addressed:

  1. On page 23, the author makes the assumption: "The Thouless time of the special banded matrix we are studying can be bounded below by the Thouless time of a banded random matrix of a similar shape." The author should clarify what is meant by a “special banded matrix.” This bound cannot universally apply to all banded matrices, as any Hermitian matrix can be transformed into a banded matrix through diagonalization using a unitary transformation.

  2. In Section 6, the author applies the LMRS diagnostic to the half-BPS states in the T^4 symmetric orbifold, projecting the twist operator (6.26) onto the subspace of half-BPS states. Since (6.26) is the operator that deforms the D1-D5 CFT away from the free orbifold point, the author should clarify the distinction between their computation and the computation of the anomalous dimension and mixing matrix in the half-BPS sector. This clarification is important because the half-BPS sector is protected under such a deformation.

Recommendation

Ask for minor revision

---

## Round 2 · Referee Report · Anonymous (Referee 2) · 2024-12-10

Strengths

  1. Provides a concrete, computable, implementation of the LMRS proposal for determining chaos in BPS states.
  2. Uses a range of techniques, different approaches and theories to produce a coherent, compelling picture of their claim that 1/2 and 1/4 BPS states do not have strong chaos.
  3. Gives new insight into the nature of extremal black holes and their relation to horizonless microstate geometries.
  4. Authors provide access to code and numerical data.

Report

In this work the authors consider quantum chaos in the context of supersymmetric gauge theory and holography. They propose that the presence of chaos in a subspace of BPS states can be detected by computing the statistical properties of the spectrum of operators projected into the subspace. They perform explicit analytical and numerical computations for scalar 1/2- and 1/4-BPS operators in N=4 SYM. They produce evidence for the absence of strong chaos in these sectors while arguing that strong chaos should appear in the 1/16-BPS sectors, in particular for the "fortuitous" operators which are BPS at only a specific value of the rank of the gauge group.

They similarly show the absence of strong chaos in the two-charge sector of the D1-D5 CFT at zero coupling. By means of the holographic duality, this provides evidence for the conjecture that BPS states corresponding to horizonless microstate geometries are distinct from chaotic macroscopic black holes.

The work is novel, of significant interest, carefully computed with multiple checks and well written. It is clearly suitable for publication.

Requested changes

  1. This is not necessary for publication, but it may be useful if the authors can provide clarification on the following. The authors seem to find that, for certain simple operators in both the 1/2 and 1/4-BPS sectors of N=4 SYM, the statistics are not only not chaotic, that is Wigner-Dyson, but Poisson. While this is of course perfectly consistent with the absence of strong chaos, it seems to raise the question of whether is there a notion of integrability in these sectors of the non-planar theory? Is there a reason to believe this is a merely a mirage?

Recommendation

Publish (surpasses expectations and criteria for this Journal; among top 10%)

---

## Round 3 · Referee Report · Anonymous (Referee 1) · 2025-1-22

Report

I recommend this paper for publication.

Recommendation

Publish (surpasses expectations and criteria for this Journal; among top 10%)

---

## Round 3 · Author Response

We would like to thank the referees for their insightful comments and questions. We have modified our manuscript to address their questions (see "List of changes").

---

## Round 3 · List of Changes

Referee 1:

- “1. On page 23, the author makes the assumption: "The Thouless time of the special banded matrix we are studying can be bounded below by the Thouless time of a banded random matrix of a similar shape." The author should clarify what is meant by a “special banded matrix.” This bound cannot universally apply to all banded matrices, as any Hermitian matrix can be transformed into a banded matrix through diagonalization using a unitary transformation.”

We agree with the referee that additional clarification is needed. The referee is correct to observe that even a typical member from random matrix ensemble can be brought into banded form. However, we note that if we transform a random matrix into banded form, it will have distinctive features that separate it from the kind of projected operators we can get in the 1/2-BPS subspace.

To address this question, we removed the vague adjective “special” from the sentence and added a paragraph below (3.33) to discuss the intended meaning of “special” in more detail, which we copy here:

“One might question our assumption that the banded matrix we have is less chaotic than a banded random matrix. After all, one can always transform a matrix, for instance a typical member of a random matrix ensemble, into a banded form. Such a banded matrix will then display stronger chaos than a banded random matrix.\footnote{One should not confuse the notion of “banded random matrix" with a random matrix that is transformed into banded form.} However, we note that if we transform a typical random matrix into banded form, the resulting matrix is expected to carry distinctive features which separate them from the matrices we have. For example, consider the banded matrix being simply diagonal. The diagonal form of a random matrix has its eigenvalues as entries, which are highly random numbers satisfying level repulsion, which differs from $\hat{O}$ in (3.26) whose entries are given by simple expressions.

As a more non-trivial example, we can consider the tridiagonal form $M_{tri-diagonal}$ of the Gaussian orthogonal ensemble. It was shown in \cite{Dumitriu_2002} that in the ensemble of transformed matrices, the entries in the matrices are drawn independently from specific probability distributions. The diagonal elements of $ M_{tri-diagonal}$ are drawn from normal distribution with the same standard deviation, while the $n$-th off-diagonal element is drawn from the $\chi_n$ distribution, resulting in a distinctive growing pattern along the off-diagonal. We expect the LMRS operators $\hat{O}$ in the $1/2$-BPS subspace to not display such features in the cases where they are tri-diagonal. Even though here we only discussed the special cases where the banded matrix is diagonal or tri-diagonal, we expect the distinctions between our matrices and random matrices that were transformed into banded form to exist more generally.”

- “2. In Section 6, the author applies the LMRS diagnostic to the half-BPS states in the T^4 symmetric orbifold, projecting the twist operator (6.26) onto the subspace of half-BPS states. Since (6.26) is the operator that deforms the D1-D5 CFT away from the free orbifold point, the author should clarify the distinction between their computation and the computation of the anomalous dimension and mixing matrix in the half-BPS sector. This clarification is important because the half-BPS sector is protected under such a deformation.”

The operator \Sigma in (6.26) that we are considering differs from the actual marginal operator \Sigma_{marginal} by \Sigma_{marginal} \sim G_{-1/2} \bar{G}_{-1/2} \Sigma. As the referee correctly pointed out, the matrix elements of \Sigma_{marginal} in half-BPS states vanish. However, the matrix elements of \Sigma do not vanish.

There is also an analogy with N=4 SYM. The twist operator we consider is analogous to Tr Z^2 in SYM. The marginal deformation in D1-D5 is a super-descendant of the twist operator, and similarly in SYM, the marginal deformation (the SYM interaction) is a super-descendant of Tr Z^2. But as is very familiar in SYM, there are non-zero 3-pt functions between Tr Z^2, Tr Z^k, and Tr Z^l.

We added a sentence after (6.26) for clarification:
“Note that the marginal operators $\Sigma_{marginal}$ which deform the theory away from the orbifold point are the superconformal descendants of $\Sigma$, i.e. $\Sigma_{marginal} \sim G_{-\frac{1}{2}}\bar{G}_{-\frac{1}{2}}\Sigma$, whose matrix elements between half-BPS states vanish due to non-renormalization theorems.”

Referee 2:

- “1. This is not necessary for publication, but it may be useful if the authors can provide clarification on the following. The authors seem to find that, for certain simple operators in both the 1/2 and 1/4-BPS sectors of N=4 SYM, the statistics are not only not chaotic, that is Wigner-Dyson, but Poisson. While this is of course perfectly consistent with the absence of strong chaos, it seems to raise the question of whether is there a notion of integrability in these sectors of the non-planar theory? Is there a reason to believe this is a merely a mirage?”

In our Figure 4, we showed an example in the 1/2-BPS sector where the projected operator whose nearest-neighbor spacings satisfy a distribution close to the Wigner surmise. We view this as evidence that the projected operators in general are not integrable, but only weakly chaotic. It does not rule out the possibility that for a special class of simple operators, such as that considered in Section 3.2, the projected operator can be integrable. We think it is an interesting future question to understand the notion of integrability in projected operators.

Other small changes:
1. Above equation (1.6), we modified misleading statement “In general, BPS states will be superconformal primaries, so we can …” into “In the situation where the BPS states are given by superconformal primaries, we can …”.

---

## Editorial Decision

published